# Transformers Learn the Optimal DDPM Denoiser for Multi-Token GMMs

**Hongkang Li** [1]   **Hancheng Min** [2]   **René Vidal** [1]

## Abstract

Transformer-based diffusion models have demonstrated remarkable performance at generating high-quality samples. However, our theoretical understanding of the reasons for this success remains limited. For instance, existing models are typically trained by minimizing a denoising objective, which is equivalent to fitting the score function of the training data. However, we do not know why transformer-based models can match the score function for denoising, or why gradient-based methods converge to the optimal denoising model despite the non-convex loss landscape. To the best of our knowledge, this paper provides the first convergence analysis for training transformer-based diffusion models. More specifically, we consider the population Denoising Diffusion Probabilistic Model (DDPM) objective for denoising data that follow a *multi-token Gaussian mixture* distribution. We theoretically quantify the required number of tokens per data point and training iterations for the global convergence towards the Bayes optimal risk of the denoising objective, thereby achieving a desired score matching error. A deeper investigation reveals that the self-attention module of the trained transformer implements a *mean denoising* mechanism that enables the trained model to approximate the oracle Minimum Mean Squared Error (MMSE) estimator of the injected noise in the diffusion steps. Numerical experiments validate these findings.

## 1. Introduction

Diffusion models (Sohl-Dickstein et al., 2015; Ho et al., 2020; Song & Ermon, 2019; Song et al., 2021) have achieved state-of-the-art performance across a wide range of generative AI tasks, including the creation of images (Rombach et al., 2022; Peebles & Xie, 2023), videos (Bar-Tal et al., 2024; Xing et al., 2024), audio (Kong et al., 2021; Zhang et al., 2023), text (Sahoo et al., 2024; Arriola et al., 2025), scientific data (Hoogeboom et al., 2022; Li et al., 2024d; Price et al., 2025), and multi-modal content (Ruan et al., 2023; Cai et al., 2025). A classical diffusion model consists of two stages: a forward process and a backward process. The forward process gradually transforms data into noise by adding white Gaussian noise, while the backward process learns a score-based model to progressively remove the injected noise and generate samples from noisy inputs. Specifically, score-based models are typically formulated as neural networks trained to approximate the score function, namely the gradient of the logarithm of the probability density of the data at each time step.

Among the various score-based diffusion models, the denoising diffusion probabilistic model (DDPM) (Ho et al., 2020) proposes a canonical training objective as the foundation of diffusion model training, which is to predict the added noise in the forward process. Early score-based generative models (Ho et al., 2020; Song et al., 2021) adopt convolution-based models, such as U-Net, as the backbone architectures for learning the score function. More recently, motivated by their superior scalability and stronger performance in visual generation, transformer-based architectures, such as DiT (Peebles & Xie, 2023), have served as effective alternatives. However, despite the remarkable empirical success of transformer-based diffusion models for score learning, the theoretical reasons for this success are much less explored. These include fundamental questions such as:

> *(Q1) Why can a nonlinear transformer match the score function and denoise?*

> *(Q2) Why gradient descent converges to the optimal nonlinear transformer under DDPM training?*

Existing theoretical work addresses these questions only in a limited and separate manner. One line of work (Wang et al., 2025; Li et al., 2024e) studies the optimal DDPM denoiser under specific data distributions via loss landscape analysis, but it does not establish convergence guarantees for training algorithms. Another line of work studies the training dynamics of neural networks for score matching (Han et al., 2024;

---

[1]Department of Electrical and Systems Engineering, University of Pennsylvania, Philadelphia, USA [2]Institute of Natural Sciences, School of Mathematical Sciences, and MOE-LSC, Shanghai Jiao Tong University, Shanghai, China. Correspondence to: Hongkang Li <lihk@seas.upenn.edu>.

*Proceedings of the $43^{rd}$ International Conference on Machine Learning*, Seoul, South Korea. PMLR 306, 2026. Copyright 2026 by the author(s).

| Theoretical Works | Network Model | Loss Landscape | Convergence Analysis | Denoising Mechanism |
| --- | --- | --- | --- | --- |
| Wang et al. (2025) | U-Net | Global optimum | ✗ | PCA |
| Han et al. (2024) | Fully-connected | N/A | ✓ | N/A |
| Wang et al. (2024) | Fully-connected | N/A | ✓ | N/A |
| Han et al. (2025) | Convolutional | Stationary point | ✗ | Balanced FL |
| Ours | Transformer | Global optimum | ✓ | Mean denoising |

Table 1. Comparison with existing works about training analysis and the optimality of denoising of diffusion models.

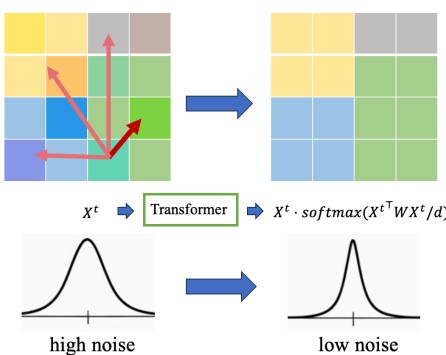

$X^t \Rightarrow \boxed{\text{Transformer}} \Rightarrow X^t \cdot softmax(X^{t\top}WX^t/d)$

high noise    low noise

Figure 1. Mean denoising mechanism by the trained Transformer. Attention reduces the noise added to the data. Dark (light) red arrows: attention weights between the query and key that share the same (different) pattern.

Wang et al., 2024; Han et al., 2025; Wang & Pehlevan, 2025; Bonnaire et al., 2026), but only for simple architectures or unrealistic regimes. As far as we know, none of these studies investigates the convergence of training algorithms or the learned denoising mechanism for Transformer-based diffusion models. Please see Section 1.2 for a more detailed comparison between our work and several representative papers, and Table 1 for a summary.

### 1.1. Main Contributions

To the best of our knowledge, this paper is the first to analyze the training dynamics of nonlinear transformers trained by gradient descent on the DDPM loss, providing theoretical convergence and score matching error guarantees. Motivated by empirical observations that data are composed of multiple patterns, we consider a Multi-Token Gaussian Mixture (MTGM) data distribution, where each data point consists of multiple tokens that are sampled from a given set of Gaussian mixture components. Following prior work (Allen-Zhu & Li, 2023; Li et al., 2023a; 2024a; Jiang et al., 2024; Li et al., 2025a; Han et al., 2025), we characterize how the transformer parameters learn the mean patterns of different Gaussian components through gradient updates. Our main contributions include:

1. **A quantitative analysis of how to optimize the DDPM loss with transformers towards convergence.** We theoretically analyze the training dynamics on a one-layer single-head transformer with softmax attention and quantify the number of training iterations and tokens per data

required to optimize the DDPM loss. Our results characterize how the convergence is affected by the imbalance among the proportions of different Gaussian components in the MTGM distribution, the number of distinct pattern types present in each data, and the time-averaged signal-to-noise ratio of the diffusion noising process.

2. **Theoretical characterization of how the trained transformer learns the oracle MMSE estimator as the optimal denoiser.** The major technical difficulty in analyzing why neural networks can converge to the optimal denoiser is that the true MMSE estimator for the DDPM loss is intractable to compute. To address this challenge, we define an *oracle MMSE estimator*, which is computed with the class of the Gaussian component of each token in the MTGM data as known. We then prove that the trained transformer can converge to this estimator. Moreover, we show that the oracle denoising risk corresponding to the oracle MMSE estimator is close to the true Bayes risk of the training problem if the number of tokens in each data is large enough. This implies that the training process globally converges to the optimal denoiser, and consequently, the trained model can be used to construct a score network that enables score matching.

3. **Theoretical understanding of how the self-attention structure performs denoising through a mean denoising mechanism.** The key challenge in characterizing how a transformer approximates the oracle MMSE estimator lies in explaining how the model parameters learn the mean patterns of an MTGM data distribution. We are the first to propose a *mean denoising* mechanism of self-attention in diffusion model training, i.e., attention aggregates queries and keys that share the same pattern, thereby producing a minimum-variance unbiased estimator (MVUE) of the mean pattern and effectively removing the noise injected by the forward diffusion process. This mechanism also enables the trained model to denoise data with the same Gaussian components but shifted mixture proportions, as long as each data point contains a sufficient number of tokens.

### 1.2. Related Work

**Theoretical analysis of diffusion models.** Recent work (Wang et al., 2025; Li et al., 2024e) analyzes the landcape of the DDPM denoiser and shows that the optimal diffusion

model essentially performs principal component analysis (PCA) for Gaussian data or low-rank Gaussian mixture data. While these results theoretically characterize the structure of optimal solutions, they do not establish whether such solutions are attainable through gradient-based training of neural networks. Wang & Pehlevan (2025); Bonnaire et al. (2026); Han et al. (2024); Wang et al. (2024) analyze the training dynamics of denoising score matching by considering linear models or by adopting theoretical assumptions that reduce nonlinear models to linear ones, such as random feature models (Rahimi & Recht, 2007) or the neural tangent kernel regime (Jacot et al., 2018). Boffi et al. (2025) study optimizing the score matching loss over infinitely wide shallow networks. Only Han et al. (2025) prove that, under the DDPM loss, a diffusion model parameterized by a two-layer convolutional neural network learns data features and noise to the same order, a phenomenon referred to as the balanced feature learning (FL) mechanism. However, their analysis does not provide a convergence guarantee for training dynamics and does not analyze transformers. Other works study convergence guarantees of DDPM samplers in terms of total variation or KL divergence (Li et al., 2023b; Chen et al., 2023; Azangulov et al., 2024; Huang et al., 2026; Li & Yan, 2024; Liang et al., 2025) or the generalization of diffusion models (Bonnaire et al., 2026; Li et al., 2025d; Sclocchi et al., 2025; Pham et al., 2025). These works do not involve the model training analysis and therefore differ from the focus of our paper.

**Optimization and generalization of transformers.** Many works study the optimization and generalization of transformers for supervised learning tasks. Jelassi et al. (2022); Li et al. (2023a; 2024b); Huang et al. (2025); Jiang et al. (2024) study the convergence of Transformer with a generalization guarantee for binary classification or linear regression tasks via feature learning analysis. Tarzanagh et al. (2023); Ataee Tarzanagh et al. (2023) show the gradient updates of weights or prompts converge to a max-margin SVM solution. None of these works involves convergence analysis of denoising tasks.

## 2. Problem Formulation

**Data distribution.** Each data point $\boldsymbol{X} = [\boldsymbol{x}_1, \cdots, \boldsymbol{x}_P] \in \mathbb{R}^{d \times P}$ contains $P$ *tokens* $\boldsymbol{x}_1, \cdots, \boldsymbol{x}_P$ in $\mathbb{R}^d$, each one sampled i.i.d. from a *Multi-Token Gaussian Mixture (MTGM)* distribution $\mathcal{D}(\tilde{\boldsymbol{\pi}}, K, \{\boldsymbol{\mu}_i\}_{i=1}^M, \rho^2)$, where $\{\boldsymbol{\mu}_i \in \mathbb{R}^d\}_{i=1}^M$ is a set of $M$ orthogonal *patterns*, i.e., $\boldsymbol{\mu}_i^\top \boldsymbol{\mu}_j = \sqrt{d}\delta_{ij}$; $\tilde{\boldsymbol{\pi}} \in \Delta^{M-1}$ is a vector in the probability simplex such that $\mathbf{1}^\top \tilde{\boldsymbol{\pi}} = 1$ and $\min_{m \in [M]} \tilde{\pi}_m > 0$; $K \leq M$ is the number of distinct patterns in $\boldsymbol{X}$; and $\rho^2 = \Theta(1)$ is the variance of Gaussian components. Specifically,

**Definition 1.** $\boldsymbol{X} = (\boldsymbol{x}_1, \cdots, \boldsymbol{x}_P) \sim \mathcal{D}(\tilde{\boldsymbol{\pi}}, K, \{\boldsymbol{\mu}_i\}_{i=1}^M, \rho)$ *is sampled according to the following procedure:*

1. *Sample* $\boldsymbol{Z} \sim \mathrm{Unif}(\{\boldsymbol{z} \in \{0,1\}^M : 0 < \|\boldsymbol{z}\|_0 = K < M\})$;

2. *Let* $\boldsymbol{\pi}(\boldsymbol{Z}) = (\pi_1, \cdots, \pi_M)$ *with* $\pi_i(\boldsymbol{Z}) = [\boldsymbol{Z}]_i \tilde{\pi}_i / \boldsymbol{Z}^\top \tilde{\boldsymbol{\pi}}$;

3. *For each* $p \in [P]$, *sample* $Y_p | \boldsymbol{Z} \sim \mathrm{Categorical}_M(\boldsymbol{\pi}(\boldsymbol{Z}))$, *and then sample* $\boldsymbol{x}_p | Y_p \sim \mathcal{N}(\boldsymbol{\mu}_{Y_p}, \rho^2 \boldsymbol{I})$.

**Remark 1.** *Definition 1 is an extension of the Gaussian mixture distribution. When $K = 1$ and $P = 1$, each data point contains one token sampled from a Gaussian Mixture Model with mutually orthogonal cluster centers, which is frequently used in theoretical studies of training neural networks for classification tasks (Min & Vidal, 2025; Shen et al., 2025). When $K > 1$, each data point consists of multiple patterns, a common, albeit simplified, assumption in computer vision to model an image composed of $K$ out of $M$ possible objects. This is because real data often exhibit feature representations that cluster around a limited number of modes. Moreover, to represent recurring semantic or textural patterns, like sky or grass in images, we formulate that multiple tokens correspond to the same pattern.*

**Learning model.** Let $\boldsymbol{f}(\Psi; \boldsymbol{X}, t) \in \mathbb{R}^{d \times P}$ be the output of the learning model, where $\Psi$ is the set of model parameters, $\boldsymbol{X}$ is the input to the model, and $t$ is the diffusion time step. We assume $\boldsymbol{f}$ is a one-layer single-head Transformer with parameters $\Psi = \{\boldsymbol{W}, \{v_t\}_{t=1}^T\} \in \mathbb{R}^{d \times d} \times \mathbb{R}^T$, i.e.,

$$\boldsymbol{f}(\Psi; \boldsymbol{X}, t) = (\boldsymbol{f}(\Psi; \boldsymbol{X}, t)_1, \cdots, \boldsymbol{f}(\Psi; \boldsymbol{X}, t)_P),$$
$$\boldsymbol{f}(\Psi; \boldsymbol{X}, t)_p = v_t(\boldsymbol{x}_p - \boldsymbol{X}\,\mathrm{softmax}(\frac{\boldsymbol{X}^\top \boldsymbol{W} \boldsymbol{x}_p}{d})), \quad (1)$$

where $\mathrm{softmax}([a_1, \cdots, a_P]^\top) = [e^{a_1}, \cdots, e^{a_P}]^\top / (\sum_j e^{a_j})$. This softmax function applies to a matrix column-wise, i.e., $\mathrm{softmax}([\boldsymbol{a}_1, \cdots, \boldsymbol{a}_P]) = [\mathrm{softmax}(\boldsymbol{a}_i)]_{i \in [P]}$. (1) contains a residual connection and an output-value layer fixed to $-\boldsymbol{I}$, making our learning model more complex than a single attention head.

**Training objective and algorithm.** We use the training objective of the Denoising Diffusion Probabilistic Model (DDPM) (Ho et al., 2020). For a given time step $t \in [T]$ and an input data point $\boldsymbol{X}^0$, we sample

$$\boldsymbol{X}^t = \sqrt{\bar{\alpha}_t}\boldsymbol{X}^0 + \sqrt{1 - \bar{\alpha}_t}\boldsymbol{E}^t, \quad (2)$$

where $\boldsymbol{E}^t = (\boldsymbol{\epsilon}_1^t, \cdots, \boldsymbol{\epsilon}_P^t) \in \mathbb{R}^{d \times P}$ is the additive white Gaussian noise with $\boldsymbol{\epsilon}_p \overset{i.i.d.}{\sim} \mathcal{N}(0, \boldsymbol{I}_d)$, and $\{\bar{\alpha}_t\}_{t=1}^T$ is the pre-determined noise scheduling coefficients. Given a sample data point $\boldsymbol{X}^0 \sim \mathcal{D}(\tilde{\boldsymbol{\pi}}, K, \{\boldsymbol{\mu}_i\}_{i=1}^M, \rho)$ and time step $t \sim \mathrm{Unif}([T])$, $\boldsymbol{X}^t$ is obtained from (2). We then minimize the following per-dimension DDPM loss in expectation as introduced in Ho et al. (2020); Bonnaire et al. (2026):

$$L(\Psi) = \sum_{t=1}^T \mathbb{E}_{\boldsymbol{X}^0, \boldsymbol{E}^t}[\|\boldsymbol{f}(\Psi; \boldsymbol{X}^t, t) - \boldsymbol{E}^t\|_F^2 / (2dPT)].$$
$$(3)$$

The above learning objective (3) is minimized via gradient descent with a learning rate $\eta > 0$. That is, at the training step $s = 0$, we set $\boldsymbol{W}^{(0)} = 0$ and randomly initialize each $v_t^{(0)}$, $t \in [T]$ from $\mathcal{N}(0, 1/d)$. Then, for each training iteration $s$, the parameters are updated as follows:

$$
\begin{aligned}
\boldsymbol{W}^{(s+1)} &= \boldsymbol{W}^{(s)} - \eta \nabla_{\boldsymbol{W}} L(\Psi), \\
v_t^{(s+1)} &= v_t^{(s)} - \eta \nabla_{v_t} L(\Psi), \forall t \in [T].
\end{aligned}
\tag{4}
$$

**Score matching.** Let $p_t(\boldsymbol{X}^t)$ be the probability density function of $\boldsymbol{X}^t$ at time step $t \in [T]$. The *score function* is defined as $\boldsymbol{s}(\boldsymbol{X}^t, t) = \nabla_{\boldsymbol{X}^t} \log p_t(\boldsymbol{X}^t)$. The goal of score matching is to train a neural network $\boldsymbol{s}_\theta(\boldsymbol{X}^t, t)$ parameterized by $\theta$ that minimizes the score matching error:

$$
\mathcal{E}(\theta) = \sum_{t=1}^{T} \mathbb{E}_{\boldsymbol{X}^0, \boldsymbol{E}^t} [\|\boldsymbol{s}_\theta(\boldsymbol{X}^t, t) - \boldsymbol{s}(\boldsymbol{X}^t, t)\|_F^2 / (2dPT)].
\tag{5}
$$

Note that we use the per-dimension score matching error in (5) following (Bonnaire et al., 2026).

## 3. Main Theoretical Results

Let us first introduce the Bayes risk for the DDPM objective.

**Definition 2.** *The **Bayes denoising risk** for the MTGM data model in Definition 1 and $t \sim \mathrm{Unif}([T])$ is defined as:*

$$
R_{Bayes} := \mathbb{E}_{\boldsymbol{X}^0, \boldsymbol{E}^t, t} [\|\boldsymbol{E}^t - \mathbb{E}[\boldsymbol{E}^t | \boldsymbol{X}^t]\|_F^2 / (2dP)].
\tag{6}
$$

$R_{\mathrm{Bayes}}$ is the optimal risk achieved if one can minimize the DDPM loss over all possible denoising models. Our main results will show that under certain conditions, the gradient descent in (9) can learn a transformer model $\boldsymbol{f}(\Psi; \boldsymbol{X}, t)$ that attains a risk of $R_{\mathrm{Bayes}} + O(\epsilon)$ for any arbitrarily small $\epsilon$.

To state our results, we need some additional notation. For $\boldsymbol{X}^0 \sim \mathcal{D}(\tilde{\boldsymbol{\pi}}, K, \{\boldsymbol{\mu}_i\}_{i=1}^M, \rho)$ with $\boldsymbol{Y}$ the latent variable in Definition 1, let the *minimal average pattern ratio* for pattern $u \in [M]$ and the *pattern imbalance ratio* be defined as:

$$
\nu_{\min}^{\tilde{\boldsymbol{\pi}}}(K) = \min_{u \in [M]} \nu_u^{\tilde{\boldsymbol{\pi}}}(K) := \min_{u \in [M]} \Pr(Y_1 = u)
\tag{7}
$$

$$
\delta(\tilde{\boldsymbol{\pi}}) = \min_{u \in [M]} \tilde{\pi}_u / \max_{u \in [M]} \tilde{\pi}_u.
\tag{8}
$$

The former is the minimum average probability of selecting a pattern, and the latter measures the degree of imbalance between the prior probabilities of different patterns. Since $\bar{\alpha}_t/(1 - \bar{\alpha}_t)$ is computed as the signal-noise ratio at time step $t$ by (Luo, 2022), we denote $\mathrm{SNR} = \mathbb{E}_t[\bar{\alpha}_t/(1 - \bar{\alpha}_t)]$ as the *time-averaged SNR* over the noise schedule.

With the above definitions and notations, we now state the following theoretical result about the convergence of diffusion model training.

**Theorem 1** (Convergence). *For any $\epsilon \in (0, \delta(\tilde{\boldsymbol{\pi}})^{\Theta(1)})$, if (i) the dimension $d \geq \Omega(\epsilon^{-1} \log(\epsilon^{-1} \nu_{\min}^{\tilde{\boldsymbol{\pi}}}(K)^{-1}))$, (ii)*

*the number of diffusion steps $T \geq \Omega(\log d)$, (iii) the data distribution satisfies $P \geq \Omega(K \nu_{\min}^{\tilde{\boldsymbol{\pi}}}(K)^{-1} M^{-1}(\rho^2 + 1)\epsilon^{-1} \log d)$, (iv) the algorithm in (4) is run with a step size $\eta \leq O((\max\{\rho, 1\} + \epsilon)^{-1})$ and (v) with a number of iterations*

$$
\begin{aligned}
S = \Omega\big( (\epsilon^{-1} + \nu_{\min}^{\tilde{\boldsymbol{\pi}}}(K)^{-3})\eta^{-1} \nu_{\min}^{\tilde{\boldsymbol{\pi}}}(K)^{-1} \\
\cdot SNR^{-3} + \log(\rho^2 + 1)\epsilon^{-1} \big),
\end{aligned}
\tag{9}
$$

*then with high probability over the Gaussian random initialization, the learned model with parameters $\Psi^{(S)}$ satisfies*

$$
L(\Psi^{(S)}) \leq R_{Bayes} + O(\epsilon).
\tag{10}
$$

Theorem 1 shows that, if each data contains a *sufficiently large number of tokens* and *the number of training iterations is large enough*, then the one-layer single-head transformer diffusion model trained by gradient descent (4) will achieve a DDPM loss that deviates from the Bayes denoising risk $R_{\mathrm{Bayes}}$ by only $O(\epsilon)$.

We shall elaborate those conditions. The requirement of the dimension in condition (i) is to ensure that the difference between attention weights of query-key pairs with the same pattern is small, as will be described in Section 4.2.

**Number of tokens.** The required number of tokens per data point in condition (iii) scales linearly in $\nu_{\min}^{\tilde{\boldsymbol{\pi}}}(K)^{-1}$ and $\rho^2$. Therefore, a less uniform distribution over the patterns and a higher noise level in a single data point increases the complexity of the denoising task, requiring more tokens in the data to learn an effective denoising model. The reason why more tokens per data benefits denoising will be explained in more detail in Section 4.

**Number of iterations.** The required number of GD iterations in condition (v) scales polynomially in $\nu_{\min}^{\tilde{\boldsymbol{\pi}}}(K)^{-1}$ and $\mathrm{SNR}^{-1}$. This means that a more uniform distribution over the patterns and a larger time-averaged SNR in the forward process help the model parameters learn all the patterns so that the self-attention identifies tokens sampled from the same Gaussian cluster, which is a core mechanism for denoising the MTGM data (discussed in Section 4.2). Moreover, we can derive the following simplification for (9) regarding different choices of $K \in [M]$.

**Corollary 1.** *(a) When $K = 1$, the number of iterations reaches its minimum, which leads to $S = \Omega((\epsilon^{-1} + M^3)\eta^{-1} M \cdot SNR^{-3})$. (b) When $K = M$, the required number of iterations reaches its maximum, which results in $S = \Omega((\epsilon^{-1} + \min_{u \in [M]} \{\tilde{\pi}_u\}^{-3})\eta^{-1} \min_{u \in [M]} \{\tilde{\pi}_u\}^{-1} SNR^{-3})$.*

Notice that $\min_{u \in [M]} \{\tilde{\pi}_u\} < 1/M$, which leads to $M^{-1} \min_{u \in [M]} \{\tilde{\pi}_u\}^{-1} > 1$, then Corollary 1 indicates that the required number of iterations for $K = M$ is at least $\Omega(M^{-1} \min_{u \in [M]} \{\tilde{\pi}_u\}^{-1})$ times larger than that for

$K = 1$. Therefore, the simplest pattern structure in the data, i.e., the case of $K = 1$, leads to the fastest convergence. This is also aligned with the previous intuition that more diverse patterns in each data increase the complexity of the denoising task, making the training more challenging.

**Constructing score model from learned denoiser.** Next, we show how the trained transformer model can achieve a desired score matching error.

**Theorem 2** (Score Matching). *Given the trained model in Theorem 1 with parameters $\Psi^{(S)}$ that satisfies (10) for some $\epsilon \in (0, \delta(\tilde{\pi})^{\Theta(1)})$, we can construct*

$$s_\theta(\boldsymbol{X}^t, t) = s_{\Psi^{(S)}}(\boldsymbol{X}^t, t) = -\frac{\boldsymbol{f}(\Psi^{(S)}; \boldsymbol{X}^t, t)}{\sqrt{1 - \bar{\alpha}_t}}, \quad (11)$$

*with $\theta = \Psi^{(S)}$, such that*

$$\mathcal{E}(\theta) = \mathcal{E}(\Psi^{(S)}) \leq \epsilon \cdot (SNR + 1). \quad (12)$$

Theorem 2 shows that a model trained under the conditions in Theorem 1 can be directly used to construct a score network that can achieve a score matching error of $O(\epsilon)$. Note that the construction in (11) is typically used to fit the conditional score function $\nabla_{\boldsymbol{X}^t} \log p_t(\boldsymbol{X}^t | \boldsymbol{X}^0)$. Theorem 2 demonstrates that, under our problem setting, (11) can also match $\nabla_{\boldsymbol{X}^t} \log p_t(\boldsymbol{X}^t)$ with an error that is close to 0.

# 4. In-Depth Analysis of Convergence and Denoising With the Trained Transformer

This section investigates why the trained Transformer can reduce the DDPM loss in (3) to the Bayes denoising risk and enable score learning, as stated in Theorem 1 and Theorem 2. In Section 4.1, we show that the Transformer $\boldsymbol{f}(\Psi)$ learns the "oracle MMSE estimator" through GD training, thereby achieving the oracle denoising risk up to an excessive $O(\epsilon)$ risk. In Section 4.2, we show that the learned self-attention structure exhibits a mean denoising mechanism to enable denoising and score matching on the MTGM data even with different Gaussian mixture proportions from training.

## 4.1. What Does the Trained Transformer Converge to?
The optimal model that achieves the Bayes denoising risk in (2) is computed as $\mathbb{E}[\boldsymbol{E}^t | \boldsymbol{X}^t]$, which is the Minimum Mean Squared Error (MMSE) estimator of the added Gaussian noise given a noisy input at time step $t$. However, computing $\mathbb{E}[\boldsymbol{E}^t | \boldsymbol{X}^t]$ for the MTGM data is challenging due to the highly complicated probability density function of $p_t(\boldsymbol{X}^t)$. In our problem setting, we define another estimator of the added noise with the data mean known, which is easier to obtain and does not need the knowledge of $p_t(\boldsymbol{X}^t)$. The specific definition is as follows.

**Definition 3** (Oracle MMSE estimator and denoising risk). *With Definition 1, let $\boldsymbol{M}_{\boldsymbol{Y}} = (\boldsymbol{\mu}_{Y_1}, \cdots, \boldsymbol{\mu}_{Y_P}) \in \mathbb{R}^{d \times P}$ be the matrix of mean patterns given $\boldsymbol{Y}$. Then, for a noisy data $\boldsymbol{X}^t$ obtained from some $\boldsymbol{X}^0$ with latent variable $\boldsymbol{Y}$, we define the **oracle MMSE estimator** of $\boldsymbol{E}^t$ given $\boldsymbol{M}_{\boldsymbol{Y}}$ as*

$$\mathbb{E}[\boldsymbol{E}^t | \boldsymbol{X}^t, \boldsymbol{M}_{\boldsymbol{Y}}] = \frac{\sqrt{1 - \bar{\alpha}_t}}{1 - \bar{\alpha}_t + \rho^2 \bar{\alpha}_t} (\boldsymbol{X}^t - \sqrt{\alpha_t} \boldsymbol{M}_{\boldsymbol{Y}}). \quad (13)$$

*We define the **oracle denoising risk**, denoted by $R_{oracle}$, as the DDPM loss with the oracle MMSE estimator as the denoising model for $t \sim \text{Unif}([T])$, which is computed as*

$$R_{oracle} := \mathbb{E}_{\boldsymbol{X}^0, \boldsymbol{E}^t, t}[\|\boldsymbol{E}^t - \mathbb{E}[\boldsymbol{E}^t | \boldsymbol{X}^t, \boldsymbol{M}_{\boldsymbol{Y}}]\|_F^2 / (2dP)]. \quad (14)$$

Note that $R_{\text{oracle}}$ is a lower bound of $R_{\text{Bayes}}$, because providing the prior knowledge of $\boldsymbol{M}_{\boldsymbol{Y}}$ gives more information than conditioning on $\boldsymbol{X}^t$ alone. Under squared loss, more information cannot worsen the optimal estimator.

**Convergence to the oracle MMSE estimator.** We then show that a one-layer, single-head Transformer trained under conditions in Theorem 1 approximates the oracle MMSE estimator. The following proposition reveals the implicit mechanism learned by the trained model.

**Proposition 1.** *Given training conditions (i)-(v) in Theorem 1, with a high probability over random initialization, the algorithm in (4) returns a model with parameters $\Psi^{(S)} = \{\boldsymbol{W}^{(S)}, \{v_t^{(S)}\}_{t=1}^T\}$ with the following properties:*

*1. The self-attention module satisfies that with a high probability over the sampling of the clean data $\boldsymbol{X}^0$ and the noise $\boldsymbol{E}$, for any computed noisy data $\boldsymbol{X}^t, t \in [T]$ (together with latent variable $\boldsymbol{Y}$), we have*

$$\frac{\|\sqrt{\bar{\alpha}_t} \boldsymbol{M}_{\boldsymbol{Y}} - \boldsymbol{X}^t \text{softmax}(\boldsymbol{X}^{t\top} \boldsymbol{W}^{(S)} \boldsymbol{X}^t / d)\|_F^2}{dP} \quad (15)$$
$$\leq O((\rho^2 + 1) \log d / (P\nu_{\min}^{\tilde{\pi}}(K))),$$

*2. The output weights satisfy that $\forall t \in [T]$,*

$$|v_t^{(S)} - \sqrt{1 - \bar{\alpha}_t} / (1 - \bar{\alpha}_t + \rho^2 \bar{\alpha}_t)| \leq O(\epsilon). \quad (16)$$

Recall that our Transformer model is defined as

$$\boldsymbol{f}(\Psi; \boldsymbol{X}^t, t) = v_t \big(\boldsymbol{X}^t - \boldsymbol{X}^t \text{softmax}(\boldsymbol{X}^{t\top} \boldsymbol{W} \boldsymbol{X}^t / d)\big),$$

then (15) and (16) in Proposition 1 show that the trained model $\boldsymbol{f}(\Psi^{(S)})$ can approximate (13), i.e., the oracle MMSE estimator under known mean pattern $\boldsymbol{M}_{\boldsymbol{Y}}$ of each data $\boldsymbol{X}^0$ with a diminishing error. Specifically, (15) indicates that the approximation error of $\boldsymbol{M}_{\boldsymbol{Y}}$ decreases with an increasing total number of tokens $P$. A larger $P$ can reduce the estimation variance introduced by the noise added through the diffusion process. This means that if condition (iii) in Theorem 1 holds, the trained self-attention structure $\boldsymbol{X}^t \text{softmax}(\boldsymbol{X}^{t\top} \boldsymbol{W}^{(S)} \boldsymbol{X}^t / d)$ can approximate $\sqrt{\bar{\alpha}_t} \boldsymbol{M}_{\boldsymbol{Y}}$ with a squared error of $O(\epsilon)$ per dimension (The key mechanism behind such an approximation is discussed in Section

4.2). Then, (16) implies that the trained $v_t^{(S)}$ for any $t \in [T]$ can approximate the linear coefficient term in(13) with an $O(\epsilon)$ error. Finally, one can conclude with (15), (16) that $\boldsymbol{f}(\Psi^{(S)})$ can approximate the oracle MMSE estimator (13) in the sense that they achieve similar denoising risks, i.e.,

**Corollary 2.** *The trained model with parameter $\Psi^{(S)}$ in Proposition 1 satisfies that $L(\Psi^{(S)}) \leq R_{oracle} + O(\epsilon)$.*

**Trained Transformer achieves near-optimal denoising.** Recall that in the DDPM loss (3) used for training, the mean pattern of each data $\boldsymbol{M_Y}$ is unknown, so in principle the best attainable loss after training should be $R_{\text{Bayes}}$ as in (10). However, (15) has a straightforward but important implication: since the latent mean patterns $\boldsymbol{M_Y}$ can be reliably estimated from the observed noisy data $\boldsymbol{X}^t$ at every diffusion time steps when the total number of tokens per data is large, the Bayes risk in (6) with only the knowledge of $\boldsymbol{X}^t$ should not be much worse than the oracle one in (14). Indeed, it can be shown independently that

**Proposition 2.** *Given condition (iii) in Theorem 1 hold for some $\epsilon \in (0, \delta^{\Theta(1)})$, we can obtain*

$$R_{Bayes} - R_{oracle} \leq O(\epsilon). \qquad (17)$$

Combining Corollary 2 and Proposition 2, one obtains our main Theorem 1, showing that the trained transformer achieves near optimal denoising risk.

**Remark 2.** *As illustrated in this Section 4.1, we have taken a novel approach to characterizes the near-optimal denoising model for the MTGM data. Prior work, Wang et al. (2025) for example, considers much simpler data distributions such as low-rank Gaussian mixture model, for which the true MMSE estimator $\mathbb{E}[\boldsymbol{E}^t | \boldsymbol{X}^t]$ can be analyzed directly. For MTGM data, however, the true MMSE estimator has a complicated expression due to the additional pattern subset selection step in the data sampling procedure (as in Definition 1). To address this challenge, our work adopts the oracle MMSE estimator, which has a simple interpretable expression and approximates the true MMSE estimator closely in the large-$P$ regime, as a bridge to characterize the near-optimal denoising model, thereby showing that Transformer can indeed learn the optimal denoiser.*

### 4.2. What Mechanism Does the Trained Transformer Parameters Learn from Diffusion Model Training?

In this section, we delve into the question of what denoising mechanism the Transformer parameters learn during training, which enables the model to approximate the oracle MMSE estimator and exhibit a desired performance of score matching. Recall the expressions for the transformer model in (1) and the oracle MMSE estimator in (13), one core question is why the self-attention structure in (1) can approximate the mean patterns $\sqrt{\alpha_t}\boldsymbol{M_Y}$ at diffusion time step $t$, which we shall explain carefully next.

**Query-key inner products reveal tokens with the same pattern.** First, the following proposition reveals how the trained self-attention behaves on different input data.

**Proposition 3.** *Consider the trained model in Theorem 1 with parameters $\Psi^{(S)} = \{\boldsymbol{W}^{(S)}, \{v_t^{(S)}\}_{t=1}^T\}$ that satisfies (10) for some $\epsilon \in (0, \delta(\tilde{\boldsymbol{\pi}})^{\Theta(1)})$. With a high probability over the sampling of the clean data $\boldsymbol{X}^0$ and the noise $\boldsymbol{E}$, any computed noisy data $\boldsymbol{X}^t, t \in [T]$ (together with latent variable $\boldsymbol{Y}$) satisfies that for any triplet $i, j, k \in [P]$ whose corresponding latent variables satisfy that $Y_i = Y_j \neq Y_k$, we have*

$$\boldsymbol{x}_j^{t\top}\boldsymbol{W}^{(S)}\boldsymbol{x}_i^t/d \geq \log\left(\Omega(\epsilon^{-1}K\delta(\tilde{\boldsymbol{\pi}}))\right)/2, \quad (18)$$

$$|\boldsymbol{x}_k^{t\top}\boldsymbol{W}^{(S)}\boldsymbol{x}_i^t/d| \leq O(\log d/\sqrt{d}) \cdot \boldsymbol{x}_j^{t\top}\boldsymbol{W}^{(S)}\boldsymbol{x}_i^t/d. \quad (19)$$

Proposition 3 shows that, on the one hand, if the query and key vectors are two tokens sampled with the same pattern, which is indicated by their latent variables $Y_i = Y_j$, then their inner product admits a large lower bound of order $\log\Omega(\epsilon^{-1})$ after training. On the other hand, if the query and key vectors are tokens with two different patterns, then the absolute value of their inner product is relatively small, which is on the order of $O(d^{-1/2})$ times that of a query–key pair with the same pattern. This implies that, even though each token of $\boldsymbol{X}^t$ contains a large amount of noise injected by the forward diffusion process, the trained self-attention layer can still capture and pair tokens with the same pattern.

**Softmax attention concentration enables mean denoising.** Based on Proposition 3, we then compute the weights output by the softmax attention of the trained model to introduce the mean denoising mechanism in the following corollary.

**Corollary 3** (Mean denoising mechanism). *For any data $\boldsymbol{X}^0 \sim \mathcal{D}(\tilde{\boldsymbol{\pi}}, \{\boldsymbol{\mu}_i\}_{i=1}^M, \rho)$ with a latent variable $\boldsymbol{Y}$, denote $\mathcal{S}_u^{\boldsymbol{Y}} = \{p \in [P] : Y_p = u\}$ for any $u \in [M]$. Then, given the same trained model as in Proposition 3, with a high probability over the sampling of the clean data $\boldsymbol{X}^0$ and the noise $\boldsymbol{E}$, for any computed noisy data $\boldsymbol{X}^t, t \in [T]$ (together with latent variable $\boldsymbol{Y}$), $u \in [M]$, and $p \in \mathcal{S}_u^{\boldsymbol{Y}}$, $p', p'' \in \mathcal{S}_u^{\boldsymbol{Y}}$, $p' \neq p''$, we have*

$$\sum_{p' \in \mathcal{S}_u} \text{softmax}(\boldsymbol{X}^{t\top}\boldsymbol{W}^{(S)}\boldsymbol{x}_p^t/d)_{p'} \geq 1 - \sqrt{\epsilon}, \quad (20)$$

$$\begin{aligned} &\text{softmax}(\boldsymbol{X}^{t\top}\boldsymbol{W}^{(S)}\boldsymbol{x}_p^t/d)_{p'} \\ &= (1 \pm \Theta(\epsilon))\text{softmax}(\boldsymbol{X}^{t\top}\boldsymbol{W}^{(S)}\boldsymbol{x}_p^t/d)_{p''}. \end{aligned} \quad (21)$$

Corollary 3 shows that: When one inspects each column of the softmax attention output of the trained model given a new noisy data $\boldsymbol{X}^t$ whose $p$-th token $\boldsymbol{x}_p^t$ is associated with the latent variable $Y_p = u$, the attention weights are concentrated among all tokens with the same pattern, as characterized by (20); Moreover, the attention probabilities are

distributed almost uniformly among those tokens, as shown by (21). As such, for each input token $x_p^t$, the self-attention structure approximately outputs the mean of tokens that share the same pattern, i.e., a minimum-variance unbiased estimator (MVUE) of the mean of the Gaussian component from which $x_p^t$ is sampled. We refer to this attention behavior as the *Mean denoising* mechanism. This mechanism suggests that self-attention can estimate $\sqrt{\alpha_t} M_Y$ in (13) reliably and with minimal bias as long as for every pattern appeared in $M_Y$, there is a sufficient number of tokens sampled with that pattern. Consequently, the total number of tokens per data $P$ is required to be large, as stated in (iii) in our Theorem 1, to achieve near-optimal denoising for the MTGM data.

**Implication of the mechanism on denoising data with a shifted $\tilde{\pi}$.** The conclusion of Proposition 3 shows that the trained self-attention module learns all patterns, regardless of their proportions in the data distribution. This motivates our discussion of the generative performance on test data with a shifted pattern proportion parameter $\tilde{\pi}$.

Specifically, consider $X^0 \sim \mathcal{D}(\tilde{\pi}', K, \{\mu_i\}_{i=1}^M, \rho)$, where $\tilde{\pi}'$ may not equal to $\tilde{\pi}$, i.e., the fraction of Gaussian components of training data. The DDPM loss in expectation and the score matching error are then computed following (3) and (5), respectively, but based on input distribution parameterized with $\tilde{\pi}'$. We obtain the following corollary.

**Corollary 4.** *Given the trained model in Theorem 1 with parameters $\Psi^{(S)}$ that satisfies (10) for some $\epsilon \in (0, \delta(\tilde{\pi})^{\Theta(1)})$, then for any $X^0 \sim \mathcal{D}(\tilde{\pi}', K, \{\mu_i\}_{i=1}^M, \rho)$ with the number of tokens $P \geq \Omega(\nu_{\min}^{\tilde{\pi}'}(K)^{-1}(\rho^2+1)\epsilon^{-1}\log d)$ in each data, we have $L(\Psi^{(S)}) \leq R_{oracle} + O(\epsilon)$, $\mathcal{E}(\Psi^{(S)}) \leq O(\epsilon)$.*

Corollary 4 shows that a model trained under the conditions of Theorem 1 can also achieve an $O(\epsilon)$ DDPM loss and score matching error on data with distribution-shifted pattern proportions if the number of tokens per data is large enough. This is because (18) shows that the trained self-attention mechanism yields a large lower bound on the inner product between queries and keys that share the same pattern. This bound holds uniformly for all patterns, and therefore the model does not fail to learn a pattern simply because it appears with low frequency. As a result, even when $\tilde{\pi}$ shifts to $\tilde{\pi}'$, as long as each data point contains sufficiently many tokens that scales with $\nu_{\min}^{\tilde{\pi}'}(K)^{-1}$ such that self-attention can denoise by averaging tokens with the same pattern, the model can successfully denoise under the shifted data distribution and achieve effective score learning. Corollary 4 is conceptually related to the compositional generalization discussed in (Favero et al., 2025). The difference is that compositional generalization can be viewed as a shift in the sampling distribution of $Z$ in Definition 1 from training to generation, rather than a shift from $\tilde{\pi}$ to $\tilde{\pi}'$.

### 4.3. Proof Idea, Technical Novelty, and Limitations

**Proof idea of Theorem 1.** In Lemma 3, we prove that the gradient updates of $W^{(s)}$ along directions corresponding to query–key pairs with the same pattern admit a lower bound, while the norm of the gradient updates along directions corresponding to query–key pairs with different patterns is very small. By accumulating gradients updates over steps, we obtain the mean denoising mechanism described in Proposition 3 and Corollary 3, at which point the optimization of $W^{(s)}$ converges (Lemma 4). Note that patterns with smaller fractions are learned more slowly than those with larger fractions. To ensure that the mean denoising mechanism holds for all patterns, the required number of training iterations we derive depends on $\nu_{\min}^{\tilde{\pi}}$, the minimum probability of selecting a pattern in data. The training of $v_t^{(s)}$ is then reduced to a linear problem, and Lemma 5 provides a proof of convergence. Since the learned parameters are close to the oracle MMSE estimator, we can show that the DDPM loss after training is close to $R_{\text{oracle}}$. Combined with Proposition 2, this yields the global convergence result stated in (10).

**Proof idea of Theorem 2.** The score matching error (5) can be decomposed into the error of fitting the conditional score function $\nabla_{X^t} \log p_t(X^t|X^0)$ and the discrepancy between the score function and the conditional score function. The former can be upper bounded by $O(\epsilon)$ since the trained Transformer can approximate the conditional score function by Proposition 1. The latter can be shown, based on Proposition 2, to be $O(\epsilon)$. Therefore, we can construct a score network in (11) via the trained model $\Psi$ such that the score matching error is as small as $O(\epsilon)$.

**Technical novelty.** Our proof technique is inspired by the feature learning technique in studying Transformers. For the first time, we extend their analysis of label-prediction tasks, such as binary classification (Li et al., 2023a; 2024a; Jiang et al., 2024) and linear regression (Zhang et al., 2024; Huang et al., 2023), to denoising tasks. Our work also extends the mechanism of attention concentration (Huang et al., 2023; Li et al., 2023a; 2024a) to diffusion models in denoising (Corollary 3). Our technique preserves the nonlinearity of diffusion models rather than linearizing the model by an impractical extremely-wide network assumption in (Han et al., 2024; Wang et al., 2024). This enables a convergence analysis beyond the NTK regime.

**Limitations and possible future extensions** Although we consider more complex models and data distributions than those in prior works as discussed in Section 1, our analysis is still under a restricted setting: the network model is a one-layer, single-head Transformer, the data follows the MTGM distribution with orthogonal patterns, and the GD algorithm is run on population DDPM loss. We emphasize, however, that the focus of this paper is to provide an initial theoretical understanding of the convergence and denois-

ing mechanisms of Transformer-based diffusion models, which could serve as building blocks for rigorous analysis in more realistic settings. One potential future extension is to study the convergence behavior of the multi-head attention Transformer when the data model possesses multiple types of internal relationships among tokens. Another is to extend the convergence results to empirical DDPM losses and analyze the generalization of DDPM by studying the gap between the empirical and population loss.

## 5. Numerical Experiments

In this section, we conduct synthetic experiments in Section 5.1 and real-data experiments in Section 5.2 to justify our findings, respectively. Due to space limitations, some additional experiments are moved to Appendix.

### 5.1. Synthetic Experiments

**Setup.** Synthetic data are generated as described in Definition 1. Let $d = 64$, $M = 8$, $P = 256$, $\rho = 0.3$. If not specified, $K = 4$. We consider generating uniform or non-uniform scenarios by varying $\min_{u \in [M]} \tilde{\pi}_u$. A smaller $\min_{u \in [M]} \tilde{\pi}_u$ indicates a more non-uniform distribution among patterns. The learning model is a one-layer single-head Transformer as formulated in (1). The total number of time steps is $T = 50$. We adopt a linear schedule, i.e., $\bar{\alpha}_t = \prod_{i=1}^{t} \alpha_i$, where $\alpha_t = \alpha_1 - (\alpha_1 - \alpha_T) \cdot (t-1)/(T-1)$. We set $\alpha_1 = 0.98$, $\alpha_T = 0.95$. Since the Bayes denoising risk may vary under different hyperparameter settings, we compute the excess risk, denoted by $L(\Psi^{(s)}) - R_{\text{oracle}}$ with $\boldsymbol{X}^0 \sim \mathcal{D}(\tilde{\boldsymbol{\pi}}', K, \{\boldsymbol{\mu}_i\}_{i=1}^{M}, \rho)$, to characterize the distance between the model and global convergent point. Note that during evaluation, we directly use $\tilde{\boldsymbol{\pi}}'$, which is a randomly generated uniform pattern distribution independent of $\tilde{\boldsymbol{\pi}}$. This is to measure the performance of the trained model under distribution shifts in the pattern proportion, thereby more accurately characterizing the quality of how the model learns the patterns.

**Convergence and score learning.** In Figure 2, we showcase the results of convergence and score matching performance of the trained model. 2 A reveals that the Bayes denoising risk is close to the oracle denoising risk, which verifies (17) of Proposition 2. In addition, the DDPM loss gradually decreases during training to a value close to these two risks, while the score matching error gradually decreases to near zero. These observations are consistent with (10) of Theorem 1 and (5) in Theorem 2, respectively. 2 B substantiates the discussion in Corollary 1 regarding the effect of $K$ on the number of training iterations required for convergence. 2 C shows that a more uniform distribution over pattern types can reduce the number of iterations needed for training. 2 D explains that sampling only at smaller time steps can accelerate convergence, because under a linear schedule, $\bar{\alpha}_t$ is a decreasing function of the time step $t$. Therefore, sampling only from the first 40% of time steps is equiva-

lent to increasing the time-averaged SNR, which speeds up convergence according to Theorem 1.

**Mean denoising mechanism.** We next verify the findings in Section 4.2 regarding the mean denoising mechanism. We demonstrate that the sum of attention weights on keys with the same pattern as the query increases to close to 1 during the training in Figure 2 (E), which justifies (20) in Corollary 3 for mean denoising.

### 5.2. Real-Data Experiments

**Setup.** We conduct experiments on the real dataset MNIST (LeCun et al., 2002). We select digits "0", "1", "2", and "3" for training and generation, where digit "2" is treated as a minority class with only 30% of its training set used, while the other digits use the full training set. The training model is a 6-layer, 4-head DiT (Peebles & Xie, 2023). We also train a CNN on MNIST to label the generated digits. During the training, we compute the FID score for each of the four generated digits to measure generation quality at different training stages.

**Training dynamics for generation.** As a minority pattern, digit "2" exhibits a slower decrease in FID score than the other digits (Figure 3). This indicates that different patterns are learned at different speeds, i.e., the high-frequency pattern is learned faster than the low-frequency pattern, which is consistent with the training dynamics from the proof of Theorem 1 in Section 4.3.

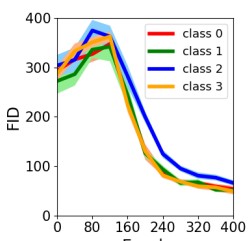

*Figure 3.* FID score of the four generated digits of MNIST. The FID of the minority, "2" decreases more slowly than the others.

## 6. Conclusion

This paper provides a global convergence and score-learning analysis for a one-layer, single-head nonlinear Transformer in diffusion model training. This work also offers a theoretical understanding of how Transformer models learn the oracle MMSE estimator of the training problem through the mean denoising mechanism. Future directions include analyzing and designing different sampling strategies, optimization algorithms, and diffusion model frameworks.

## Acknowledgements

The authors acknowledge the support of the NSF under grant 2031985, the Simons Foundation under grant 814201, the ONR MURI Program under grant 503405-78051, and the University of Pennsylvania Startup Funds. Morever, Hancheng Min thanks the support of Science and Technology Commission of Shanghai Municipality under Shanghai Pujiang Programme (grant number 25PJA070), and the

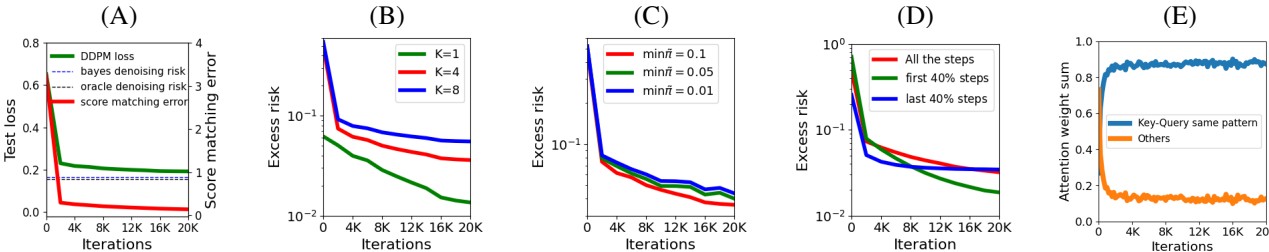

*Figure 2.* The convergence performance and the attention behavior of the trained model. (A) The green and red curves are the test loss and score matching error during diffusion model training, respectively. Blue dashed line: Bayes denoising risk. Black dashed line: oracle denoising risk. (B) Excess risk with varying $K$, the number of Gaussian components per data. (C) Excess risk with varying $\min_{u \in [M]} \tilde{\pi}_u$, i.e., the minimal fraction among all the Gaussian components. A larger $\min_{u \in [M]} \tilde{\pi}_u$ indicates a more uniform distribution of all the patterns. (D) Excess risk with different sampling strategies. Red curve: uniform sampling $t \sim \mathrm{Unif}([T])$. Green curve: sampling from the first 40% time steps, i.e., $t \sim \mathrm{Unif}([1, 0.4 \cdot T])$. Blue curve: sampling from the last 40% time steps, i.e., $t \sim \mathrm{Unif}(0.6 \cdot T, T)$. (E) The attention weight summation on keys with the same pattern as the query and on other keys.

Shanghai Jiao Tong University Startup Funds.

## Impact Statement

This paper aims to explore the convergence analysis and the denoising mechanism of diffusion model parameterized by Transformers. The primary focus is on the mathematical analysis of convergence and training dynamics. To the best of our knowledge, no potential societal consequences are associated with our work.

## The Use of Large Language Models

We used large-language models (ChatGPT) to help polish the writing of this paper.

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

## A. Additional Experiments

We first plot the query-key inner product of synthetic data. Figure 4 shows that, first, the query–key inner products corresponding to the same pattern are large and grow along the training, while those corresponding to different patterns remain small. Second, different patterns are learned at different speeds: the high-frequency pattern associated with $\max_{u \in [M]} \tilde{\pi}_u$ is learned faster and with smaller magnitude fluctuations than the low-frequency pattern associated with $\min_{u \in [M]} \tilde{\pi}_u$. We use a red dashed line and a red solid line to show the different required number of iterations of the inner products corresponding to different patterns. This result is aligned with Proposition 3.

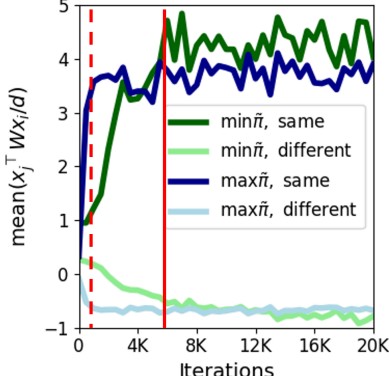

*Figure 4.* Query-key inner products with the same or different patterns, where query patterns are the minimal or maximal of $\tilde{\boldsymbol{\pi}}$. $\min_{u \in [M]} \tilde{\pi}_u = 0.01$.

In Figure 5, we show the visualization of the four generated MNIST digits using DiT (Peebles & Xie, 2023). The result shows that the final generation quality of the minority digit "2", is relatively worse.

## B. Preliminaries

We first present Table 2 for a summary of notations used in the proof.

**Lemma 1.** *(Multiplicative Chernoff bounds, Theorem D.4 of (Mohri et al., 2018)) Let $X_1, \cdots, \boldsymbol{X}_m$ be independent random variables drawn according to some distribution $\mathcal{D}$ with mean $p$ and support included in $[0, 1]$. Then, for any $\gamma \in [0, \frac{1}{p} - 1]$,*

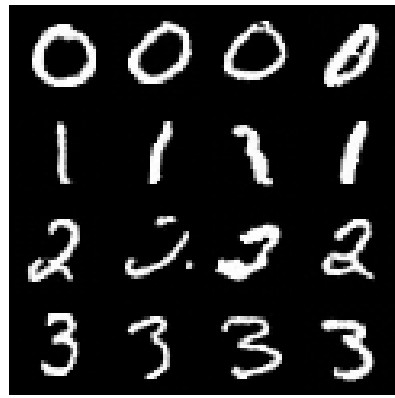

*Figure 5.* Visualization of the generated digits.

*Table 2.* Summary of Notations

| Notations | Annotation |
|---|---|
| $d, P$ | $d$ is the dimension of each token of data. $P$ is the total number of tokens in each data. |
| $M, K$ | $M$ is the total number of mean patterns. $K$ is the number of distinct patterns in each data. |
| $\tilde{\boldsymbol{\pi}}, \{\boldsymbol{\mu}_i\}_{i=1}^M, \rho$ | $\tilde{\boldsymbol{\pi}}$ is the fraction vector of all the Gaussian components of the MTGM distribution. $\{\boldsymbol{\mu}_i\}_{i=1}^M$ is the set of all the mean patterns. $\rho^2$ is the variance of the Gaussian components. |
| $\boldsymbol{X}, \boldsymbol{Y}, \boldsymbol{Z}$ | $\boldsymbol{X}$ denotes the data. $\boldsymbol{Y}$ ad $\boldsymbol{Z}$ are the latent variable to define the distribution of $\boldsymbol{X}$. |
| $\Psi, \boldsymbol{W}, T, \{v_t\}_{t=1}^T$ | $\Psi$ is the set of parameters in the learning model. In our work, $\Psi$ contains $\boldsymbol{W}$ and $\{v_t\}_{t=1}^T$, where $\boldsymbol{W}$ is the self-attention parameter, and $\{v_t\}_{t=1}^T$ is the coefficient parameter. $T$ is the total number of diffusion time steps. |
| $\boldsymbol{X}^0, \{\bar{\alpha}_t\}_{t=1}^T, \boldsymbol{E}, \boldsymbol{X}^t$ | $\boldsymbol{X}^t$ is the noisy input at time step $t$. $\boldsymbol{X}^0$ is the clean input before adding the noise. $\{\bar{\alpha}_t\}_{t=1}^T$ is the noise schedule coefficients. $\boldsymbol{E}$ is the additive Gaussian noise. |
| $s(\boldsymbol{X}^t, t), s_\theta(\boldsymbol{X}^t, t)$ | $s(\boldsymbol{X}^t, t)$ is the score function. $s_\theta(\boldsymbol{X}^t, t)$ is the neural network parameterized by $\theta$ to learn the score function. |
| $\nu_{\min}^{\tilde{\boldsymbol{\pi}}}, \delta(\boldsymbol{\pi}), \text{SNR}$ | $\nu_{\min}^{\tilde{\boldsymbol{\pi}}}$ is the minimum average probability of selecting a pattern in the data. $\delta(\boldsymbol{\pi})$ is the degree of imbalance between the prior probabilities of the least and the most probable patterns. SNR is the time-averaged signal-noise-ratio over the noise schedule. |
| $R_{\text{Bayes}}, \boldsymbol{M_Y}, R_{\text{oracle}}$ | $R_{\text{Bayes}}$ is the optimal risk of minimizing the DDPM loss, where the denoising model is chosen as the MMSE estimator of the noise. $\boldsymbol{M_Y}$ is the matrix of mean patterns given $\boldsymbol{Y}$, the latent variable of $\boldsymbol{X}$. $R_{\text{oracle}}$ is the risk if the denoising model is chosen as the oracle MMSE estimator with $\boldsymbol{M_Y}$ as known. |
| $\mathcal{O}(), \Omega(), \Theta()$ | We follow the convention that $f(x) = O(g(x))$ (or $\Omega(g(x))$, $\Theta(g(x))$)) means that $f(x)$ increases at most, at least, or in the order of $g(x)$, respectively. Specifically, if $f(x) = O(g(x))$, then there exists $C > 0$ and $a > 0$, such that $f(x) \leq C \cdot g(x)$ when $x > a$. If $f(x) = \Omega(g(x))$, then there exists $c > 0$ and $a > 0$, such that $f(x) \geq c \cdot g(x)$ when $x > a$. If $f(x) = \Theta(g(x))$, then there exists $C > c > 0$ and $a > 0$, such that $c \cdot g(x) \leq f(x) \leq C \cdot g(x)$ when $x > a$. |
| $\gtrsim, \lesssim$ | $f(x) \gtrsim g(x)$ (or $f(x) \lesssim g(x)$ ) means that $f(x) \geq \Omega(g(x))$ (or $f(x) \lesssim \mathcal{O}(g(x))$). |
| $\text{poly}()$ | If $f(x) = \text{poly}(x)$, then there exists $k > 0$ and a set of constants $\{c_i\}_{i=0}^k$, such that $f(x) = \sum_{i=0}^k c_i x^i$, which means $f(x)$ is a polynomial function of $x$ with a finite maximal power. |

the following inequality holds for $\hat{p} = \frac{1}{m} \sum_{i=1}^m X_i$:

$$\Pr(\hat{p} \geq (1 + \gamma)p) \leq e^{-\frac{mp\gamma^2}{3}}, \tag{22}$$

$$\Pr(\hat{p} \leq (1 - \gamma)p) \leq e^{-\frac{mp\gamma^2}{2}}. \tag{23}$$

**Definition 4.** *(Vershynin, 2012) We say $X$ is a sub-Gaussian random variable with sub-Gaussian norm $K > 0$, if $(\mathbb{E}|X|^p)^{\frac{1}{p}} \leq K\sqrt{p}$ for all $p \geq 1$. In addition, the sub-Gaussian norm of $X$, denoted $\|X\|_{\psi_2}$, is defined as $\|X\|_{\psi_2} = \sup_{p \geq 1} p^{-\frac{1}{2}} (\mathbb{E}|X|^p)^{\frac{1}{p}}$.*

**Lemma 2.** *((Vershynin, 2012) Proposition 5.1, Hoeffding's inequality) Let $X_1, X_2, \cdots, X_N$ be independent centered sub-gaussian random variables, and let $K = \max_i \|\boldsymbol{X}_i\|_{\psi_2}$. Then for every $\boldsymbol{a} = (a_1, \cdots, a_N) \in \mathbb{R}^N$ and every $t \geq 0$, we have*

$$\Pr\left(\left|\sum_{i=1}^N a_i X_i\right| \geq t\right) \leq e \cdot \exp\left(-\frac{ct^2}{K^2 \|\boldsymbol{a}\|^2}\right), \tag{24}$$

*where $c > 0$ is an absolute constant.*

**Definition 5.** *For $i, p \in [P]$, $t \in [T]$, $u \in [M]$, and $\boldsymbol{X}$ that follows Definition 1, we denote $\zeta_{i,p,t}^u(s) = \text{softmax}_p(\boldsymbol{x}_i^{t\top} \boldsymbol{W}^{(s)} \boldsymbol{x}_p^t / d) \mathbb{1}[Y_p = u]$.*

**Lemma 3.** *Given conditions (i)-(v) in Theorem 1, we have that for any $\tilde{\boldsymbol{X}}$ that follows the distribution in Definition 1, where*

$\tilde{\boldsymbol{x}}_j$ and $\tilde{\boldsymbol{x}}_{j'}$ have $\boldsymbol{\mu}_u$ as the mean, and $\tilde{\boldsymbol{x}}_k$ has $\boldsymbol{\mu}_{u'}$ as the mean ($u \neq u'$), then we have for any $s > 0$,

$$
\begin{aligned}
&(-\tilde{\boldsymbol{x}}_j^{t\top})\frac{1}{T}\sum_{t=1}^{T}\mathbb{E}_{\boldsymbol{E}^t,\boldsymbol{X}^0}\Big[\frac{\partial L(\Psi^{(s_0)})}{\partial \boldsymbol{W}}\Big]\tilde{\boldsymbol{x}}_{j'}^t\\
&\gtrsim \frac{1}{T}\sum_{t=1}^{T}\mathbb{E}_{\boldsymbol{E}^t,\boldsymbol{X}^0}[(v_t^{(s_0)})^2\bar{\alpha}_t^3 d(1-\sum_{l=1}^{P}\zeta_{l,p,t}^u(s_0))^2\sum_{i=1}^{P}\zeta_{i,p,t}^u(s_0)]\cdot\nu_u^{\tilde{\boldsymbol{\pi}}}(K),
\end{aligned}
\tag{25}
$$

$$
(-\tilde{\boldsymbol{x}}_k^{\top})\frac{1}{T}\sum_{t=1}^{T}\mathbb{E}_{\boldsymbol{E}^t,\boldsymbol{X}^0}\Big[\frac{\partial L(\Psi^{(s_0)})}{\partial \boldsymbol{W}}\Big]\tilde{\boldsymbol{x}}_{j'}^t \lesssim \frac{\log d}{\sqrt{d}}\cdot(-\tilde{\boldsymbol{x}}_j^{\top})\frac{1}{T}\sum_{t=1}^{T}\mathbb{E}_{\boldsymbol{E}^t,\boldsymbol{X}^0}\Big[\frac{\partial L(\Psi^{(s_0)})}{\partial \boldsymbol{W}}\Big]\tilde{\boldsymbol{x}}_{j'}^t.
\tag{26}
$$

**Lemma 4.** *For any $\epsilon \in (0, \delta^{\Theta(1)})$, when the number of iterations satisfies*

$$
I_1 \geq \Omega((\epsilon^{-1} + \nu_{\min}^{\tilde{\boldsymbol{\pi}}}(K)^{-3})\eta^{-1}\nu_{\min}^{\tilde{\boldsymbol{\pi}}}(K)^{-1}SNR^{-3}),
\tag{27}
$$

$P \geq \Omega(\nu_{\min}^{\tilde{\boldsymbol{\pi}}}(K)^{-1}(\rho^2+1)\epsilon^{-1}\log d)$, $d \geq \Omega(\epsilon^{-1}\log(\epsilon^{-1}\nu_{\min}^{\tilde{\boldsymbol{\pi}}}(K)^{-1}))$, *and* $T \geq \Omega(\log d)$, *with the step size* $\eta_1 \leq O(1)$, *then w.h.p., the learned model returns* $\boldsymbol{W}^{(I_1)}$ *such that*

$$
\Big\|\frac{1}{T}\sum_{t=1}^{T}\nabla_{\boldsymbol{W}}\mathbb{E}_{\boldsymbol{X}^0,\boldsymbol{E}^t}\|\boldsymbol{f}(\Psi^{(I_1)};\sqrt{\bar{\alpha}_t}\boldsymbol{X}^0+\sqrt{1-\bar{\alpha}_t}\boldsymbol{E}^t,t)-\boldsymbol{E}^t\|_F^2/(dP)\Big\| \leq \epsilon^2\cdot\eta\cdot K^{-1}\delta(\tilde{\boldsymbol{\pi}})^{-1}\cdot\log(I_1(\rho^2+1)\epsilon^{-1}).
\tag{28}
$$

**Lemma 5.** *For any $\epsilon \in (0, \delta^{\Theta(1)})$, $s > \Omega(I_1)$, with the step size $\eta \leq O((\max\{\rho,1\}+\epsilon)^{-1})$, then $v_t$ converges linearly to $v_t^*$ with*

$$
|v_t^{(s)} - v_t^*| \leq (1 - 2\eta(\rho\sqrt{\bar{\alpha}_t} + \sqrt{1-\bar{\alpha}_t}+\epsilon))^s|v_t^{(I_1)} - v_t^*|,
\tag{29}
$$

*where*

$$
|v_t^* - \sqrt{1-\bar{\alpha}_t}/(\bar{\alpha}_t\rho^2 + 1 - \bar{\alpha}_t)| \leq \epsilon/(\bar{\alpha}_t\rho^2 + 1 - \bar{\alpha}_t).
\tag{30}
$$

*When (i) the number of iterations satisfies*

$$
I_2 \geq \Omega(\log \epsilon^{-1}I_1(1+\rho^2)),
\tag{31}
$$

*and (ii) $P \geq \Omega(\nu_{\min}^{\tilde{\boldsymbol{\pi}}}(K)^{-1}(\rho^2+1)\epsilon^{-1}\log d)$, then w.h.p., we have for $s \geq \Omega(I_1 + I_2)$*

$$
|v_t^{(s)} - v_t^*| \leq \epsilon.
\tag{32}
$$

## C. Proof of Main Theorems

### C.1. Proof of Theorem 1

*Proof.* Following the analytical framework used in (Li et al., 2023a; Luo et al., 2024; Li et al., 2024c; Zhang et al., 2025b; Li et al., 2025c; Sun et al., 2025; Li et al., 2025a; Zhang et al., 2025a; Li et al., 2025b), we provide a convergence analysis of the DDPM training. This part mainly introduces the proof steps that combine the lemmas of gradient updates of different model parameters and training stages to derive the final convergence conclusion. The overall proof idea is summarized in Section 4.3. Note that the oracle denoising risk with the mean matrix of $\boldsymbol{X}^t$ known is

$$
R_{oracle} = \frac{\rho^2\bar{\alpha}_t}{\rho^2\bar{\alpha}_t + 1 - \bar{\alpha}_t}.
\tag{33}
$$

Therefore, by the Mean Value Theorem, for $\tilde{v}$ between $v_t^{(I_2)}$ and $\frac{\sqrt{1-\bar{\alpha}_t}}{1-\bar{\alpha}_t+\rho^2\bar{\alpha}_t}$,

$$
\begin{aligned}
&\mathbb{E}_{\boldsymbol{E}^t,\boldsymbol{X}^0}[\|v_t^{(I_2)}(\boldsymbol{x}_p^t - \sum_{i=1}^{P}\boldsymbol{x}_i^t\mathrm{softmax}_p(\frac{\boldsymbol{x}_i^{t\top}\boldsymbol{W}^{(I_1)}\boldsymbol{x}_p^t}{d})) - \boldsymbol{\epsilon}_p\|^2/d] - R_{oracle}\\
&\leq \epsilon\cdot 2(\tilde{v}\cdot\mathbb{E}_{\boldsymbol{E}^t,\boldsymbol{X}^0}[\bar{\alpha}_t\rho^2 + 1 - \bar{\alpha}_t + \|\sqrt{\bar{\alpha}_t}\boldsymbol{\mu}_u - \sum_{i=1}^{P}\boldsymbol{x}_i^t\mathrm{softmax}_p(\frac{\boldsymbol{x}_i^{t\top}\boldsymbol{W}^{(I_1)}\boldsymbol{x}_p^t}{d})\|^2/d\\
&\quad + \frac{2\beta_1\rho\sqrt{\bar{\alpha}_t}}{d} + \frac{2\beta_2\sqrt{1-\bar{\alpha}_t}}{d}] - (\sqrt{1-\bar{\alpha}_t} + \frac{\beta_2}{d}))\\
&\lesssim O(\epsilon),
\end{aligned}
\tag{34}
$$

and we can obtain that for any $t$,

$$\mathbb{E}_{\boldsymbol{X}^0, \boldsymbol{E}^t} \| \boldsymbol{f}(\Psi^{(S)}; \sqrt{\bar{\alpha}_t} \boldsymbol{X}^0 + \sqrt{1 - \bar{\alpha}_t} \boldsymbol{E}^t, t) - \boldsymbol{E}^t \|_F^2 / (dP) \| \leq R_{oracle} + O(\epsilon). \tag{35}$$

Note that

$$I_1 + I_2 \lesssim I_1 + \log(1 + \rho^2)\epsilon^{-1}. \tag{36}$$

Therefore, by combining Lemma 4 and 5, we can obtain the following result. For any $\epsilon \in (0, \delta^{\Theta(1)})$, when the number of iterations satisfies

$$s \geq \Omega((\epsilon^{-1} + \nu_{\min}^{\tilde{\boldsymbol{\pi}}}(K)^{-3})\eta^{-1}\nu_{\min}^{\tilde{\boldsymbol{\pi}}}(K)^{-1}\mathrm{SNR}^{-3} + \log(1 + \rho^2)\epsilon^{-1}), \tag{37}$$

and the number of tokens

$$P \geq \Omega(\nu_{\min}^{\tilde{\boldsymbol{\pi}}}(K)^{-1}(\rho^2 + 1)\epsilon^{-1} \log d), \tag{38}$$

with the step size $\eta \leq (\max\{\rho, 1\} + \epsilon)^{-1}$, then with a high probability, the learned model $\Psi^{(S)}$ satisfies

$$\frac{1}{T} \sum_{t=1}^{T} \mathbb{E}_{\boldsymbol{X}^0 \sim \mathcal{D}(\tilde{\boldsymbol{\pi}}, \{\boldsymbol{\mu}_i\}_{i=1}^M, \rho), \boldsymbol{E}^t} \left[ \| \boldsymbol{f}(\Psi^{(S)}; \sqrt{\bar{\alpha}_t} \boldsymbol{X}^0 + \sqrt{1 - \bar{\alpha}_t} \boldsymbol{E}^t, t) - \boldsymbol{E}^t \|_F^2 / dP \right] \\ \leq R_{\text{oracle}} + O(\epsilon). \tag{39}$$

Combining Corollary 2, we have

$$\frac{1}{T} \sum_{t=1}^{T} \mathbb{E}_{\boldsymbol{X}^0 \sim \mathcal{D}(\tilde{\boldsymbol{\pi}}, \{\boldsymbol{\mu}_i\}_{i=1}^M, \rho), \boldsymbol{E}^t} \left[ \| \boldsymbol{f}(\Psi^{(S)}; \sqrt{\bar{\alpha}_t} \boldsymbol{X}^0 + \sqrt{1 - \bar{\alpha}_t} \boldsymbol{E}^t, t) - \boldsymbol{E}^t \|_F^2 / dP \right] \\ \leq R_{\text{Bayes}} + O(\epsilon). \tag{40}$$

$\square$

## C.2. Proof of Corollary 1

*Proof.* Given a fixed $\tilde{\boldsymbol{\pi}}$ and $M$, when $K = 1$, we have for any $u \in [M]$,

$$\mathbb{E}[\pi_u] = \mathbb{E}_{\boldsymbol{Z} \in \{0,1\}^M, \|\boldsymbol{Z}\|_0 = K} \left[ \frac{\tilde{\pi}_u}{\boldsymbol{Z}^\top \tilde{\boldsymbol{\pi}}} \right], \tag{41}$$

$$\min_{u \in [M]} \{\mathbb{E}[\pi_u]\} = \frac{1}{K}. \tag{42}$$

Since that

$$\sum_{u \in [M]} \mathbb{E}[\pi_u] = 1, \tag{43}$$

$K = 1$ is the case where $\min_{u \in [M]}\{\mathbb{E}[\pi_u]\} \cdot \frac{K}{M}$ reaches its maximal. In this case, we have

$$I_1 = (\epsilon^{-1} + M^3)\eta^{-1}M \cdot \mathrm{SNR}^{-3} \tag{44}$$

When $K = M$, we have $\mathbb{E}[\pi_u] = \tilde{\pi}_u$ for any $u \in [M]$. Then,

$$\min_{u \in [M]} \{\mathbb{E}[\pi_u]\} \cdot \frac{K}{M} = \min_{u \in [M]} \{\tilde{\pi}_u\}. \tag{45}$$

By Jensen's inequality,

$$\begin{aligned} \mathbb{E}[\pi_u] &= \mathbb{E}_{\boldsymbol{Z} \in \{0,1\}^M, \|\boldsymbol{Z}\|_0 = K} \left[ \frac{\tilde{\pi}_u}{\boldsymbol{Z}^\top \tilde{\boldsymbol{\pi}}} \right] \\ &= \mathbb{E}_{\boldsymbol{Z} \in \{0,1\}^M, \|\boldsymbol{Z}\|_0 = K} \left[ \frac{\tilde{\pi}_u}{(\boldsymbol{Z}^\top \tilde{\boldsymbol{\pi}} - \tilde{\pi}_u) + \tilde{\pi}_u} \right] \\ &\geq \frac{\tilde{\pi}_u}{\mathbb{E}_{\boldsymbol{Z} \in \{0,1\}^M, \|\boldsymbol{Z}\|_0 = K}[\boldsymbol{Z}^\top \tilde{\boldsymbol{\pi}} - \tilde{\pi}_u] + \tilde{\pi}_u} \\ &= \frac{\tilde{\pi}_u}{\frac{(K-1)(1-\tilde{\pi}_u)}{M-1} + \tilde{\pi}_u} \end{aligned} \tag{46}$$

Then, for any $u \in [M]$, we have

$$\mathbb{E}_{K=M}[\pi_u] = \tilde{\pi}_{u^*} = \frac{\tilde{\pi}_u}{\frac{(M-1)(1-\tilde{\pi}_u)}{M-1} + \tilde{\pi}_u} \leq \frac{\tilde{\pi}_u}{\frac{(K-1)(1-\tilde{\pi}_u)}{M-1} + \tilde{\pi}_u} \leq \mathbb{E}_{K<M}[\pi_u], \tag{47}$$

where the first inequality comes from the fact that $g(K) = \frac{\tilde{\pi}_u}{\frac{(K-1)(1-\tilde{\pi}_u)}{M-1} + \tilde{\pi}_u}$ is a decreasing function of $K$. Therefore, $K = M$ is the case where $\min_{u \in [M]}\{\mathbb{E}[\pi_u]\} \cdot \frac{K}{M}$ reaches its minimal. In this case, we have

$$I_1 = (\epsilon^{-1} + \min_{u \in [M]}\{\tilde{\pi}_u\}^{-3})\eta^{-1} \min_{u \in [M]}\{\tilde{\pi}_u\}^{-1}\mathrm{SNR}^{-3} \tag{48}$$

$\square$

## C.3. Proof of Theorem 2

*Proof.* By Fisher identity, we have

$$\begin{aligned}
\nabla_{\boldsymbol{X}^t} \log q(\boldsymbol{X}^t) &= \mathbb{E}[\nabla_{\boldsymbol{X}^t} \log q(\boldsymbol{X}^t|\boldsymbol{X}^0)|\boldsymbol{X}^t] \\
&= \mathbb{E}\Big[ -\frac{1}{1-\bar{\alpha}_t}(\boldsymbol{X}^t - \sqrt{\bar{\alpha}_t}\boldsymbol{X}^0)\Big|\boldsymbol{X}^t\Big] \\
&= -\frac{1}{\sqrt{1-\bar{\alpha}_t}}\mathbb{E}[\boldsymbol{E}^t|\boldsymbol{X}^t]
\end{aligned} \tag{49}$$

where the second step is by $\boldsymbol{X}^T = \sqrt{\bar{\alpha}_t}\boldsymbol{X}^0 + \sqrt{1-\bar{\alpha}_t}\boldsymbol{E}^t$. Let

$$s_{\boldsymbol{\Psi}^{(S)}}(\boldsymbol{X}^t, \boldsymbol{E}^t, t) = -\frac{1}{\sqrt{1-\bar{\alpha}_t}}\boldsymbol{f}(\boldsymbol{W}^{(S)}, \boldsymbol{v}^{(S)}; \sqrt{\bar{\alpha}_t}\boldsymbol{X}^0 + \sqrt{1-\bar{\alpha}_t}\boldsymbol{E}^t, t) \tag{50}$$

for the required number of iterations in (9). Then, we can obtain that with $s \geq \Omega(I_1 + I_2)$,

$$\begin{aligned}
&\mathbb{E}_{\boldsymbol{X}^0, \boldsymbol{E}^t}\big[\|\nabla_{\boldsymbol{X}^t} \log q(\boldsymbol{X}^t) - s_{\boldsymbol{\Psi}^{(S)}}(\boldsymbol{X}^t, \boldsymbol{E}^t, t)\|^2\big] \\
&= \mathbb{E}_{\boldsymbol{X}^0, \boldsymbol{E}^t}\Big[\Big\|\frac{1}{\sqrt{1-\bar{\alpha}_t}}\mathbb{E}[\boldsymbol{E}^t|\boldsymbol{X}^t] - \frac{1}{\sqrt{1-\bar{\alpha}_t}}\mathbb{E}[\boldsymbol{E}^t|\boldsymbol{X}^t, \boldsymbol{Y}] \\
&\quad + \frac{1}{\sqrt{1-\bar{\alpha}_t}}\mathbb{E}[\boldsymbol{E}^t|\boldsymbol{X}^t, \boldsymbol{Y}] + s_{\boldsymbol{\Psi}^{(S)}}(\boldsymbol{X}^t, \boldsymbol{E}^t, t)\Big\|^2\Big] \\
&\leq 2\mathbb{E}_{\boldsymbol{X}^0, \boldsymbol{E}^t}\Big[\Big\|\frac{1}{\sqrt{1-\bar{\alpha}_t}}\mathbb{E}[\boldsymbol{E}^t|\boldsymbol{X}^t] - \frac{1}{\sqrt{1-\bar{\alpha}_t}}\mathbb{E}[\boldsymbol{E}^t|\boldsymbol{X}^t, \boldsymbol{Y}]\Big\|^2\Big] \\
&\quad + 2\mathbb{E}_{\boldsymbol{X}^0, \boldsymbol{E}^t}\Big[\Big\|\frac{1}{\sqrt{1-\bar{\alpha}_t}}\mathbb{E}[\boldsymbol{E}^t|\boldsymbol{X}^t, \boldsymbol{Y}] + s_{\boldsymbol{\Psi}^{(S)}}(\boldsymbol{X}^t, \boldsymbol{E}^t, t)\Big\|^2\Big] \\
&:= 2C_1 + 2C_2.
\end{aligned} \tag{51}$$

Note that

$$\mathbb{E}[\boldsymbol{E}^t|\boldsymbol{X}^t, \boldsymbol{Y}] = \frac{\sqrt{1-\bar{\alpha}_t}}{1-\bar{\alpha}_t + \rho^2\bar{\alpha}_t}(\boldsymbol{X}^t - \sqrt{\bar{\alpha}_t}\boldsymbol{M}_{\boldsymbol{Y}}). \tag{52}$$

Therefore,

$$\begin{aligned}
C_2 &= \mathbb{E}_{\boldsymbol{X}^0, \boldsymbol{E}^t}\Big[\frac{1}{1-\bar{\alpha}_t}\Big\|\boldsymbol{f}(\boldsymbol{W}, \boldsymbol{v}; \sqrt{\bar{\alpha}_t}\boldsymbol{X}^0 + \sqrt{1-\bar{\alpha}_t}\boldsymbol{E}^t, t) - \mathbb{E}[\boldsymbol{E}^t|\boldsymbol{X}^t, \boldsymbol{Y}]\Big\|^2\Big] \\
&= \mathbb{E}_{\boldsymbol{X}^0, \boldsymbol{E}^t}\Big[\frac{\sum_{p=1}^P}{1-\bar{\alpha}_t}\Big\|(v_t^{(s)} - \frac{\sqrt{1-\bar{\alpha}_t}}{1-\bar{\alpha}_t + \rho^2\bar{\alpha}_t})(\boldsymbol{x}_p^t - \sum_{i=1}^P \boldsymbol{x}_i^t \mathrm{softmax}_p(\frac{\boldsymbol{x}_i^{t\top}\boldsymbol{W}^{(s)}\boldsymbol{x}_p^t}{d})) \\
&\quad + \frac{\sqrt{1-\bar{\alpha}_t} \cdot (\sum_{i=1}^P \boldsymbol{x}_i^t \mathrm{softmax}_p(\frac{\boldsymbol{x}_i^{t\top}\boldsymbol{W}^{(s)}\boldsymbol{x}_p^t}{d}) - \sqrt{\bar{\alpha}_t}(\boldsymbol{M}_{\boldsymbol{Y}})_p)}{1-\bar{\alpha}_t + \rho^2\bar{\alpha}_t}\Big\|^2\Big] \\
&\lesssim dP \cdot \frac{1}{1-\bar{\alpha}_t}(\epsilon^2 + \frac{\epsilon/d}{(1-\bar{\alpha}_t)^2}) + dP\epsilon \cdot \frac{1}{(1-\bar{\alpha}_t + \rho^2\bar{\alpha}_t)^2}
\end{aligned} \tag{53}$$

where the last step is by (142), and

$$
\begin{aligned}
&|v_t^{(s)} - \frac{\sqrt{1-\bar{\alpha}_t}}{1-\bar{\alpha}_t + \rho^2 \bar{\alpha}_t}| \\
&\leq |v_t^{(s)} - v_t^*| + |v_t^* - \frac{\sqrt{1-\bar{\alpha}_t}}{1-\bar{\alpha}_t + \rho^2 \bar{\alpha}_t}| \\
&\leq \epsilon + \frac{\sqrt{\epsilon/d}}{\bar{\alpha}_t \rho^2 + 1 - \bar{\alpha}_t} + O(\frac{\sqrt{1-\bar{\alpha}_t}\sqrt{\epsilon/d}}{(\bar{\alpha}_t \rho^2 + 1 - \bar{\alpha}_t)^{\frac{3}{2}}}) \\
&\leq \epsilon + O(\frac{\sqrt{\epsilon/d}}{1-\bar{\alpha}_t}).
\end{aligned}
\tag{54}
$$

Consider the Hilbert space $\mathcal{H} := L^2(\Omega, \mathcal{F}, \mathbb{P})$ equipped with the inner product $\langle X, Y \rangle = \mathbb{E}[XY]$, where $(\Omega, \mathcal{F}, \mathbb{P})$ is the underlying probability space. We define

$$
H_{\mathcal{A}} := \{X \in \mathcal{H} : X \text{ is } \mathcal{A}\text{-measurable}\}
\tag{55}
$$

as the closed subspace for any sub-$\sigma$-algebra $\mathcal{A} \subseteq \mathcal{F}$. Let $\mathcal{G}_1 = \sigma(\boldsymbol{X}^t)$ and $\mathcal{G}_2 = \sigma(\boldsymbol{X}^t, \boldsymbol{Y})$. We have $\mathcal{G}_1 \subseteq \mathcal{G}_2$. We know that $\mathbb{E}[\boldsymbol{E}^t | \boldsymbol{X}^t]$ and $\mathbb{E}[\boldsymbol{E}^t | \boldsymbol{X}^t, \boldsymbol{Y}]$ are orthogonal projections from $\boldsymbol{E}^t$ onto $\mathcal{G}_1$ and $\mathcal{G}_2$, respectively. Then, by Pythagorean identity, we have

$$
\mathbb{E}[\|\boldsymbol{E}^t - \mathbb{E}[\boldsymbol{E}^t | \boldsymbol{X}^t]\|^2] = \mathbb{E}[\|\boldsymbol{E}^t - \mathbb{E}[\boldsymbol{E}^t | \boldsymbol{X}^t, \boldsymbol{Y}]\|^2] + \mathbb{E}[\|\mathbb{E}[\boldsymbol{E}^t | \boldsymbol{X}^t] - \mathbb{E}[\boldsymbol{E}^t | \boldsymbol{X}^t, \boldsymbol{Y}]\|^2].
\tag{56}
$$

Hence,

$$
\begin{aligned}
C_1 &\leq \frac{1}{1-\bar{\alpha}_t}(\mathbb{E}[\|\boldsymbol{E}^t - \mathbb{E}[\boldsymbol{E}^t | \boldsymbol{X}^t]\|^2] - \mathbb{E}[\|\boldsymbol{E}^t - \mathbb{E}[\boldsymbol{E}^t | \boldsymbol{X}^t, \boldsymbol{Y}]\|^2]) \\
&\leq \frac{1}{1-\bar{\alpha}_t} \cdot dP\epsilon.
\end{aligned}
\tag{57}
$$

Combining (57) and (53), we have

$$
\mathbb{E}_{\boldsymbol{X}^0, \boldsymbol{E}^t}\left[\|\nabla_{\boldsymbol{X}^t} \log q(\boldsymbol{X}^t) - s_{\Psi(S)}(\boldsymbol{X}^t, \boldsymbol{E}^t, t)\|^2\right] \leq dP\epsilon \cdot (\frac{1}{1-\bar{\alpha}_t}).
\tag{58}
$$

By Hoeffding's inequality (24), we have that with a probability of $1 - d^{-C}$ for a large $C > 1$,

$$
\begin{aligned}
&\frac{1}{T}\sum_{t=1}^{T} \mathbb{E}_{\boldsymbol{X}^0, \boldsymbol{E}^t}\left[\|\nabla_{\boldsymbol{X}^t} \log q(\boldsymbol{X}^t) - s_{\Psi(S)}(\boldsymbol{X}^t, \boldsymbol{E}^t, t)\|^2\right] \\
&\leq \mathbb{E}_{\boldsymbol{X}^0, \boldsymbol{E}^t, t}\left[\|\nabla_{\boldsymbol{X}^t} \log q(\boldsymbol{X}^t) - s_{\Psi(S)}(\boldsymbol{X}^t, \boldsymbol{E}^t, t)\|^2\right] + dP\epsilon \cdot \frac{1}{(1-\bar{\alpha}_1)} \cdot \sqrt{\frac{\log d}{T}} \\
&\lesssim dP\epsilon(\text{SNR} + 1),
\end{aligned}
\tag{59}
$$

where the first step is by Hoeffding's inequality (24), and the last step holds if $T \geq \Omega(\log d)$. Hence,

$$
\frac{1}{dPT}\sum_{t=1}^{T} \mathbb{E}_{\boldsymbol{X}^0, \boldsymbol{E}^t}\left[\|\nabla_{\boldsymbol{X}^t} \log q(\boldsymbol{X}^t) - s_{\Psi(S)}(\boldsymbol{X}^t, \boldsymbol{E}^t, t)\|^2\right] \lesssim \epsilon(\text{SNR} + 1),
\tag{60}
$$

$\square$

## C.4. Proof of Proposition 1

*Proof.* From (142), we have

$$
\begin{aligned}
&\frac{\|\sqrt{\bar{\alpha}_t}\boldsymbol{M}_{\boldsymbol{Y}, \boldsymbol{Z}} - \boldsymbol{X}^t \text{softmax}(\boldsymbol{X}^{t\top}\boldsymbol{W}^{(S)}\boldsymbol{X}^t/d)\|_F^2}{dP} \\
&\leq O(\frac{(\rho^2 + 1)\log d}{\sum_{p=1}^{P} \mathbb{1}[Y_p = \arg\min_{u \in [M]} \nu_u^{\bar{\pi}}(K)]}) \\
&\lesssim \frac{(\rho^2 + 1)\log d}{P\nu_{\min}^{\bar{\pi}}(K)}.
\end{aligned}
\tag{61}
$$

By (54), we have

$$|v_t^{(s)} - \frac{\sqrt{1-\bar{\alpha}_t}}{1-\bar{\alpha}_t+\rho^2\bar{\alpha}_t}| \le \epsilon + O(\frac{\sqrt{\epsilon/d}}{1-\bar{\alpha}_t}) \le O(\epsilon + \sqrt{\epsilon/d}) \le O(\epsilon), \tag{62}$$

where the last step comes from $d \gtrsim \epsilon^{-1}$. $\qquad\square$

## C.5. Proof of Corollary 2

*Proof.* Note that

$$R_{\text{oracle}} = \frac{1}{dP}\mathbb{E}[\|\boldsymbol{E}^t - \mathbb{E}[\boldsymbol{E}^t|\boldsymbol{X}^t,\boldsymbol{Y}]\|^2] = \frac{1}{dP}\mathbb{E}[\text{Var}(\mathbb{E}|\boldsymbol{X}^t,\boldsymbol{Y})], \tag{63}$$

while

$$R_{\text{Bayes}} = \frac{1}{dP}\mathbb{E}[\|\boldsymbol{E}^t - \mathbb{E}[\boldsymbol{E}^t|\boldsymbol{X}^t]\|^2] = \frac{1}{dP}\mathbb{E}[\text{Var}(\boldsymbol{E}^t|\boldsymbol{X}^t)]. \tag{64}$$

By the law of total variance, we have

$$\text{Var}(\boldsymbol{E}^t|\boldsymbol{X}^t) = \mathbb{E}[\text{Var}(\boldsymbol{E}^t|\boldsymbol{X}^t,Y,Z)|\boldsymbol{X}^t] + \text{Var}(\mathbb{E}[\boldsymbol{E}^t|\boldsymbol{X}^t,\boldsymbol{Y}]|\boldsymbol{X}^t). \tag{65}$$

Therefore,

$$R_{\text{Bayes}} - R_{\text{oracle}} = \frac{1}{dP}\mathbb{E}[\text{Var}(\mathbb{E}[\boldsymbol{E}^t|\boldsymbol{X}^t,\boldsymbol{Y}]|\boldsymbol{X}^t)] > 0. \tag{66}$$

Recall that

$$\mathbb{E}[\boldsymbol{E}^t|\boldsymbol{X}^t,\boldsymbol{Y}] = \frac{\sqrt{1-\bar{\alpha}_t}}{1-\bar{\alpha}_t+\rho^2\bar{\alpha}_t}(\boldsymbol{X}^t - \sqrt{\bar{\alpha}_t}\boldsymbol{M_Y}), \tag{67}$$

where $\boldsymbol{M_Y}$ denotes the mean matrix of $\boldsymbol{X}^t$ given $\boldsymbol{Y}$. Then,

$$\begin{aligned}\mathbb{E}[\boldsymbol{X}^0|\boldsymbol{X}^t,\boldsymbol{Y}] &= \mathbb{E}\Big[\frac{\boldsymbol{X}^t - \sqrt{1-\bar{\alpha}_t}\boldsymbol{E}^t}{\sqrt{\bar{\alpha}_t}}\Big|\boldsymbol{X}^t,\boldsymbol{Y}\Big] \\ &= \frac{\rho^2\sqrt{\bar{\alpha}_t}}{1-\bar{\alpha}_t+\rho^2\bar{\alpha}_t}\boldsymbol{X}^t + \frac{1-\bar{\alpha}_t}{1-\bar{\alpha}_t+\rho^2\bar{\alpha}_t}\boldsymbol{M_Y},\end{aligned} \tag{68}$$

$$\text{Var}(\mathbb{E}[\boldsymbol{X}^0|\boldsymbol{X}^t,\boldsymbol{Y}]|\boldsymbol{X}^t) = (\frac{1-\bar{\alpha}_t}{1-\bar{\alpha}_t+\rho^2\bar{\alpha}_t})^2\text{Var}(\boldsymbol{M_Y}|\boldsymbol{X}^t). \tag{69}$$

Note that

$$\begin{aligned}\mathbb{E}[\text{Var}(\boldsymbol{M_Y}|\boldsymbol{X}^t)] &= \mathbb{E}[\|\boldsymbol{M_Y} - \mathbb{E}[\boldsymbol{M_Y}|\boldsymbol{X}^t]\|^2] \\ &= \mathbb{E}[\|\boldsymbol{M_Y} - \underset{\boldsymbol{A}(\boldsymbol{X}^t)}{\arg\min}\mathbb{E}[\|\boldsymbol{M_Y} - \boldsymbol{A}(\boldsymbol{X}^t)\|^2]\|^2] \\ &\le \mathbb{E}[\|\boldsymbol{M_Y} - \boldsymbol{A}(\boldsymbol{X}^t)\|^2].\end{aligned} \tag{70}$$

Let

$$\boldsymbol{A}(\boldsymbol{X}^t) = \Big(\frac{1}{|\{i:Y_i=Y_p\}|}\sum_{i:Y_i=Y_p}\boldsymbol{x}_i^t\Big)_{p=1}^{P}. \tag{71}$$

Since that

$$\frac{1}{|\{i:Y_i=Y_p\}|}\sum_{i:Y_i=Y_p}\boldsymbol{x}_i^t \sim \mathcal{N}(\boldsymbol{\mu}_{Y_p}, \frac{(\bar{\alpha}_t\rho^2 + (1-\bar{\alpha}_t))\cdot\boldsymbol{I}}{|\{i:Y_i=Y_p\}|}), \tag{72}$$

we have

$$\mathbb{E}[\|\boldsymbol{M_Y} - \boldsymbol{A}(\boldsymbol{X}^t)\|^2] \le \sum_{p=1}^{P}\frac{\rho^2}{|\{i:Y_i=Y_p\}|} = \rho^2 K. \tag{73}$$

Hence, we can obtain

$$R_{\text{Bayes}} - R_{\text{oracle}} \le (\frac{1-\bar{\alpha}_t}{1-\bar{\alpha}_t+\rho^2\bar{\alpha}_t})^2 \cdot \frac{\rho^2}{|\{i:Y_i=Y_p\}|} \le \epsilon, \tag{74}$$

as long as for any $p \in [P], t \in [T]$,

$$|\{i:Y_i=Y_p\}| \ge \epsilon^{-1}\rho^2, \tag{75}$$

which holds if $P \ge \nu_{\min}^{\tilde{\pi}}(K)^{-1}(\rho^2+1)\epsilon^{-1}\log d$.

$\qquad\square$

## C.6. Proof of Proposition 3

*Proof.* By (129), we have that for $Y_i = Y_j$,

$$
\begin{aligned}
&\boldsymbol{x}_j^{t\,\top} \boldsymbol{W}^{(S)} \boldsymbol{x}_i^t / d \\
&\gtrsim \frac{1}{2} \log(\epsilon^{-1} (\frac{(1 - \max_{u \in [M]} \nu_u^{\tilde{\boldsymbol{\pi}}}(K))}{\max_{u \in [M]} \nu_u^{\tilde{\boldsymbol{\pi}}}(K)})^2) \\
&\gtrsim \frac{1}{2} \log \epsilon^{-1} K \delta(\tilde{\boldsymbol{\pi}}),
\end{aligned}
\tag{76}
$$

where the last step holds since

$$
\max_{u \in [M]} \nu_u^{\tilde{\boldsymbol{\pi}}}(K) \le \frac{\max_{u \in [M]} \tilde{\boldsymbol{\pi}}}{\max_{u \in [M]} \tilde{\boldsymbol{\pi}} + (K - 1) \min_{u \in [M]} \tilde{\boldsymbol{\pi}}} \le \frac{1}{K \delta(\tilde{\boldsymbol{\pi}})}.
\tag{77}
$$

This leads to (18). (19) comes from an accumulation of (26). $\qquad\square$

## C.7. Proof of Corollary 3

*Proof.* (20) comes from (140). (21) is derived by (115) plus the training condition (i) in Theorem 1. $\qquad\square$

## C.8. Proof of Corollar 4

*Proof.* We still need the condition (143) to hold for generation on $\mathcal{D}(\tilde{\pi}', K, \{\boldsymbol{\mu}_i\}_{i=1}^M, \rho)$. Then, the required number of tokens per data becomes

$$
P \gtrsim \nu_{\min}^{\tilde{\pi}'}(K)^{-1}(\rho^2 + 1)\epsilon^{-1} \log d.
\tag{78}
$$

Combining Corollary (2), we can obtain the desired result. $\qquad\square$

# D. Proof of Key Lemmas

## D.1. Proof of Lemma 3

*Proof.* Let $\boldsymbol{E}^t = (\boldsymbol{\epsilon}_1, \boldsymbol{\epsilon}_2, \cdots, \boldsymbol{\epsilon}_P)$. Then, we can compute

$$
\begin{aligned}
&\frac{\partial L(\Psi^{(s_0)})}{\partial \boldsymbol{W}} \\
&= \sum_{p=1}^P \frac{\partial \|\boldsymbol{f}(\boldsymbol{W}; \sqrt{\bar{\alpha}_t}\boldsymbol{X}^0 + \sqrt{1 - \bar{\alpha}_t}\boldsymbol{E}^t, t)_p - \boldsymbol{\epsilon}_p\|^2}{\partial \boldsymbol{W}} \cdot \frac{1}{dP} \\
&= \sum_{p=1}^P (\boldsymbol{f}(\boldsymbol{W}; \sqrt{\bar{\alpha}_t}\boldsymbol{X}^0 + \sqrt{1 - \bar{\alpha}_t}\boldsymbol{E}^t, t)_p - \boldsymbol{\epsilon}_p)^\top \frac{\partial \boldsymbol{f}(\boldsymbol{W}; \sqrt{\bar{\alpha}_t}\boldsymbol{X}^0 + \sqrt{1 - \bar{\alpha}_t}\boldsymbol{E}^t, t)}{\partial \boldsymbol{W}} \cdot \frac{1}{dP} \\
&= \sum_{p=1}^P (\boldsymbol{f}(\boldsymbol{W}; \sqrt{\bar{\alpha}_t}\boldsymbol{X}^0 + \sqrt{1 - \bar{\alpha}_t}\boldsymbol{E}^t, t)_p - \boldsymbol{\epsilon}_p)^\top (-\frac{v_t}{d^2 P}) \sum_{i=1}^P \boldsymbol{x}_i^t \mathrm{softmax}_p(\frac{\boldsymbol{x}_i^{t\,\top} \boldsymbol{W} \boldsymbol{x}_p^t}{d}) \\
&\quad \cdot (\boldsymbol{x}_i^t - \sum_{r=1}^P \mathrm{softmax}_p(\frac{\boldsymbol{x}_r^{t\,\top} \boldsymbol{W} \boldsymbol{x}_p^t}{d})\boldsymbol{x}_r^t)\boldsymbol{x}_p^{t\,\top}.
\end{aligned}
\tag{79}
$$

We then complete the proof using induction. When iterations $s = 0$, we have that with a high probability,

$$
\|\boldsymbol{x}_p^t\|^2 \gtrsim (1 + \bar{\alpha}_t \rho^2)d \log d.
\tag{80}
$$

If $\boldsymbol{x}_i^t$ and $\boldsymbol{x}_p^t$ share the same feature $\boldsymbol{\mu}_u$ as the mean, then

$$\boldsymbol{\mu}_u^\top (\boldsymbol{x}_i^t - \sum_{r=1}^{P} \mathrm{softmax}_p(\frac{\boldsymbol{x}_r^{t\top} \boldsymbol{W}^{(0)} \boldsymbol{x}_p^t}{d}) \boldsymbol{x}_r^t) \boldsymbol{x}_p^{t\top} \boldsymbol{\mu}_u$$

$$= (\sqrt{\bar{\alpha}_t} d + \boldsymbol{\mu}_u^\top (\boldsymbol{x}_i^t - \sqrt{\bar{\alpha}_t} \boldsymbol{\mu}_u) - \sum_{r=1}^{P} \mathrm{softmax}_p(\frac{\boldsymbol{x}_r^{t\top} \boldsymbol{W}^{(0)} \boldsymbol{x}_p^t}{d}) \boldsymbol{\mu}_u^\top \boldsymbol{x}_r^t)(\sqrt{\bar{\alpha}_t} d + \boldsymbol{\mu}_u^\top (\boldsymbol{x}_p^t - \sqrt{\bar{\alpha}_t} \boldsymbol{\mu}_u)). \tag{81}$$

Consider $\tilde{\boldsymbol{X}}^0$ independently sampled from Definition x so that $\boldsymbol{x}_j^0$ and $\boldsymbol{x}_{j'}^0$ shares the same mean as $\boldsymbol{x}_p^0$. $\tilde{\boldsymbol{X}}^0 \neq \boldsymbol{X}^0$. Then,

$$\tilde{\boldsymbol{x}}_j^{t\top} (\boldsymbol{x}_i^t - \sum_{r=1}^{P} \mathrm{softmax}_p(\frac{\boldsymbol{x}_r^{t\top} \boldsymbol{W}^{(0)} \boldsymbol{x}_p^t}{d}) \boldsymbol{x}_r^t) \boldsymbol{x}_p^{t\top} \tilde{\boldsymbol{x}}_{j'}^t,$$

$$\gtrsim (\bar{\alpha}_t d - \sum_{r=1}^{P} \mathrm{softmax}_p(\frac{\boldsymbol{x}_r^{t\top} \boldsymbol{W}^{(0)} \boldsymbol{x}_p^t}{d}) \tilde{\boldsymbol{x}}_j^{t\top} \boldsymbol{x}_r^t) \cdot \bar{\alpha}_t d, \tag{82}$$

Meanwhile,

$$(\boldsymbol{f}(\boldsymbol{W}; \sqrt{\bar{\alpha}_t} \boldsymbol{X}^0 + \sqrt{1-\bar{\alpha}_t} \boldsymbol{E}^t, t)_p - \boldsymbol{\epsilon}_p)^\top \boldsymbol{x}_i^t$$

$$= (v_t^{(0)} \sqrt{\bar{\alpha}_t} \rho \boldsymbol{\epsilon}_p' + v_t^{(0)} (\sqrt{\bar{\alpha}_t} \boldsymbol{\mu}_u - \sum_{i=1}^{P} \boldsymbol{x}_i^t \mathrm{softmax}_p(\frac{\boldsymbol{x}_i^{t\top} \boldsymbol{W}^{(0)} \boldsymbol{x}_p^t}{d})) - (v_t^{(0)} \sqrt{1-\bar{\alpha}_t} - 1) \boldsymbol{\epsilon}_p)(\sqrt{\bar{\alpha}_t}(\boldsymbol{\mu}_u + \rho \boldsymbol{\epsilon}_p') \tag{83}$$

$$+ \sqrt{1-\bar{\alpha}_t} \boldsymbol{\epsilon}_p).$$

If $\boldsymbol{x}_i^t$ and $\boldsymbol{x}_p^t$ does not share the same feature $\boldsymbol{\mu}_u$ as the mean, then for $\tilde{\boldsymbol{x}}_j^t$ and $\tilde{\boldsymbol{x}}_{j'}^t$ that is from another $\tilde{\boldsymbol{X}}^t$ with the same mean as $\boldsymbol{x}_p^t$,

$$\tilde{\boldsymbol{x}}_j^\top (\boldsymbol{x}_i^t - \sum_{r=1}^{P} \mathrm{softmax}_p(\frac{\boldsymbol{x}_r^{t\top} \boldsymbol{W}^{(0)} \boldsymbol{x}_p^t}{d}) \boldsymbol{x}_r^t) \boldsymbol{x}_p^{t\top} \tilde{\boldsymbol{x}}_{j'}$$

$$= (\tilde{\boldsymbol{x}}_j^\top (\boldsymbol{x}_i^t - \sqrt{\bar{\alpha}_t} \boldsymbol{\mu}_{u'}) - \sum_{r=1}^{P} \mathrm{softmax}_p(\frac{\boldsymbol{x}_r^{t\top} \boldsymbol{W}^{(0)} \boldsymbol{x}_p^t}{d}) \tilde{\boldsymbol{x}}_j^\top \boldsymbol{x}_r^t) \cdot (\bar{\alpha}_t d + \tilde{\boldsymbol{x}}_{j'}^\top (\boldsymbol{x}_p^t - \sqrt{\bar{\alpha}_t} \boldsymbol{\mu}_u)), \tag{84}$$

$$(\boldsymbol{f}(\boldsymbol{W}; \sqrt{\bar{\alpha}_t} \boldsymbol{X}^0 + \sqrt{1-\bar{\alpha}_t} \boldsymbol{E}^t, t)_p - \boldsymbol{\epsilon}_p)^\top \boldsymbol{x}_i^t$$

$$= (v_t^{(0)} \sqrt{\bar{\alpha}_t} \rho \boldsymbol{\epsilon}_p' + v_t^{(0)} (\sqrt{\bar{\alpha}_t} \boldsymbol{\mu}_u - \sum_{i=1}^{P} \boldsymbol{x}_i^t \mathrm{softmax}_p(\frac{\boldsymbol{x}_i^{t\top} \boldsymbol{W}^{(0)} \boldsymbol{x}_p^t}{d})) - (v_t^{(0)} \sqrt{1-\bar{\alpha}_t} - 1) \boldsymbol{\epsilon}_p)^\top (\sqrt{\bar{\alpha}_t}(\boldsymbol{\mu}_{u'} + \rho \boldsymbol{\epsilon}_i') \tag{85}$$

$$+ \sqrt{1-\bar{\alpha}_t} \boldsymbol{\epsilon}_i),$$

if the mean of $\boldsymbol{x}_i^t$ is $\boldsymbol{\mu}_{u'}$. Therefore, combining (82) and (83), we have that for $\tilde{\boldsymbol{x}}_j^t$ and $\tilde{\boldsymbol{x}}_{j'}^t$ that share the same mean as $\boldsymbol{x}_p^t$,

$$(-\tilde{\boldsymbol{x}}_j^{t\top}) \mathbb{E}_{\boldsymbol{E}^t, \boldsymbol{X}^0} \big[ (\boldsymbol{f}(\boldsymbol{W}^{(0)}; \sqrt{\bar{\alpha}_t} \boldsymbol{X}^0 + \sqrt{1-\bar{\alpha}_t} \boldsymbol{E}^t, t)_p - \boldsymbol{\epsilon}_p)^\top (-\frac{v_t^{(0)}}{d^2 P}) \sum_{i=1}^{P} \boldsymbol{x}_i^t \mathrm{softmax}_p(\frac{\boldsymbol{x}_i^{t\top} \boldsymbol{W}^{(0)} \boldsymbol{x}_p^t}{d})$$

$$\cdot (\boldsymbol{x}_i^t - \sum_{r=1}^{P} \mathrm{softmax}(\frac{\boldsymbol{x}_r^{t\top} \boldsymbol{W}^{(0)} \boldsymbol{x}_p^t}{d}) \boldsymbol{x}_r^t) \boldsymbol{x}_p^{t\top} \big] \tilde{\boldsymbol{x}}_{j'}^t$$

$$\gtrsim \mathbb{E}_{\boldsymbol{E}^t, \boldsymbol{X}^0} \big[ \frac{1}{P} \mathbb{1}[Y_p = u] \sum_{i=1}^{P} \zeta_{i,p,t}(0)(v_t^{(0)})^2 \cdot (d\bar{\alpha}_t^3 (1 - \sum_{l=1}^{P} \zeta_{l,p,t}^u(0))^2 + \bar{\alpha}_t^3 d\rho^2 - \bar{\alpha}_t^2 (1 - \bar{\alpha}_t - \frac{\sqrt{1-\bar{\alpha}_t}}{v_t^{(0)}}) d \mathbb{1}[i = p]$$

$$- d\bar{\alpha}_t^3 \zeta_{i,p,t}^u(0)\rho^2) \big], \tag{86}$$

$$\frac{1}{T}\sum_{t=1}^{T}(-\tilde{\boldsymbol{x}}_j^{t\top})\mathbb{E}_{\boldsymbol{E}^t,\boldsymbol{X}^0}\Big[\sum_{p=1}^{P}(\boldsymbol{f}(\boldsymbol{W}^{(1)};\sqrt{\bar{\alpha}_t}\boldsymbol{X}^0+\sqrt{1-\bar{\alpha}_t}\boldsymbol{E}^t,t)_p-\boldsymbol{\epsilon}_p)^{\top}(-\frac{v_t^{(1)}}{d^2P})\sum_{i=1}^{P}\boldsymbol{x}_i^t$$

$$\cdot\,\text{softmax}_p(\frac{\boldsymbol{x}_i^{t\top}\boldsymbol{W}^{(1)}\boldsymbol{x}_p^t}{d})\cdot(\boldsymbol{x}_i^t-\sum_{r=1}^{P}\text{softmax}_p(\frac{\boldsymbol{x}_r^{t\top}\boldsymbol{W}^{(1)}\boldsymbol{x}_p^t}{d})\boldsymbol{x}_r^t)\boldsymbol{x}_p^{t\top}]\tilde{\boldsymbol{x}}_{j'}^t \tag{87}$$

$$\gtrsim\frac{1}{T}\sum_{t=1}^{T}\sum_{p=1}^{P}\frac{\mathbb{1}[Y_p=u]}{P^2}\mathbb{E}_{\boldsymbol{E}^t,\boldsymbol{X}^0}[v_t^{(0)}\bar{\alpha}_t^2\sqrt{1-\bar{\alpha}_t}d(1-\sum_{l=1}^{P}\zeta_{l,p,t}^u(0))^2]$$

$$\gtrsim v_t^{(0)}\cdot\frac{d}{P},$$

which means that the update from $\boldsymbol{W}^{(0)}$ to $\boldsymbol{W}^{(1)}$ almost makes no difference to the attention map. Note that

$$\frac{\partial\mathbb{E}_{\boldsymbol{E}^t,\boldsymbol{X}^0}[\|v_t^{(0)}(\boldsymbol{x}_p^t-\sum_{i=1}^{P}\boldsymbol{x}_i^t\text{softmax}_p(\frac{\boldsymbol{x}_i^{t\top}\boldsymbol{W}^{(0)}\boldsymbol{x}_p^t}{d}))-\boldsymbol{\epsilon}_p\|^2/d]}{\partial v_t}$$

$$=v_t^{(0)}\rho^2\bar{\alpha}_t+\sqrt{1-\bar{\alpha}_t}(v_t^{(0)}\sqrt{1-\bar{\alpha}_t}-1)+\mathbb{E}_{\boldsymbol{E}^t,\boldsymbol{X}^0}[v_t^{(0)}\|\sqrt{\bar{\alpha}_t}\boldsymbol{\mu}_u-\sum_{i=1}^{P}\boldsymbol{x}_i^t\text{softmax}_p(\frac{\boldsymbol{x}_i^{t\top}\boldsymbol{W}^{(0)}\boldsymbol{x}_p^t}{d})\|^2/d]$$

$$+2\mathbb{E}_{\boldsymbol{E}^t,\boldsymbol{X}^0}\Big[\frac{\beta_1 v_t^{(0)}\rho\sqrt{\bar{\alpha}_t}}{d}\Big]+\mathbb{E}_{\boldsymbol{E}^t,\boldsymbol{X}^0}\Big[\frac{2\beta_2\sqrt{1-\bar{\alpha}_t}v_t^{(0)}-\beta_2}{d}\Big] \tag{88}$$

$$=(v_t^{(0)}\cdot\mathbb{E}_{\boldsymbol{E}^t,\boldsymbol{X}^0}[\bar{\alpha}_t\rho^2+1-\bar{\alpha}_t+\|\sqrt{\bar{\alpha}_t}\boldsymbol{\mu}_u-\sum_{i=1}^{P}\boldsymbol{x}_i^t\text{softmax}_p(\frac{\boldsymbol{x}_i^{t\top}\boldsymbol{W}^{(0)}\boldsymbol{x}_p^t}{d})\|^2/d$$

$$+\frac{2\beta_1\rho\sqrt{\bar{\alpha}_t}}{d}+\frac{2\beta_2\sqrt{1-\bar{\alpha}_t}}{d}]-(\sqrt{1-\bar{\alpha}_t}+\frac{\beta_2}{d})),$$

where we denote $\beta_1=\boldsymbol{\epsilon}_p'^{\top}(\sqrt{\bar{\alpha}_t}\boldsymbol{\mu}_u-\sum_{i=1}^{P}\boldsymbol{x}_i^t\text{softmax}_p(\frac{\boldsymbol{x}_i^{t\top}\boldsymbol{W}^{(0)}\boldsymbol{x}_p^t}{d}))$ and $\beta_2=\boldsymbol{\epsilon}_p^{\top}(\sqrt{\bar{\alpha}_t}\boldsymbol{\mu}_u-\sum_{i=1}^{P}\boldsymbol{x}_i^t\text{softmax}_p(\frac{\boldsymbol{x}_i^{t\top}\boldsymbol{W}^{(0)}\boldsymbol{x}_p^t}{d}))$. We have $\beta_1,\beta_2\lesssim\sqrt{d}$. Hence, since $|v_t^{(0)}|\lesssim\sqrt{\log d/d}$, we have

$$v_t^{(1)}=v_t^{(0)}-\eta\partial\mathbb{E}_{\boldsymbol{E}^t,\boldsymbol{X}^0}[\|v_t^{(0)}(\boldsymbol{x}_p^t-\sum_{i=1}^{P}\boldsymbol{x}_i^t\text{softmax}_p(\frac{\boldsymbol{x}_i^{t\top}\boldsymbol{W}^{(0)}\boldsymbol{x}_p^t}{d}))-\boldsymbol{\epsilon}_p\|^2/d]/\partial v_t$$

$$<\eta\sqrt{1-\bar{\alpha}_t}, \tag{89}$$

and

$$1-\bar{\alpha}_t-\frac{\sqrt{1-\bar{\alpha}_t}}{v_t^{(0)}}<1-\eta^{-1}-\bar{\alpha}_t<0. \tag{90}$$

Then, we can obtain

$$(-\tilde{\boldsymbol{x}}_j^{t\top})\mathbb{E}_{\boldsymbol{E}^t,\boldsymbol{X}^0}\Big[(\boldsymbol{f}(\boldsymbol{W}^{(1)};\sqrt{\bar{\alpha}_t}\boldsymbol{X}^0+\sqrt{1-\bar{\alpha}_t}\boldsymbol{E}^t,t)_p-\boldsymbol{\epsilon}_p)^{\top}(-\frac{v_t^{(1)}}{d^2P})\sum_{i=1}^{P}\boldsymbol{x}_i^t\text{softmax}_p(\frac{\boldsymbol{x}_i^{t\top}\boldsymbol{W}^{(1)}\boldsymbol{x}_p^t}{d})$$

$$\cdot(\boldsymbol{x}_i^t-\sum_{r=1}^{P}\text{softmax}(\frac{\boldsymbol{x}_r^{t\top}\boldsymbol{W}^{(1)}\boldsymbol{x}_p^t}{d})\boldsymbol{x}_r^t)\boldsymbol{x}_p^{t\top}]\tilde{\boldsymbol{x}}_{j'}^t$$

$$\gtrsim\mathbb{E}_{\boldsymbol{E}^t,\boldsymbol{X}^0}\Big[\frac{1}{P}\mathbb{1}[Y_p=u]\sum_{i=1}^{P}\zeta_{i,p,t}(1)(v_t^{(1)})^2\cdot(d\bar{\alpha}_t^3(1-\sum_{l=1}^{P}\zeta_{l,p,t}^u(1))^2+\bar{\alpha}_t^3d\rho^2-\bar{\alpha}_t^2(1-\bar{\alpha}_t-\frac{\sqrt{1-\bar{\alpha}_t}}{v_t^{(1)}})d\mathbb{1}[i=p]$$

$$-d\bar{\alpha}_t^3\zeta_{i,p,t}^u(1)\rho^2)]$$

$$\gtrsim\mathbb{E}_{\boldsymbol{E}^t,\boldsymbol{X}^0}\Big[\frac{1}{P}\mathbb{1}[Y_p=u]\sum_{i=1}^{P}\zeta_{i,p,t}(1)(v_t^{(1)})^2\cdot(d\bar{\alpha}_t^3(1-\sum_{l=1}^{P}\zeta_{l,p,t}^u(1))^2],$$

$$\tag{91}$$

where the last step holds if $v_t^{(0)} > 0$. Therefore, by summing up over (91), we can obtain that for any $\tilde{\boldsymbol{x}}_j^t$ and $\tilde{\boldsymbol{x}}_{j'}$ with $\boldsymbol{\mu}_u$ as the mean,

$$
\frac{1}{T}\sum_{t=1}^{T}(-\tilde{\boldsymbol{x}}_j^{t\top})\mathbb{E}_{\boldsymbol{E}^t,\boldsymbol{X}^0}\Big[\sum_{p=1}^{P}(\boldsymbol{f}(\boldsymbol{W}^{(1)};\sqrt{\bar{\alpha}_t}\boldsymbol{X}^0+\sqrt{1-\bar{\alpha}_t}\boldsymbol{E}^t,t)_p-\boldsymbol{\epsilon}_p)^{\top}(-\frac{v_t^{(1)}}{d^2P})\sum_{i=1}^{P}\boldsymbol{x}_i^t
$$
$$
\cdot\mathrm{softmax}_p(\frac{\boldsymbol{x}_i^{t\top}\boldsymbol{W}^{(1)}\boldsymbol{x}_p^t}{d})\cdot(\boldsymbol{x}_i^t-\sum_{r=1}^{P}\mathrm{softmax}_p(\frac{\boldsymbol{x}_r^{t\top}\boldsymbol{W}^{(1)}\boldsymbol{x}_p^t}{d})\boldsymbol{x}_r^t)\boldsymbol{x}_p^{t\top}\Big]\tilde{\boldsymbol{x}}_{j'}^t
$$
$$
\gtrsim\frac{1}{T}\sum_{t=1}^{T}\sum_{p=1}^{P}\frac{\mathbb{1}[Y_p=u]}{P}\mathbb{E}_{\boldsymbol{E}^t,\boldsymbol{X}^0}[(v_t^{(1)})^2\bar{\alpha}_t^3d(1-\sum_{l=1}^{P}\zeta_{l,p,t}^u(0))^2\sum_{i=1}^{P}\zeta_{i,p,t}^u(0)]
$$
$$
\gtrsim\frac{1}{T}\sum_{t=1}^{T}\mathbb{E}_{\boldsymbol{E}^t,\boldsymbol{X}^0}[(v_t^{(1)})^2\bar{\alpha}_t^3d(1-\sum_{l=1}^{P}\zeta_{l,p,t}^u(0))^2\sum_{i=1}^{P}\zeta_{i,p,t}^u(0)\cdot\nu_u^{\tilde{\boldsymbol{\pi}}}(K),
\tag{92}
$$

where the last step holds with a high probability if

$$
P\cdot\nu_u^{\tilde{\boldsymbol{\pi}}}(K)\gtrsim\log dK,
\tag{93}
$$

by Lemma 1. For $\boldsymbol{\mu}_u\neq\boldsymbol{\mu}_{u'}$, we consider two cases of $\boldsymbol{x}_i^t$ as follows, where the mean of $\tilde{\boldsymbol{\mu}}_k$ is $\boldsymbol{\mu}_{u'}$.

1. The corresponding mean of $\boldsymbol{x}_i^t$ is $\boldsymbol{\mu}_{u'}$: Then, $\boldsymbol{\mu}_{u'}^{\top}(\boldsymbol{x}_i^t-\sum_{r=1}^{P}\mathrm{softmax}_p(\frac{\boldsymbol{x}_r^{t\top}\boldsymbol{W}^{(1)}\boldsymbol{x}_p^t}{d})\boldsymbol{x}_r^t)\boldsymbol{x}_p^{t\top}\boldsymbol{\mu}_u$ is in the order of $d^2$. $(\boldsymbol{f}(\boldsymbol{W}^{(1)};\sqrt{\bar{\alpha}_t}\boldsymbol{X}^0+\sqrt{1-\bar{\alpha}_t}\boldsymbol{E}^t,t)_p-\boldsymbol{\epsilon}_p)^{\top}\boldsymbol{x}_i^t$ is in the order of $-d$ if $d$ is large enough, and the order of positive term is no more than $\sqrt{d}\log d$.

2. The corresponding mean of $\boldsymbol{x}_i^t$ is not $\boldsymbol{\mu}_{u'}$: Then, the order of $\boldsymbol{\mu}_{u'}^{\top}(\boldsymbol{x}_i^t-\sum_{r=1}^{P}\mathrm{softmax}_p(\frac{\boldsymbol{x}_r^{t\top}\boldsymbol{W}^{(1)}\boldsymbol{x}_p^t}{d})\boldsymbol{x}_r^t)\boldsymbol{x}_p^{t\top}\boldsymbol{\mu}_u$ is at most $d^{\frac{3}{2}}\log d$, which is already smaller than the order of $d^2$. $(\boldsymbol{f}(\boldsymbol{W}^{(1)};\sqrt{\bar{\alpha}_t}\boldsymbol{X}^0+\sqrt{1-\bar{\alpha}_t}\boldsymbol{E}^t,t)_p-\boldsymbol{\epsilon}_p)^{\top}\boldsymbol{x}_i^t$ is in the order of at most $\sqrt{d}\log d$.

The above discussion indicates that for $\tilde{\boldsymbol{x}}_j$ and $\tilde{\boldsymbol{x}}_{j'}$ with the mean of $\boldsymbol{\mu}_u$ and $\tilde{\boldsymbol{x}}_k$ with the mean of $\boldsymbol{\mu}_{u'}$,

$$
(-\tilde{\boldsymbol{x}}_k^{\top})\mathbb{E}_{\boldsymbol{E}^t,\boldsymbol{X}^0}\Big[(\boldsymbol{f}(\boldsymbol{W}^{(1)};\sqrt{\bar{\alpha}_t}\boldsymbol{X}^0+\sqrt{1-\bar{\alpha}_t}\boldsymbol{E}^t,t)_p-\boldsymbol{\epsilon}_p)^{\top}(-\frac{v_t^{(1)}}{d^2P})\sum_{i=1}^{P}\boldsymbol{x}_i^t
$$
$$
\cdot\mathrm{softmax}_p(\frac{\boldsymbol{x}_i^{t\top}\boldsymbol{W}^{(1)}\boldsymbol{x}_p^t}{d})\cdot(\boldsymbol{x}_i^t-\sum_{r=1}^{P}\mathrm{softmax}_p(\frac{\boldsymbol{x}_r^{t\top}\boldsymbol{W}^{(1)}\boldsymbol{x}_p^t}{d})\boldsymbol{x}_r^t)\boldsymbol{x}_p^{t\top}\Big]\tilde{\boldsymbol{x}}_j
$$
$$
\lesssim\frac{\log d}{\sqrt{d}}\cdot(-\tilde{\boldsymbol{x}}_{j'}^{\top})\mathbb{E}_{\boldsymbol{E}^t,\boldsymbol{X}^0}\Big[(\boldsymbol{f}(\boldsymbol{W}^{(1)};\sqrt{\bar{\alpha}_t}\boldsymbol{X}^0+\sqrt{1-\bar{\alpha}_t}\boldsymbol{E}^t,t)_p-\boldsymbol{\epsilon}_p)^{\top}(-\frac{v_t^{(1)}}{d^2P})\sum_{i=1}^{P}\boldsymbol{x}_i^t
$$
$$
\cdot\mathrm{softmax}_p(\frac{\boldsymbol{x}_i^{t\top}\boldsymbol{W}^{(1)}\boldsymbol{x}_p^t}{d})\cdot(\boldsymbol{x}_i^t-\sum_{r=1}^{P}\mathrm{softmax}_p(\frac{\boldsymbol{x}_r^{t\top}\boldsymbol{W}^{(1)}\boldsymbol{x}_p^t}{d})\boldsymbol{x}_r^t)\boldsymbol{x}_p^{t\top}\Big]\tilde{\boldsymbol{x}}_j
\tag{94}
$$

$$
\frac{1}{T}\sum_{t=1}^{T}(-\tilde{\boldsymbol{x}}_k^{\top})\mathbb{E}_{\boldsymbol{E}^t,\boldsymbol{X}^0}\Big[(\boldsymbol{f}(\boldsymbol{W}^{(1)};\sqrt{\bar{\alpha}_t}\boldsymbol{X}^0+\sqrt{1-\bar{\alpha}_t}\boldsymbol{E}^t,t)_p-\boldsymbol{\epsilon}_p)^{\top}(-\frac{v_t^{(1)}}{d^2P})\sum_{i=1}^{P}\boldsymbol{x}_i^t
$$
$$
\cdot\mathrm{softmax}_p(\frac{\boldsymbol{x}_i^{t\top}\boldsymbol{W}^{(1)}\boldsymbol{x}_p^t}{d})\cdot(\boldsymbol{x}_i^t-\sum_{r=1}^{P}\mathrm{softmax}_p(\frac{\boldsymbol{x}_r^{t\top}\boldsymbol{W}^{(1)}\boldsymbol{x}_p^t}{d})\boldsymbol{x}_r^t)\boldsymbol{x}_p^{t\top}\Big]\tilde{\boldsymbol{x}}_j
$$
$$
\leq\frac{\log d}{\sqrt{d}}\cdot(-\tilde{\boldsymbol{x}}_{j'}^{\top})\frac{1}{T}\sum_{t=1}^{T}\mathbb{E}_{\boldsymbol{E}^t,\boldsymbol{X}^0}\Big[(\boldsymbol{f}(\boldsymbol{W}^{(1)};\sqrt{\bar{\alpha}_t}\boldsymbol{X}^0+\sqrt{1-\bar{\alpha}_t}\boldsymbol{E}^t,t)_p-\boldsymbol{\epsilon}_p)^{\top}(-\frac{v_t^{(1)}}{d^2P})\sum_{i=1}^{P}\boldsymbol{x}_i^t
$$
$$
\cdot\mathrm{softmax}_p(\frac{\boldsymbol{x}_i^{t\top}\boldsymbol{W}^{(1)}\boldsymbol{x}_p^t}{d})\cdot(\boldsymbol{x}_i^t-\sum_{r=1}^{P}\mathrm{softmax}_p(\frac{\boldsymbol{x}_r^{t\top}\boldsymbol{W}^{(1)}\boldsymbol{x}_p^t}{d})\boldsymbol{x}_r^t)\boldsymbol{x}_p^{t\top}\Big]\tilde{\boldsymbol{x}}_j.
\tag{95}
$$

Suppose that when iterations $s = s_0 > 1$, the conclusion holds. Then, when $s = s_0 + 1$, we have that if $\boldsymbol{x}_i^t$ and $\boldsymbol{x}_p^t$ share the same feature $\boldsymbol{\mu}_u$ as the mean, then

$$\boldsymbol{\mu}_u^\top(\boldsymbol{x}_i^t - \sum_{r=1}^{P} \text{softmax}_p(\frac{\boldsymbol{x}_r^{t\top}\boldsymbol{W}^{(s_0)}\boldsymbol{x}_p^t}{d})\boldsymbol{x}_r^t)\boldsymbol{x}_p^{t\top}\boldsymbol{\mu}_u$$

$$= (\sqrt{\bar{\alpha}_t}d + \boldsymbol{\mu}_u^\top(\boldsymbol{x}_i^t - \sqrt{\bar{\alpha}_t}\boldsymbol{\mu}_u)) - \sum_{r=1}^{P}\text{softmax}_p(\frac{\boldsymbol{x}_r^{t\top}\boldsymbol{W}^{(s_0)}\boldsymbol{x}_p^t}{d})\boldsymbol{\mu}_u^\top\boldsymbol{x}_r^t)(\sqrt{\bar{\alpha}_t}d + \boldsymbol{\mu}_u^\top(\boldsymbol{x}_p^t - \sqrt{\bar{\alpha}_t}\boldsymbol{\mu}_u)). \tag{96}$$

Consider $\tilde{\boldsymbol{X}}^0$ independently sampled from Definition x so that $\boldsymbol{x}_j^0$ and $\boldsymbol{x}_{j'}^0$ share the same mean as $\boldsymbol{x}_p^0$. $\tilde{\boldsymbol{X}}^0 \neq \boldsymbol{X}^0$. Then,

$$\tilde{\boldsymbol{x}}_j^{t\top}(\boldsymbol{x}_i^t - \sum_{r=1}^{P}\text{softmax}_p(\frac{\boldsymbol{x}_r^{t\top}\boldsymbol{W}^{(s_0)}\boldsymbol{x}_p^t}{d})\boldsymbol{x}_r^t)\boldsymbol{x}_p^{t\top}\tilde{\boldsymbol{x}}_{j'}^t$$

$$\gtrsim (\bar{\alpha}_t d - \sum_{r=1}^{P}\text{softmax}_p(\frac{\boldsymbol{x}_r^{t\top}\boldsymbol{W}^{(s_0)}\boldsymbol{x}_p^t}{d})\tilde{\boldsymbol{x}}_j^{t\top}\boldsymbol{x}_r^t) \cdot \bar{\alpha}_t d, \tag{97}$$

Meanwhile,

$$(\boldsymbol{f}(\boldsymbol{W}^{(s_0)}; \sqrt{\bar{\alpha}_t}\boldsymbol{X}^0 + \sqrt{1 - \bar{\alpha}_t}\boldsymbol{E}^t, t)_p - \boldsymbol{\epsilon}_p)^\top\boldsymbol{x}_i^t$$

$$= (v_t^{(s_0)}\sqrt{\bar{\alpha}_t}\rho\boldsymbol{\epsilon}_p' + v_t^{(s_0)}(\sqrt{\bar{\alpha}_t}\boldsymbol{\mu}_u - \sum_{i=1}^{P}\boldsymbol{x}_i^t\text{softmax}_p(\boldsymbol{x}_i^{t\top}\boldsymbol{W}^{(s_0)}\boldsymbol{x}_p^t)) - (v_t^{(s_0)}\sqrt{1 - \bar{\alpha}_t} - 1)\boldsymbol{\epsilon}_p)(\sqrt{\bar{\alpha}_t}(\boldsymbol{\mu}_u + \rho\boldsymbol{\epsilon}_p') \tag{98}$$

$$+ \sqrt{1 - \bar{\alpha}_t}\boldsymbol{\epsilon}_p).$$

If $\boldsymbol{x}_i^t$ and $\boldsymbol{x}_p^t$ do not share the same feature $\boldsymbol{\mu}_u$ as the mean, then for $\boldsymbol{x}_j^t$ and $\boldsymbol{x}_{j'}^t$ that share the same mean as $\boldsymbol{x}_p^t$,

$$\tilde{\boldsymbol{x}}_j^\top(\boldsymbol{x}_i^t - \sum_{r=1}^{P}\text{softmax}_p(\frac{\boldsymbol{x}_r^{t\top}\boldsymbol{W}^{(s_0)}\boldsymbol{x}_p^t}{d})\boldsymbol{x}_r^t)\boldsymbol{x}_p^{t\top}\tilde{\boldsymbol{x}}_{j'}$$

$$= (\tilde{\boldsymbol{x}}_j^\top(\boldsymbol{x}_i^t - \sqrt{\bar{\alpha}_t}\boldsymbol{\mu}_{u'}) - \sum_{r=1}^{P}\text{softmax}_p(\frac{\boldsymbol{x}_r^{t\top}\boldsymbol{W}^{(s_0)}\boldsymbol{x}_p^t}{d})\tilde{\boldsymbol{x}}_j^\top\boldsymbol{x}_r^t) \cdot (\sqrt{\bar{\alpha}_t}d + \tilde{\boldsymbol{x}}_{j'}^\top(\boldsymbol{x}_p^t - \sqrt{\bar{\alpha}_t}\boldsymbol{\mu}_u)), \tag{99}$$

$$(\boldsymbol{f}(\boldsymbol{W}^{(s_0)}; \sqrt{\bar{\alpha}_t}\boldsymbol{X}^0 + \sqrt{1 - \bar{\alpha}_t}\boldsymbol{E}^t, t)_p - \boldsymbol{\epsilon}_p)^\top\boldsymbol{x}_i^t$$

$$= (v_t^{(s_0)}\sqrt{\bar{\alpha}_t}\rho\boldsymbol{\epsilon}_p' + v_t^{(s_0)}(\sqrt{\bar{\alpha}_t}\boldsymbol{\mu}_u - \sum_{i=1}^{P}\boldsymbol{x}_i^t\text{softmax}_p(\boldsymbol{x}_i^{t\top}\boldsymbol{W}^{(s_0)}\boldsymbol{x}_p^t)) - (v_t^{(s_0)}\sqrt{1 - \bar{\alpha}_t} - 1)\boldsymbol{\epsilon}_p)(\sqrt{\bar{\alpha}_t}(\boldsymbol{\mu}_{u'} + \rho\boldsymbol{\epsilon}_i') \tag{100}$$

$$+ \sqrt{1 - \bar{\alpha}_t}\boldsymbol{\epsilon}_i),$$

if the mean of $\boldsymbol{x}_i^t$ is $\boldsymbol{\mu}_{u'}$. Therefore, combining (97) and (98), we have that for $\tilde{\boldsymbol{x}}_j^t$ and $\tilde{\boldsymbol{x}}_{j'}^t$ from another $\tilde{\boldsymbol{X}}^t$ that share the same mean as $\boldsymbol{x}_p^t$,

$$(-\tilde{\boldsymbol{x}}_j^{t\top})\mathbb{E}_{\boldsymbol{E}^t, \boldsymbol{X}^0}\Big[(\boldsymbol{f}(\boldsymbol{W}^{(s_0)}; \sqrt{\bar{\alpha}_t}\boldsymbol{X}^0 + \sqrt{1 - \bar{\alpha}_t}\boldsymbol{E}^t, t)_p - \boldsymbol{\epsilon}_p)^\top(-\frac{v_t^{(s_0)}}{d^2 P})\sum_{i=1}^{P}\boldsymbol{x}_i^t\text{softmax}_p(\frac{\boldsymbol{x}_i^{t\top}\boldsymbol{W}^{(s_0)}\boldsymbol{x}_p^t}{d})$$

$$\cdot (\boldsymbol{x}_i^t - \sum_{r=1}^{P}\text{softmax}(\frac{\boldsymbol{x}_r^{t\top}\boldsymbol{W}^{(s_0)}\boldsymbol{x}_p^t}{d})\boldsymbol{x}_r^t)\boldsymbol{x}_p^{t\top}\Big]\tilde{\boldsymbol{x}}_{j'}^t$$

$$\gtrsim \mathbb{E}_{\boldsymbol{E}^t, \boldsymbol{X}^0}\Big[\frac{1}{P}\mathbb{1}[Y_p = u]\sum_{i=1}^{P}\zeta_{i,p,t}(s_0)(v_t^{(s_0)})^2 \cdot (d\bar{\alpha}_t^3(1 - \sum_{l=1}^{P}\zeta_{l,p,t}^u(s_0)) + \bar{\alpha}_t^3 d\rho^2 - \bar{\alpha}_t^2(1 - \bar{\alpha}_t - \frac{\sqrt{1 - \bar{\alpha}_t}}{v_t^{(s_0)}})d\mathbb{1}[i = p]$$

$$- d\bar{\alpha}_t^3\sum_{p=1}^{P}\zeta_{i,p,t}^u(s_0)\rho^2)\Big]$$

$$\gtrsim \mathbb{E}_{\boldsymbol{E}^t, \boldsymbol{X}^0}\Big[\frac{1}{P}\mathbb{1}[Y_p = u]\sum_{i=1}^{P}\zeta_{i,p,t}(s_0)(v_t^{(s_0)})^2 \cdot (d\bar{\alpha}_t^3(1 - \sum_{l=1}^{P}\zeta_{l,p,t}^u(s_0))^2\Big], \tag{101}$$

where the last step holds because if $v_t^{(1)} < \frac{\sqrt{1-\bar{\alpha}_t}}{\bar{\alpha}_t \rho^2 + 1 - \bar{\alpha}_t} + o(1)$, then $v_t^{(s_0)} < \frac{\sqrt{1-\bar{\alpha}_t}}{\bar{\alpha}_t \rho^2 + 1 - \bar{\alpha}_t} + o(1)$ and

$$1 - \bar{\alpha}_t - \frac{\sqrt{1-\bar{\alpha}_t}}{v_t^{(s_0)}} < -\rho^2 < 0; \tag{102}$$

and if $v_t^{(1)} > \frac{\sqrt{1-\bar{\alpha}_t}}{\bar{\alpha}_t \rho^2 + 1 - \bar{\alpha}_t} + o(1)$, then $v_t^{(s_0)} > \frac{\sqrt{1-\bar{\alpha}_t}}{\bar{\alpha}_t \rho^2 + 1 - \bar{\alpha}_t} + o(1)$ and

$$1 - \bar{\alpha}_t - \frac{\sqrt{1-\bar{\alpha}_t}}{v_t^{(0)}} < 0. \tag{103}$$

For $\boldsymbol{\mu}_u \neq \boldsymbol{\mu}_{u'}$ and $\tilde{\boldsymbol{x}}_j, \tilde{\boldsymbol{x}}_{j'}$ with the mean of $\boldsymbol{\mu}_u$ and $\tilde{\boldsymbol{x}}_k$ with the mean of $\boldsymbol{\mu}_{u'}$, we have that

$$
\begin{aligned}
&(-\tilde{\boldsymbol{x}}_k^\top) \mathbb{E}_{\boldsymbol{E}^t, \boldsymbol{X}^0} \Big[ (\boldsymbol{f}(\boldsymbol{W}^{(s_0)}; \sqrt{\bar{\alpha}_t} \boldsymbol{X}^0 + \sqrt{1-\bar{\alpha}_t} \boldsymbol{E}^t, t)_p - \boldsymbol{\epsilon}_p)^\top (-\frac{v_t^{(s_0)}}{d^2 P}) \sum_{i=1}^P \boldsymbol{x}_i^t \\
&\quad \cdot \mathrm{softmax}_p(\frac{\boldsymbol{x}_i^{t\top} \boldsymbol{W}^{(s_0)} \boldsymbol{x}_p^t}{d}) \cdot (\boldsymbol{x}_i^t - \sum_{r=1}^P \mathrm{softmax}_p(\frac{\boldsymbol{x}_r^{t\top} \boldsymbol{W}^{(s_0)} \boldsymbol{x}_p^t}{d}) \boldsymbol{x}_r^t) \boldsymbol{x}_p^{t\top} \Big] \tilde{\boldsymbol{x}}_{j'} \\
\leq & \frac{\log d}{\sqrt{d}} \cdot (-\tilde{\boldsymbol{x}}_j^\top) \mathbb{E}_{\boldsymbol{E}^t, \boldsymbol{X}^0} \Big[ (\boldsymbol{f}(\boldsymbol{W}^{(s_0)}; \sqrt{\bar{\alpha}_t} \boldsymbol{X}^0 + \sqrt{1-\bar{\alpha}_t} \boldsymbol{E}^t, t)_p - \boldsymbol{\epsilon}_p)^\top (-\frac{v_t^{(s_0)}}{d^2 P}) \sum_{i=1}^P \boldsymbol{x}_i^t \\
&\quad \cdot \mathrm{softmax}_p(\frac{\boldsymbol{x}_i^{t\top} \boldsymbol{W}^{(s_0)} \boldsymbol{x}_p^t}{d}) \cdot (\boldsymbol{x}_i^t - \sum_{r=1}^P \mathrm{softmax}_p(\frac{\boldsymbol{x}_r^{t\top} \boldsymbol{W}^{(s_0)} \boldsymbol{x}_p^t}{d}) \boldsymbol{x}_r^t) \boldsymbol{x}_p^{t\top} \Big] \tilde{\boldsymbol{x}}_{j'}
\end{aligned}
\tag{104}
$$

$$
\begin{aligned}
&(-\tilde{\boldsymbol{x}}_k^\top) \frac{1}{T} \sum_{t=1}^T \mathbb{E}_{\boldsymbol{E}^t, \boldsymbol{X}^0} \Big[ (\boldsymbol{f}(\boldsymbol{W}^{(s_0)}; \sqrt{\bar{\alpha}_t} \boldsymbol{X}^0 + \sqrt{1-\bar{\alpha}_t} \boldsymbol{E}^t, t)_p - \boldsymbol{\epsilon}_p)^\top (-\frac{v_t^{(s_0)}}{d^2 P}) \sum_{i=1}^P \boldsymbol{x}_i^t \\
&\quad \cdot \mathrm{softmax}_p(\frac{\boldsymbol{x}_i^{t\top} \boldsymbol{W}^{(s_0)} \boldsymbol{x}_p^t}{d}) \cdot (\boldsymbol{x}_i^t - \sum_{r=1}^P \mathrm{softmax}_p(\frac{\boldsymbol{x}_r^{t\top} \boldsymbol{W}^{(s_0)} \boldsymbol{x}_p^t}{d}) \boldsymbol{x}_r^t) \boldsymbol{x}_p^{t\top} \Big] \tilde{\boldsymbol{x}}_{j'} \\
\leq & \frac{\log d}{\sqrt{d}} \cdot (-\tilde{\boldsymbol{x}}_j^\top) \frac{1}{T} \sum_{t=1}^T \mathbb{E}_{\boldsymbol{E}^t, \boldsymbol{X}^0} \Big[ (\boldsymbol{f}(\boldsymbol{W}^{(s_0)}; \sqrt{\bar{\alpha}_t} \boldsymbol{X}^0 + \sqrt{1-\bar{\alpha}_t} \boldsymbol{E}^t, t)_p - \boldsymbol{\epsilon}_p)^\top (-\frac{v_t^{(s_0)}}{d^2 P}) \sum_{i=1}^P \boldsymbol{x}_i^t \\
&\quad \cdot \mathrm{softmax}_p(\frac{\boldsymbol{x}_i^{t\top} \boldsymbol{W}^{(s_0)} \boldsymbol{x}_p^t}{d}) \cdot (\boldsymbol{x}_i^t - \sum_{r=1}^P \mathrm{softmax}_p(\frac{\boldsymbol{x}_r^{t\top} \boldsymbol{W}^{(s_0)} \boldsymbol{x}_p^t}{d}) \boldsymbol{x}_r^t) \boldsymbol{x}_p^{t\top} \Big] \tilde{\boldsymbol{x}}_{j'}.
\end{aligned}
\tag{105}
$$

Since

$$\boldsymbol{W}^{(s_0+1)} = \boldsymbol{W}^{(s_0)} - \eta \frac{1}{T} \sum_{t=1}^T \sum_{q=1}^{s_0} \mathbb{E}_{\boldsymbol{E}^t, \boldsymbol{X}^0, t} \Big[ \frac{\partial L(\Psi^{(s_0)}; \boldsymbol{X}^t)}{\partial \boldsymbol{W}} \Big], \tag{106}$$

where $\frac{1}{T} \sum_{t=1}^T \mathbb{E}_{\boldsymbol{E}^t, \boldsymbol{X}^0, t} \Big[ \frac{\partial L(\Psi^{(s_0)}; \boldsymbol{X}^t)}{\partial \boldsymbol{W}} \Big]$ is not a function of any noise term, we have

$$
\begin{aligned}
&\Big\| \tilde{\boldsymbol{x}}_j^{t\top} \frac{1}{T} \sum_{t=1}^T \mathbb{E}_{\boldsymbol{E}^t, \boldsymbol{X}^0, t} \Big[ \frac{\partial L(\Psi^{(s_0)}; \boldsymbol{X}^t)}{\partial \boldsymbol{W}} \Big] \tilde{\boldsymbol{x}}_p^t - \frac{1}{T} \sum_{t=1}^T \mathbb{E}_{\tilde{\boldsymbol{X}}^0} \Big[ \tilde{\boldsymbol{x}}_j^{t\top} \mathbb{E}_{\boldsymbol{E}^t, \boldsymbol{X}^0} \Big[ \frac{\partial L(\Psi^{(s_0)})}{\partial \boldsymbol{W}} \Big] \tilde{\boldsymbol{x}}_p^t \Big] \Big\| \\
= & \Big\| \tilde{\boldsymbol{x}}_j^{t\top} \frac{1}{T} \sum_{t=1}^T \mathbb{E}_{\boldsymbol{E}^t, \boldsymbol{X}^0} \Big[ \frac{\partial L(\Psi^{(s_0)})}{\partial \boldsymbol{W}} \Big] \tilde{\boldsymbol{x}}_p^t - \frac{1}{T} \sum_{t=1}^T \mathbb{E}_{\tilde{\boldsymbol{X}}^0} \Big[ \boldsymbol{\mu}_u^\top \mathbb{E}_{\boldsymbol{E}^t, \boldsymbol{X}^0} \Big[ \frac{\partial L(\Psi^{(s_0)})}{\partial \boldsymbol{W}} \Big] \boldsymbol{\mu}_u \Big] \Big\| \\
= & \Big\| \boldsymbol{a}_1 \frac{1}{T} \sum_{t=1}^T \mathbb{E}_{\boldsymbol{E}^t, \boldsymbol{X}^0} \Big[ \frac{\partial L(\Psi^{(s_0)})}{\partial \boldsymbol{W}} \Big] \boldsymbol{a}_2 + \frac{1}{T} \sum_{t=1}^T \boldsymbol{a}_1^\top \mathbb{E}_{\boldsymbol{E}^t, \boldsymbol{X}^0} \Big[ \frac{\partial L(\Psi^{(s_0)})}{\partial \boldsymbol{W}} \Big] \boldsymbol{\mu}_u \\
& + \frac{1}{T} \sum_{t=1}^T \boldsymbol{\mu}_u^\top \mathbb{E}_{\boldsymbol{E}^t, \boldsymbol{X}^0} \Big[ \frac{\partial L(\Psi^{(s_0)})}{\partial \boldsymbol{W}} \Big] \boldsymbol{a}_2 \Big\|,
\end{aligned}
\tag{107}
$$

where $\boldsymbol{a}_1 = \tilde{\boldsymbol{x}}_j^t - \sqrt{\bar{\alpha}_t}\boldsymbol{\mu}_u$, $\boldsymbol{a}_2 = \tilde{\boldsymbol{x}}_p^t - \sqrt{\bar{\alpha}_t}\boldsymbol{\mu}_u$. We can obtain $\boldsymbol{a}_1, \boldsymbol{a}_2 \sim \mathcal{N}(0, \bar{\alpha}_t\rho^2 + 1 - \bar{\alpha}_t)$. We then have the following discussion on $\boldsymbol{x}_i^t$ in the gradient (79). $\boldsymbol{a}_1^\top(\boldsymbol{x}_i^t - \sum_{r=1}^P \text{softmax}_p(\frac{\boldsymbol{x}_r^{t\top}\boldsymbol{W}^{(0)}\boldsymbol{x}_p^t}{d})\boldsymbol{x}_r^t)\boldsymbol{x}_p^{t\top}\boldsymbol{a}_2$ is in the order of $d \log d$. $\boldsymbol{a}_1^\top(\boldsymbol{x}_i^t - \sum_{r=1}^P \text{softmax}_p(\frac{\boldsymbol{x}_r^{t\top}\boldsymbol{W}^{(0)}\boldsymbol{x}_p^t}{d})\boldsymbol{x}_r^t)\boldsymbol{x}_p^{t\top}\boldsymbol{\mu}_u$ is in the order of $d^{\frac{3}{2}} \log d$. $\boldsymbol{\mu}_u^\top(\boldsymbol{x}_i^t - \sum_{r=1}^P \text{softmax}_p(\frac{\boldsymbol{x}_r^{t\top}\boldsymbol{W}^{(0)}\boldsymbol{x}_p^t}{d})\boldsymbol{x}_r^t)\boldsymbol{x}_p^{t\top}\boldsymbol{a}_2$ is in the order of $d^{\frac{3}{2}} \log d$. $(\boldsymbol{f}(\boldsymbol{W}; \sqrt{\bar{\alpha}_t}\boldsymbol{X}^0 + \sqrt{1 - \bar{\alpha}_t}\boldsymbol{E}^t, t)_p - \boldsymbol{\epsilon}_p)^\top\boldsymbol{x}_i^t$ is in the order of $-d$ if $d$ is large enough, and the order of positive term is no more than $\sqrt{d} \log d$. Therefore,

$$\left\|\frac{1}{T}\sum_{t=1}^T \boldsymbol{a}_1 \mathbb{E}_{\boldsymbol{E}^t,\boldsymbol{X}^0}[\frac{\partial L(\Psi^{(s_0)})}{\partial \boldsymbol{W}}]\boldsymbol{a}_2\right\| \lesssim \frac{\log d}{d^{\frac{3}{2}}} \cdot \frac{1}{T}\sum_{t=1}^T \mathbb{E}_{\tilde{\boldsymbol{X}}^0}[\tilde{\boldsymbol{x}}_j^{t\top}\mathbb{E}_{\boldsymbol{E}^t,\boldsymbol{X}^0,t}[\frac{\partial L(\Psi^{(s_0)})}{\partial \boldsymbol{W}}]\tilde{\boldsymbol{x}}_p^t], \tag{108}$$

$$\left\|\frac{1}{T}\sum_{t=1}^T \boldsymbol{a}_1 \mathbb{E}_{\boldsymbol{E}^t,\boldsymbol{X}^0,t}[\frac{\partial L(\Psi^{(s_0)})}{\partial \boldsymbol{W}}]\boldsymbol{\mu}_U\right\| \lesssim \frac{\log d}{d} \cdot \frac{1}{T}\sum_{t=1}^T \mathbb{E}_{\tilde{\boldsymbol{X}}^0}[\tilde{\boldsymbol{x}}_j^{t\top}\mathbb{E}_{\boldsymbol{E}^t,\boldsymbol{X}^0,t}[\frac{\partial L(\Psi^{(s_0)})}{\partial \boldsymbol{W}}]\tilde{\boldsymbol{x}}_p^t], \tag{109}$$

$$\left\|\frac{1}{T}\sum_{t=1}^T \boldsymbol{\mu}_u \mathbb{E}_{\boldsymbol{E}^t,\boldsymbol{X}^0,t}[\frac{\partial L(\Psi^{(s_0)})}{\partial \boldsymbol{W}}]\boldsymbol{a}_2\right\| \lesssim \frac{\log d}{d} \cdot \frac{1}{T}\sum_{t=1}^T \mathbb{E}_{\tilde{\boldsymbol{X}}^0}[\tilde{\boldsymbol{x}}_j^{t\top}\mathbb{E}_{\boldsymbol{E}^t,\boldsymbol{X}^0,t}[\frac{\partial L(\Psi^{(s_0)})}{\partial \boldsymbol{W}}]\tilde{\boldsymbol{x}}_p^t], \tag{110}$$

for any $Y_j = Y_p = u$. Hence, we have that for any $u \in [M]$ and $Y_j = Y_p = u$,

$$\left\|\frac{1}{T}\sum_{t=1}^T \tilde{\boldsymbol{x}}_j^{t\top}\mathbb{E}_{\boldsymbol{E}^t,\boldsymbol{X}^0,t}[\frac{\partial L(\Psi^{(s_0)})}{\partial \boldsymbol{W}}]\tilde{\boldsymbol{x}}_p^t - \frac{1}{T}\sum_{t=1}^T \mathbb{E}_{\tilde{\boldsymbol{X}}^0}[\tilde{\boldsymbol{x}}_j^{t\top}\mathbb{E}_{\boldsymbol{E}^t,\boldsymbol{X}^0,t}[\frac{\partial L(\Psi^{(s_0)})}{\partial \boldsymbol{W}}]\tilde{\boldsymbol{x}}_p^t]\right\|$$
$$\lesssim \frac{\log d}{d} \cdot \frac{1}{T}\sum_{t=1}^T \mathbb{E}_{\tilde{\boldsymbol{X}}^0}[\tilde{\boldsymbol{x}}_j^{t\top}\mathbb{E}_{\boldsymbol{E}^t,\boldsymbol{X}^0,t}[\frac{\partial L(\Psi^{(s_0)})}{\partial \boldsymbol{W}}]\tilde{\boldsymbol{x}}_p^t]. \tag{111}$$

We can also derive

$$\left\|\tilde{\boldsymbol{x}}_j^{t\top}\boldsymbol{W}^{(0)}\tilde{\boldsymbol{x}}_p^t - \mathbb{E}_{\tilde{\boldsymbol{X}}^0}[\tilde{\boldsymbol{x}}_j^{t\top}\boldsymbol{W}^{(0)}\tilde{\boldsymbol{x}}_p^t]\right\| \lesssim \frac{\log d}{d} \cdot \mathbb{E}_{\tilde{\boldsymbol{X}}^0}[\tilde{\boldsymbol{x}}_j^{t\top}\boldsymbol{W}^{(0)}\tilde{\boldsymbol{x}}_p^t], \tag{112}$$

by replacing $\mathbb{E}_{\boldsymbol{E}^t,\boldsymbol{X}^0}[\frac{\partial L(\Psi^{(s_0)})}{\partial \boldsymbol{W}}]$ with an arbitrarily initialized $\boldsymbol{W}^{(0)}$. Thus, we have We can also derive

$$\left\|\tilde{\boldsymbol{x}}_j^{t\top}\boldsymbol{W}^{(s_0+1)}\tilde{\boldsymbol{x}}_p^t - \mathbb{E}_{\tilde{\boldsymbol{X}}^0}[\tilde{\boldsymbol{x}}_j^{t\top}\boldsymbol{W}^{(s_0+1)}\tilde{\boldsymbol{x}}_p^t]\right\| \lesssim \frac{\log d}{d} \cdot \mathbb{E}_{\tilde{\boldsymbol{X}}^0}[\tilde{\boldsymbol{x}}_j^{t\top}\boldsymbol{W}^{(s_0+1)}\tilde{\boldsymbol{x}}_p^t]. \tag{113}$$

This ensures that for any $u \in [M]$ and $Y_i = Y_p = Y_j = u$,

$$\|\text{softmax}_p(\frac{\boldsymbol{x}_i^{t\top}\boldsymbol{W}^{(s_0+1)}\boldsymbol{x}_p^t}{d}) - \mathbb{E}[\text{softmax}_p(\frac{\boldsymbol{x}_i^{t\top}\boldsymbol{W}^{(s_0+1)}\boldsymbol{x}_p^t}{d})]\|$$
$$\lesssim \mathbb{E}[\text{softmax}_p(\frac{\boldsymbol{x}_i^{t\top}\boldsymbol{W}^{(s_0+1)}\boldsymbol{x}_p^t}{d})] \cdot \frac{\log d \log \epsilon^{-1}(\min_{u \in [M]}\{\min_{u \in [M]}(\nu_u^{\tilde{\pi}}(K))\}^{-1} - 1)^2}{d}. \tag{114}$$

Hence,

$$\left|\text{softmax}_p(\frac{\boldsymbol{x}_i^{t\top}\boldsymbol{W}^{(s_0+1)}\boldsymbol{x}_p^t}{d}) - \text{softmax}_p(\frac{\boldsymbol{x}_j^{t\top}\boldsymbol{W}^{(s_0+1)}\boldsymbol{x}_p^t}{d})\right| \lesssim \frac{\log d \log \epsilon^{-1}((\min_{u \in [M]}\nu_u^{\tilde{\pi}}(K))^{-1} - 1)^2}{d\sum_{q=1}^P \mathbb{1}[Y_q = u]}. \tag{115}$$

Therefore, by summing up over (101), we can obtain that for $\boldsymbol{x}_j^t$ and $\boldsymbol{x}_{j'}^t$ with $\boldsymbol{\mu}_u$ as the mean,

$$\frac{1}{T}\sum_{t=1}^T (-\tilde{\boldsymbol{x}}_j^{t\top})\mathbb{E}_{\boldsymbol{E}^t,\boldsymbol{X}^0}\left[\frac{\partial L(\Psi^{(s_0)})}{\partial \boldsymbol{W}}\right]\tilde{\boldsymbol{x}}_{j'}^t$$
$$= \frac{1}{T}\sum_{t=1}^T (-\tilde{\boldsymbol{x}}_j^{t\top})\mathbb{E}_{\boldsymbol{E}^t,\boldsymbol{X}^0}\left[\sum_{p=1}^P (\boldsymbol{f}(\boldsymbol{W}^{(s_0)}; \sqrt{\bar{\alpha}_t}\boldsymbol{X}^0 + \sqrt{1 - \bar{\alpha}_t}\boldsymbol{E}^t, t)_p - \boldsymbol{\epsilon}_p)^\top(-\frac{v_t^{(s_0)}}{d^2 P})\sum_{i=1}^P \boldsymbol{x}_i^t\right.$$
$$\left. \cdot \text{softmax}_p(\frac{\boldsymbol{x}_i^{t\top}\boldsymbol{W}^{(s_0)}\boldsymbol{x}_p^t}{d}) \cdot (\boldsymbol{x}_i^t - \sum_{r=1}^P \text{softmax}_p(\frac{\boldsymbol{x}_r^{t\top}\boldsymbol{W}^{(s_0)}\boldsymbol{x}_p^t}{d})\boldsymbol{x}_r^t)\boldsymbol{x}_p^{t\top}\right]\tilde{\boldsymbol{x}}_{j'}^t \tag{116}$$
$$\gtrsim \frac{1}{T}\sum_{t=1}^T \mathbb{E}_{\boldsymbol{E}^t,\boldsymbol{X}^0}[(v_t^{(s_0)})^2\bar{\alpha}_t^3 d(1 - \sum_{l=1}^P \zeta_{l,p,t}^u(s_0))^2 \sum_{i=1}^P \zeta_{i,p,t}^u(s_0)] \cdot \nu_u^{\tilde{\pi}}(K),$$

where the last step holds with a high probability if

$$P \cdot \nu_u^{\tilde{\pi}}(K) \gtrsim \log dK. \tag{117}$$

Similarly, we hvae that for $\boldsymbol{\mu}_u \neq \boldsymbol{\mu}_{u'}$ and $\tilde{\boldsymbol{x}}_j$, $\tilde{\boldsymbol{x}}_{j'}$ with the mean of $\boldsymbol{\mu}_u$ and $\tilde{\boldsymbol{x}}_k$ with the mean of $\boldsymbol{\mu}_{u'}$, we have that

$$
\begin{aligned}
&(-\tilde{\boldsymbol{x}}_k^\top)\mathbb{E}_{\boldsymbol{E}^t, \boldsymbol{X}^0}\Big[\frac{\partial L(\Psi^{(s_0)})}{\partial \boldsymbol{W}}\Big]\tilde{\boldsymbol{x}}_{j'}^t \\
&=(-\tilde{\boldsymbol{x}}_k^\top)\mathbb{E}_{\boldsymbol{E}^t, \boldsymbol{X}^0}\Big[\sum_{p=1}^P(\boldsymbol{f}(\boldsymbol{W}^{(s_0)}; \sqrt{\bar{\alpha}_t}\boldsymbol{X}^0 + \sqrt{1-\bar{\alpha}_t}\boldsymbol{E}^t, t)_p - \boldsymbol{\epsilon}_p)^\top(-\frac{v_t^{(s_0)}}{d^2 P})\sum_{i=1}^P \boldsymbol{x}_i^t \\
&\qquad \cdot \mathrm{softmax}_p(\frac{\boldsymbol{x}_i^{t\top}\boldsymbol{W}^{(s_0)}\boldsymbol{x}_p^t}{d}) \cdot (\boldsymbol{x}_i^t - \sum_{r=1}^P \mathrm{softmax}_p(\frac{\boldsymbol{x}_r^{t\top}\boldsymbol{W}^{(s_0)}\boldsymbol{x}_p^t}{d})\boldsymbol{x}_r^t)\boldsymbol{x}_p^{t\top}\Big]\tilde{\boldsymbol{x}}_{j'}^t \\
&\leq \frac{\log d}{\sqrt{d}} \cdot (-\tilde{\boldsymbol{x}}_j^\top)\mathbb{E}_{\boldsymbol{E}^t, \boldsymbol{X}^0}\Big[\sum_{p=1}^P(\boldsymbol{f}(\boldsymbol{W}^{(s_0)}; \sqrt{\bar{\alpha}_t}\boldsymbol{X}^0 + \sqrt{1-\bar{\alpha}_t}\boldsymbol{E}^t, t)_p - \boldsymbol{\epsilon}_p)^\top(-\frac{v_t^{(s_0)}}{d^2 P})\sum_{i=1}^P \boldsymbol{x}_i^t \\
&\qquad \cdot \mathrm{softmax}_p(\frac{\boldsymbol{x}_i^{t\top}\boldsymbol{W}^{(s_0)}\boldsymbol{x}_p^t}{d}) \cdot (\boldsymbol{x}_i^t - \sum_{r=1}^P \mathrm{softmax}_p(\frac{\boldsymbol{x}_r^{t\top}\boldsymbol{W}^{(s_0)}\boldsymbol{x}_p^t}{d})\boldsymbol{x}_r^t)\boldsymbol{x}_p^{t\top}\Big]\tilde{\boldsymbol{x}}_{j'}^t \\
&= \frac{\log d}{\sqrt{d}} \cdot (-\tilde{\boldsymbol{x}}_j^\top)\mathbb{E}_{\boldsymbol{E}^t, \boldsymbol{X}^0}\Big[\frac{\partial L(\Psi^{(s_0)})}{\partial \boldsymbol{W}}\Big]\tilde{\boldsymbol{x}}_{j'}^t
\end{aligned} \tag{118}
$$

$$
\begin{aligned}
&\frac{1}{T}\sum_{t=1}^T(-\tilde{\boldsymbol{x}}_k^\top)\mathbb{E}_{\boldsymbol{E}^t, \boldsymbol{X}^0}\Big[\frac{\partial L(\Psi^{(s_0)})}{\partial \boldsymbol{W}}\Big]\tilde{\boldsymbol{x}}_{j'}^t \\
&=\frac{1}{T}\sum_{t=1}^T(-\tilde{\boldsymbol{x}}_k^\top)\mathbb{E}_{\boldsymbol{E}^t, \boldsymbol{X}^0}\Big[\sum_{p=1}^P(\boldsymbol{f}(\boldsymbol{W}^{(s_0)}; \sqrt{\bar{\alpha}_t}\boldsymbol{X}^0 + \sqrt{1-\bar{\alpha}_t}\boldsymbol{E}^t, t)_p - \boldsymbol{\epsilon}_p)^\top(-\frac{v_t^{(s_0)}}{d^2 P})\sum_{i=1}^P \boldsymbol{x}_i^t \\
&\qquad \cdot \mathrm{softmax}_p(\frac{\boldsymbol{x}_i^{t\top}\boldsymbol{W}^{(s_0)}\boldsymbol{x}_p^t}{d}) \cdot (\boldsymbol{x}_i^t - \sum_{r=1}^P \mathrm{softmax}_p(\frac{\boldsymbol{x}_r^{t\top}\boldsymbol{W}^{(s_0)}\boldsymbol{x}_p^t}{d})\boldsymbol{x}_r^t)\boldsymbol{x}_p^{t\top}\Big]\tilde{\boldsymbol{x}}_{j'} \\
&\leq \frac{1}{T}\sum_{t=1}^T(-\tilde{\boldsymbol{x}}_j^\top)\frac{\log d}{\sqrt{d}} \cdot \mathbb{E}_{\boldsymbol{E}^t, \boldsymbol{X}^0}\Big[\sum_{p=1}^P(\boldsymbol{f}(\boldsymbol{W}^{(s_0)}; \sqrt{\bar{\alpha}_t}\boldsymbol{X}^0 + \sqrt{1-\bar{\alpha}_t}\boldsymbol{E}^t, t)_p - \boldsymbol{\epsilon}_p)^\top(-\frac{v_t^{(s_0)}}{d^2 P})\sum_{i=1}^P \boldsymbol{x}_i^t \\
&\qquad \cdot \mathrm{softmax}_p(\frac{\boldsymbol{x}_i^{t\top}\boldsymbol{W}^{(s_0)}\boldsymbol{x}_p^t}{d}) \cdot (\boldsymbol{x}_i^t - \sum_{r=1}^P \mathrm{softmax}_p(\frac{\boldsymbol{x}_r^{t\top}\boldsymbol{W}^{(s_0)}\boldsymbol{x}_p^t}{d})\boldsymbol{x}_r^t)\boldsymbol{x}_p^{t\top}\Big]\tilde{\boldsymbol{x}}_{j'} \\
&= \frac{\log d}{\sqrt{d}} \cdot \frac{1}{T}\sum_{t=1}^T(-\tilde{\boldsymbol{x}}_j^\top)\mathbb{E}_{\boldsymbol{E}^t, \boldsymbol{X}^0}\Big[\frac{\partial L(\Psi^{(s_0)})}{\partial \boldsymbol{W}}\Big]\tilde{\boldsymbol{x}}_{j'}^t.
\end{aligned} \tag{119}
$$

$\square$

### D.2. Proof of Lemma 4

*Proof.* For any $\tilde{\boldsymbol{x}}_j$ and $\tilde{\boldsymbol{x}}_{j'}$ with $\boldsymbol{\mu}_u$ as the mean, where $\tilde{\boldsymbol{X}}$ is generated by Definition, denote

$$
\begin{aligned}
Z_{j,j'}^u(s) &= \frac{\tilde{\boldsymbol{x}}_j^\top \boldsymbol{W}^{(s)}\tilde{\boldsymbol{x}}_{j'}}{d} \\
&= \frac{\tilde{\boldsymbol{x}}_j^\top \boldsymbol{W}^{(0)}\tilde{\boldsymbol{x}}_{j'}}{d} - \eta\frac{1}{T}\sum_{t=1}^T\sum_{s_0=0}^{s-1}\sum_{p=1}^P \mathbb{E}_{\boldsymbol{E}^t, \boldsymbol{X}^0}\Big[-\tilde{\boldsymbol{x}}_j^\top(\boldsymbol{f}(\boldsymbol{W}^{(s_0)}; \sqrt{\bar{\alpha}_t}\boldsymbol{X}^0 + \sqrt{1-\bar{\alpha}_t}\boldsymbol{E}^t, t)_p - \boldsymbol{\epsilon}_p)^\top \\
&\qquad \cdot (-\frac{v_t^{(s)}}{d^3 P})\sum_{i=1}^P \boldsymbol{x}_i^t\mathrm{softmax}_p(\frac{\boldsymbol{x}_i^{t\top}\boldsymbol{W}^{(s_0)}\boldsymbol{x}_p^t}{d}) \cdot (\boldsymbol{x}_i^t - \sum_{r=1}^P \mathrm{softmax}_p(\frac{\boldsymbol{x}_r^{t\top}\boldsymbol{W}^{(s_0)}\boldsymbol{x}_p^t}{d})\boldsymbol{x}_r^t)\boldsymbol{x}_p^{t\top}\tilde{\boldsymbol{x}}_{j'}\Big]
\end{aligned} \tag{120}
$$

Denote $\gamma(u) = \eta \cdot \frac{1}{T} \sum_{t=1}^{T} (v_t^{(s)})^2 \bar{\alpha}_t^3 \cdot \nu_u^{\tilde{\pi}}(K)$. Note that

$$\sum_{j=1}^{P} \zeta_{j,j',t}^{u}(s) \gtrsim \frac{\sum_{p=1}^{P} \mathbb{1}[Y_p = u] \cdot e^{Z_{j,j'}^{u}(s)}}{\sum_{p=1}^{P} \mathbb{1}[Y_p = u] \cdot e^{Z_{j,j'}^{u}(s)} + \sum_{p=1}^{P} \mathbb{1}[Y_p \neq u]}. \tag{121}$$

When $Z_{j,j'}^{u}(s) \leq \Theta(\log \frac{1-\nu_u^{\tilde{\pi}}(K)}{\nu_u^{\tilde{\pi}}(K)})$, we have that by Lemma 1, with a high probability,

$$1 - \sum_{j=1}^{P} \zeta_{j,j',t}^{u}(s) \gtrsim \frac{\sum_{p=1}^{P} \mathbb{1}[Y_p \neq u]}{\sum_{p=1}^{P} \mathbb{1}[Y_p = u] \cdot e^{Z_{j,j'}^{u}(s)} + \sum_{p=1}^{P} \mathbb{1}[Y_p \neq u]} \gtrsim (1 - \nu_u^{\tilde{\pi}}(K)) e^{-Z_{j,j'}^{u}(s)}. \tag{122}$$

Otherwise,

$$1 - \sum_{j=1}^{P} \zeta_{j,j',t}^{u}(s) \gtrsim \frac{\sum_{p=1}^{P} \mathbb{1}[Y_p \neq u]}{\sum_{p=1}^{P} \mathbb{1}[Y_p = u] \cdot e^{Z_{j,j'}^{u}(s)} + \sum_{p=1}^{P} \mathbb{1}[Y_p \neq u]} \gtrsim \frac{1 - \nu_u^{\tilde{\pi}}(K)}{\nu_u^{\tilde{\pi}}(K)} \cdot e^{-Z_{j,j'}^{u}(s)}. \tag{123}$$

Note that by (25) in Lemma 3,

$$Z_{j,j'}^{u}(s) \gtrsim \eta \frac{1}{T} \sum_{t=1}^{T} \sum_{b=0}^{s-1} \mathbb{E}_{\boldsymbol{E}^t, \boldsymbol{X}^0} [\bar{\alpha}_t^3 (v_t^{(s)})^2 (1 - \sum_{l=1}^{P} \zeta_{l,p,t}^{u}(b))^2 \sum_{i=1}^{P} \zeta_{i,p,t}^{u}(b)] \cdot \nu_u^{\tilde{\pi}}(K) \tag{124}$$

We first prove that when $Z_{j,j'}^{u}(s) \leq \Theta(\log \frac{1-\nu_u^{\tilde{\pi}}(K)}{\nu_u^{\tilde{\pi}}(K)})$, we have $Z_{j,j'}^{u}(s) \geq \frac{1}{2} \log(1 + 2\gamma(u)(1 - \nu_u^{\tilde{\pi}}(K))^2 \nu_u^{\tilde{\pi}}(K) \cdot s)$ by induction. The conclusion holds when $s = 0$. Suppose that this conclusion holds when $s \leq s_0$. Then,

$$
\begin{aligned}
&Z_{j,j'}^{u}(s+1) \\
&\geq \eta \frac{1}{T} \sum_{t=1}^{T} \sum_{b=0}^{s-1} \mathbb{E}_{\boldsymbol{E}^t, \boldsymbol{X}^0} [(v_t^{(s)})^2 \bar{\alpha}_t^3 (1 - \sum_{l=1}^{P} \zeta_{l,p,t}^{u}(b))^2 \sum_{i=1}^{P} \zeta_{i,p,t}^{u}(b)] \cdot \nu_u^{\tilde{\pi}}(K) \\
&\quad + \frac{1}{T} \sum_{t=1}^{T} \mathbb{E}_{\boldsymbol{E}^t, \boldsymbol{X}^0} [(v_t^{(s)})^2 \bar{\alpha}_t^3 (1 - \sum_{l=1}^{P} \zeta_{l,p,t}^{u}(s))^2 \sum_{i=1}^{P} \zeta_{i,p,t}^{u}(s)] \cdot \nu_u^{\tilde{\pi}}(K) \\
&\geq \frac{1}{2} \log(1 + 2\gamma(u)(1 - \nu_u^{\tilde{\pi}}(K))^2 \nu_u^{\tilde{\pi}}(K) s) + \frac{1}{T} \sum_{t=1}^{T} \mathbb{E}_{\boldsymbol{E}^t, \boldsymbol{X}^0} [(v_t^{(s)})^2 \bar{\alpha}_t^3 (1 - \sum_{l=1}^{P} \zeta_{l,p,t}^{u}(s))^2 \sum_{i=1}^{P} \zeta_{i,p,t}^{u}(s)] \cdot \nu_u^{\tilde{\pi}}(K) \\
&\geq \frac{1}{2} \log(1 + 2 \cdot \gamma(u)(1 - \nu_u^{\tilde{\pi}}(K))^2 \nu_u^{\tilde{\pi}}(K)(s+1)),
\end{aligned}
\tag{125}
$$

where the last step is by

$$
\begin{aligned}
&\frac{1}{T} \sum_{t=1}^{T} \mathbb{E}_{\boldsymbol{E}^t, \boldsymbol{X}^0} [(v_t^{(s)})^2 \bar{\alpha}_t^3 (1 - \sum_{l=1}^{P} \zeta_{l,p,t}^{u}(s_0))^2 \sum_{i=1}^{P} \zeta_{i,p,t}^{u}(s_0)] \cdot \nu_u^{\tilde{\pi}}(K) \\
&\geq \frac{1}{T} \sum_{t=1}^{T} \mathbb{E}_{\boldsymbol{E}^t, \boldsymbol{X}^0} [(v_t^{(s)})^2 \bar{\alpha}_t^3 \cdot e^{-2Z_{j,j'}(s)}] \cdot (1 - \nu_u^{\tilde{\pi}}(K))^2 \cdot (\nu_u^{\tilde{\pi}}(K))^2 \\
&\geq \frac{1}{T} \sum_{t=1}^{T} \mathbb{E}_{\boldsymbol{E}^t, \boldsymbol{X}^0} [(v_t^{(s)})^2 \bar{\alpha}_t^3 \cdot \frac{1}{1 + 2\gamma(u)(1 - \nu_u^{\tilde{\pi}}(K))^2 \nu_u^{\tilde{\pi}}(K)s}] \cdot (\nu_u^{\tilde{\pi}}(K))^2 (1 - \nu_u^{\tilde{\pi}}(K))^2 \\
&\geq \frac{1}{2} \log(1 + \frac{2\gamma(u)\nu_u^{\tilde{\pi}}(K)(1 - \nu_u^{\tilde{\pi}}(K))^2}{1 + 2\gamma(u)\nu_u^{\tilde{\pi}}(K)(1 - \nu_u^{\tilde{\pi}}(K))^2 \cdot s}).
\end{aligned}
\tag{126}
$$

Therefore, the conclusion holds when $s = s_0 + 1$. When $s \geq I_0(u) := \frac{1}{\gamma(u)(\nu_u^{\tilde{\pi}}(K))^3}$, we have $Z_{j,j'}^{u}(s) \geq \log \frac{1-\nu_u^{\tilde{\pi}}(K)}{\nu_u^{\tilde{\pi}}(K)}$. Then, we prove that $Z_{j,j'}^{u}(s) \geq \frac{1}{2} \log(1 + 2\gamma(u) \cdot (\frac{1-\nu_u^{\tilde{\pi}}(K)}{\nu_u^{\tilde{\pi}}(K)})^2 (s - I_0(u)) + 2\gamma(u)(1 - \nu_u^{\tilde{\pi}}(K))^2 \nu_u^{\tilde{\pi}}(K) I_0(u))$ by induction

for $Z_{j,j'}^u(s) \geq \Theta(\log \frac{1-\nu_u^{\tilde{\pi}}(K)}{\nu_u^{\tilde{\pi}}(K)})$. Suppose that this conclusion holds when $s \leq s_0$. Then,

$$
\begin{aligned}
Z_{j,j'}^u(s+1) \geq & \eta \frac{1}{T} \sum_{t=1}^T \sum_{b=0}^{s-1} \mathbb{E}_{\boldsymbol{E}^t, \boldsymbol{X}^0}[(v_t^{(s)})^2 \bar{\alpha}_t^3 (1 - \sum_{l=1}^P \zeta_{l,p,t}^u(b))^2 \sum_{i=1}^P \zeta_{i,p,t}^u(b)] \cdot \nu_u^{\tilde{\pi}}(K) \\
& + \frac{1}{T} \sum_{t=1}^T \mathbb{E}_{\boldsymbol{E}^t, \boldsymbol{X}^0}[(v_t^{(s)})^2 \bar{\alpha}_t^3 (1 - \sum_{l=1}^P \zeta_{l,p,t}^u(s))^2 \sum_{i=1}^P \zeta_{i,p,t}^u(s)] \cdot \nu_u^{\tilde{\pi}}(K) \\
\geq & \frac{1}{2} \log(1 + 2\gamma(u) \cdot (\frac{1-\nu_u^{\tilde{\pi}}(K)}{\nu_u^{\tilde{\pi}}(K)})^2 (s - I_0(u)) + 2\gamma(u)(1-\nu_u^{\tilde{\pi}}(K))^2 \nu_u^{\tilde{\pi}}(K) I_0(u)) \\
& + \frac{1}{T} \sum_{t=1}^T \mathbb{E}_{\boldsymbol{E}^t, \boldsymbol{X}^0}[(v_t^{(s)})^2 \bar{\alpha}_t^3 (1 - \sum_{l=1}^P \zeta_{l,p,t}^u(s))^2 \sum_{i=1}^P \zeta_{i,p,t}^u(s)] \nu_u^{\tilde{\pi}}(K) \\
\geq & \frac{1}{2} \log(1 + 2 \cdot (\frac{1-\nu_u^{\tilde{\pi}}(K)}{\nu_u^{\tilde{\pi}}(K)})^2 \gamma(u)(s + 1 - I_0(u)) + 2\gamma(u)(1-\nu_u^{\tilde{\pi}}(K))^2 \nu_u^{\tilde{\pi}}(K) I_0(u)),
\end{aligned}
\tag{127}
$$

where the last step is by

$$
\begin{aligned}
& \frac{1}{T} \sum_{t=1}^T \mathbb{E}_{\boldsymbol{E}^t, \boldsymbol{X}^0}[(v_t^{(s)})^2 \bar{\alpha}_t^3 (1 - \sum_{l=1}^P \zeta_{l,p,t}^u(s))^2 \sum_{i=1}^P \zeta_{i,p,t}^u(s)] \cdot \nu_u^{\tilde{\pi}}(K) \\
\geq & \frac{1}{T} \sum_{t=1}^T \mathbb{E}_{\boldsymbol{E}^t, \boldsymbol{X}^0}[(v_t^{(s)})^2 \bar{\alpha}_t^3 e^{-2Z_{j,j}^u(s)}] \cdot \nu_u^{\tilde{\pi}}(K) \cdot (\frac{1-\nu_u^{\tilde{\pi}}(K)}{\nu_u^{\tilde{\pi}}(K)})^2 \\
\geq & \frac{\gamma(u) \cdot \frac{(1-\nu_u^{\tilde{\pi}}(K))^2}{(\nu_u^{\tilde{\pi}}(K))^2}}{1 + 2\gamma(u) \cdot \frac{(1-\nu_u^{\tilde{\pi}}(K))^2}{(\nu_u^{\tilde{\pi}}(K))^2}(s - I_0(u)) + 2\gamma(u)(1-\nu_u^{\tilde{\pi}}(K))^2 \nu_u^{\tilde{\pi}}(K) I_0(u)} \\
\geq & \frac{1}{2} \log(1 + \frac{2\gamma(u) \cdot \frac{(1-\nu_u^{\tilde{\pi}}(K))^2}{(\nu_u^{\tilde{\pi}}(K))^2}}{1 + 2\gamma(u) \cdot \frac{(1-\nu_u^{\tilde{\pi}}(K))^2}{(\nu_u^{\tilde{\pi}}(K))^2}(s - I_0(u)) + 2\gamma(u)(1-\nu_u^{\tilde{\pi}}(K))^2 \nu_u^{\tilde{\pi}}(K) I_0(u)}).
\end{aligned}
\tag{128}
$$

Then,

$$
\begin{aligned}
Z_{j,j'}(\gamma(u)^{-1}\epsilon^{-1} + I_0(u)) \geq & \frac{1}{2} \log(\gamma(u)(\frac{(1-\nu_u^{\tilde{\pi}}(K))}{\nu_u^{\tilde{\pi}}(K)})^2 \gamma(u)^{-1}\epsilon^{-1} + (\frac{1-\nu_u^{\tilde{\pi}}(K)}{\nu_u^{\tilde{\pi}}(K)})^2) \\
\gtrsim & \frac{1}{2} \log((\frac{(1-\nu_u^{\tilde{\pi}}(K))}{\nu_u^{\tilde{\pi}}(K)})^2 \epsilon^{-1}),
\end{aligned}
\tag{129}
$$

where the second step holds if $\epsilon \leq 1$. Let $u^* = \arg\min_{u \in [M]} \{\nu_u^{\tilde{\pi}}(K)\}$. We have

$$
\begin{aligned}
& Z_{j,j'}(\gamma(u^*)^{-1}\epsilon^{-1} + I_0(u^*)) \\
\gtrsim & \frac{1}{2} \log((\frac{1-\nu_u^{\tilde{\pi}}(K)}{\nu_u^{\tilde{\pi}}(K)})^2 \epsilon^{-1} + \gamma(u)(\frac{1-\nu_u^{\tilde{\pi}}(K)}{\nu_u^{\tilde{\pi}}(K)})^2 (I_0(u^*) - I_0(u) + \epsilon^{-1}(\gamma(u^*)^{-1} - \gamma(u)^{-1}))).
\end{aligned}
\tag{130}
$$

Hence, as long as

$$
\gamma(u)(\frac{1-\nu_u^{\tilde{\pi}}(K)}{\nu_u^{\tilde{\pi}}(K)})^2 \epsilon^{-1}(\gamma(u^*)^{-1} - \gamma(u)^{-1}) \lesssim \text{poly}((\frac{(1-\nu_u^{\tilde{\pi}}(K))}{\nu_u^{\tilde{\pi}}(K)})^2 \epsilon^{-1}),
\tag{131}
$$

and

$$
\gamma(u)(\frac{1-\nu_u^{\tilde{\pi}}(K)}{\nu_u^{\tilde{\pi}}(K)})^2 (I_0(u^*) - I_0(u)) \lesssim \text{poly}((\frac{(1-\nu_u^{\tilde{\pi}}(K))}{\nu_u^{\tilde{\pi}}(K)})^2 \epsilon^{-1}),
\tag{132}
$$

which hold if

$$
\epsilon \leq \delta^{\Theta(1)},
\tag{133}
$$

we have

$$Z_{j,j'}(\epsilon^{-1}\gamma(u)^{-1} + I_0(u)) = \Theta(Z_{j,j'}(\epsilon^{-1}\gamma(u^*)^{-1} + I_0(u^*))). \tag{134}$$

We then have

$$\begin{aligned}
&\left\| \frac{\partial L(\Psi^{(s)}; \boldsymbol{X}^t, t)}{\partial \boldsymbol{W}^{(s)}} \Big|_{s=I} \right\| \\
&= \max_{\boldsymbol{x}\neq 0} \frac{\left\| \frac{\partial L(\Psi^{(s)}; \boldsymbol{X}^t, t)}{\partial \boldsymbol{W}^{(s)}} \Big|_{s=I} \boldsymbol{x} \right\|}{\|\boldsymbol{x}\|} \\
&\leq \max_{u\in[M]} \left\| \boldsymbol{\mu}_u \frac{\partial L(\Psi^{(s)}; \boldsymbol{X}^t, t)}{\partial \boldsymbol{W}^{(s)}} \Big|_{s=I} \boldsymbol{\mu}_u \right\| \cdot \frac{1}{d} \\
&\lesssim \gamma(u) \cdot \left( \frac{1 - \nu_u^{\tilde{\boldsymbol{\pi}}}(K)}{\nu_u^{\tilde{\boldsymbol{\pi}}}(K)} \right)^2 \cdot e^{-2Z_{j,j}^u(s)} \\
&\leq \gamma(u)\epsilon^2,
\end{aligned} \tag{135}$$

as long as

$$\begin{aligned}
s &\gtrsim \gamma(u)^{-1}\epsilon^{-1} + I_0(u) \\
&\gtrsim (\epsilon^{-1} + \nu_u^{\tilde{\boldsymbol{\pi}}}(K)^{-3})\eta^{-1}\nu_u^{\tilde{\boldsymbol{\pi}}}(K)^{-1}\mathbb{E}_t[\bar{\alpha}_t/(1-\bar{\alpha}_t)]^{-3}(1-\alpha_1)^{-3} \\
&\gtrsim (\epsilon^{-1} + \nu_u^{\tilde{\boldsymbol{\pi}}}(K)^{-3})\eta^{-1}\nu_u^{\tilde{\boldsymbol{\pi}}}(K)^{-1}\text{SNR}^{-3}(1-\alpha_1)^{-3},
\end{aligned} \tag{136}$$

since $\bar{\alpha}_t^3 \geq (1-\alpha_1)^3\bar{\alpha}_t^3/(1-\bar{\alpha}_t)^3$. By Jensen's inequality, we have

$$\frac{1}{T}\sum_{t=1}^T \frac{\bar{\alpha}_t^3}{(1-\bar{\alpha}_t)^3} \geq \left( \frac{1}{T}\sum_{t=1}^T \frac{\bar{\alpha}_t}{(1-\bar{\alpha}_t)} \right)^3. \tag{137}$$

$\frac{1}{T}\sum_{t=1}^T \frac{\bar{\alpha}_t}{(1-\bar{\alpha}_t)}$ can be approximated by SNR, which comes from Hoeffding's inequality (24), i.e., with a high probability,

$$\left| \frac{1}{T}\sum_{t=1}^T \frac{\bar{\alpha}_t}{1-\bar{\alpha}_t} - \text{SNR} \right| \leq \frac{\alpha_1}{1-\alpha_1}\sqrt{\frac{\log d}{T}} \tag{138}$$

Therefore, when $T \gtrsim \log d$, the required condition for the number of iterations is

$$s \gtrsim I_1 := (\epsilon^{-1} + \min_{u\in[M]}\{\nu_u^{\tilde{\boldsymbol{\pi}}}(K)\}^{-3})\eta^{-1}\min_{u\in[M]}\{\nu_u^{\tilde{\boldsymbol{\pi}}}(K)\}^{-1}\text{SNR}^{-3}. \tag{139}$$

When $s \gtrsim I_1$, we have

$$1 - \sum_{j=1}^P \zeta_{j,j',t}^u(s) \gtrsim \sqrt{\epsilon}. \tag{140}$$

$\square$

### D.3. Proof of Lemma 5

*Proof.* First, for any $u \in [M]$ and $Y_p = u$, by (115), we know that for $i$, $i'$, such that $Y_i = Y_{i'} = u$, we have $\zeta_{i,p,t}^u(s) = (1 + \log \epsilon^{-1}(\min_{u\in[M]}\mathbb{E}[\pi_u])^{-1} - 1)^2/d) \cdot \zeta_{i',p,t}^u(s)$ for any $s \leq I$. Then, by Hoeffding's inequality (24),

$$\begin{aligned}
&\left\| \sum_{i=1}^P \mathbb{1}[Y_i = u]\zeta_{i,p,t}^u(s)(\boldsymbol{x}_i^t - \sqrt{\bar{\alpha}_t}\boldsymbol{\mu}_u) \right\| \\
&= \left\| \sum_{i=1}^P \mathbb{1}[Y_i = u]\zeta_{i,p,t}^u(s)(\boldsymbol{x}_i^t - \sqrt{\bar{\alpha}_t}\boldsymbol{\mu}_u) - \mathbb{E}[\sum_{i=1}^P \mathbb{1}[Y_i = u]\zeta_{i,p,t}^u(s)(\boldsymbol{x}_i^t - \sqrt{\bar{\alpha}_t}\boldsymbol{\mu}_u)] \right\| \\
&\leq \left\| \sum_{i=1}^P \mathbb{1}[Y_i = u]\frac{(1 + \frac{\log d\log\epsilon^{-1}(\min_{u\in[M]}(\nu_u^{\tilde{\boldsymbol{\pi}}}(K))^{-1}-1)^2}{d})}{\sum_{i=1}^P \mathbb{1}[Y_i = u]}(\boldsymbol{x}_i^t - \sqrt{\bar{\alpha}_t}\boldsymbol{\mu}_u) - \mathbb{E}[\sum_{i=1}^P \mathbb{1}[Y_i = u]\zeta_{i,p,t}^u(s)(\boldsymbol{x}_i^t - \sqrt{\bar{\alpha}_t}\boldsymbol{\mu}_u)] \right\| \\
&\lesssim (1 + \frac{\log d\log\epsilon^{-1}(\min_{u\in[M]}(\nu_u^{\tilde{\boldsymbol{\pi}}}(K))^{-1} - 1)^2}{d}) \cdot \sqrt{1 - \bar{\alpha}_t + \rho^2\bar{\alpha}_t} \cdot \sqrt{\frac{d\log d}{\sum_{i=1}^P \mathbb{1}[Y_i = u]}},
\end{aligned} \tag{141}$$

with a high probability. We also have a similar result for $i$ such that $Y_i \neq u$.

We next study the training phase of the model after $I_1$ iterations. Let $\boldsymbol{\mu}_u$ be the mean of $\boldsymbol{x}_p^t$. Then, we have

$$
\begin{aligned}
&\|(\sqrt{\bar{\alpha}_t}\boldsymbol{\mu}_u - \sum_{i=1}^{P} \boldsymbol{x}_i^t \mathrm{softmax}_p(\frac{\boldsymbol{x}_i^{t\top} \boldsymbol{W}^{(I_1)} \boldsymbol{x}_p^t}{d}))\|^2 \\
&\lesssim ((1 - \sum_{i=1}^{P} \zeta_{i,p,t}^u(I_1))^2 \bar{\alpha}_t d + \frac{(\rho^2 \bar{\alpha}_t + 1 - \bar{\alpha}_t)(1 + \frac{\log d \log \epsilon^{-1}(\min_{u \in [M]}((\min_{u \in [M]}(\nu_u^{\tilde{\pi}}(K))^{-1} - 1)^2)^{-1} - 1)^2}{d})^2 \log d}{\sum_{i=1}^{P} \mathbb{1}[Y_p = u]} d) \\
&\leq d\epsilon,
\end{aligned} \tag{142}
$$

where the second term of the first step is by (141), and the last step is by (140) and holds if

$$
\sum_{i=1}^{P} \mathbb{1}[Y_i = u] \geq (\rho^2 + 1)\epsilon^{-1} \log d. \tag{143}
$$

for any $u \in [M]$ as long as $\epsilon \in (0, \delta^{\Theta(1)})$ with $d \gtrsim \log \epsilon^{-1} \nu_{\min}^{\tilde{\pi}}(K)^{-1}$, which is equivalent to

$$
P \gtrsim \min_{u \in [M]} \{\nu_u^{\tilde{\pi}}(K)\}^{-1}(\rho^2 + 1)\epsilon^{-1} \log d, \tag{144}
$$

with a high probability, because

$$
\begin{aligned}
&\Pr\left(\frac{1}{P}\sum_{i=1}^{P} \mathbb{1}[Y_i = \arg\min_{u \in [M]} \gamma(u)] \geq \left(1 + \sqrt{\frac{\log dK}{P\min_{u \in [M]}\{\nu_u^{\tilde{\pi}}(K)\}}}\right) \min_{u \in [M]}\{\nu_u^{\tilde{\pi}}(K)\}\right) \\
&\leq e^{-\frac{P\min_{u \in [M]}\{\nu_u^{\tilde{\pi}}(K)\} \cdot \frac{\log dK}{P\min_{u \in [M]}\{\nu_u^{\tilde{\pi}}(K)\}}}{3}} \\
&\leq (dK)^{-C}
\end{aligned} \tag{145}
$$

for some $C > 1$. We reduce $\log dK$ to $\log d$ in the final bound of $P$ since $d \geq M \geq K$. Denote $\beta_1 = \boldsymbol{\epsilon}_p'^{\top}(\sqrt{\bar{\alpha}_t}\boldsymbol{\mu}_u - \sum_{i=1}^{P} \boldsymbol{x}_i^t \mathrm{softmax}_p(\frac{\boldsymbol{x}_i^{t\top} \boldsymbol{W}^{(I_1)} \boldsymbol{x}_p^t}{d}))$ and $\beta_2 = \boldsymbol{\epsilon}_p^{\top}(\sqrt{\bar{\alpha}_t}\boldsymbol{\mu}_u - \sum_{i=1}^{P} \boldsymbol{x}_i^t \mathrm{softmax}_p(\frac{\boldsymbol{x}_i^{t\top} \boldsymbol{W}^{(I_1)} \boldsymbol{x}_p^t}{d}))$. Note that by Cauchy-Schwarz inequality,

$$
\mathbb{E}_{\boldsymbol{E}^t, \boldsymbol{X}^0}[\beta_1], \mathbb{E}_{\boldsymbol{E}^t, \boldsymbol{X}^0}[\beta_2] \leq \sqrt{d}\epsilon \tag{146}
$$

Therefore, with (142), we have

$$
\begin{aligned}
&\mathbb{E}_{\boldsymbol{E}^t, \boldsymbol{X}^0}[\|v_t^{(I_1)}(\boldsymbol{x}_p^t - \sum_{i=1}^{P} \boldsymbol{x}_i^t \mathrm{softmax}_p(\frac{\boldsymbol{x}_i^{t\top} \boldsymbol{W}^{(I_1)} \boldsymbol{x}_p^t}{d})) - \boldsymbol{\epsilon}_p\|^2/d] \\
&= \mathbb{E}_{\boldsymbol{E}^t, \boldsymbol{X}^0}[\|v_t^{(I_1)}\rho\sqrt{\bar{\alpha}_t}\boldsymbol{\epsilon}_p' + (v_t^{(I_1)}\sqrt{1 - \bar{\alpha}_t} - 1)\boldsymbol{\epsilon}_p + v_t^{(I_1)}(\sqrt{\bar{\alpha}_t}\boldsymbol{\mu}_u - \sum_{i=1}^{P} \boldsymbol{x}_i^t \mathrm{softmax}_p(\frac{\boldsymbol{x}_i^{t\top} \boldsymbol{W}^{(I_1)} \boldsymbol{x}_p^t}{d}))\|^2/d] \\
&= \mathbb{E}_{\boldsymbol{E}^t, \boldsymbol{X}^0}[\|v_t^{(I_1)}\rho\sqrt{\bar{\alpha}_t}\boldsymbol{\epsilon}_p' + (v_t^{(I_1)}\sqrt{1 - \bar{\alpha}_t} - 1)\boldsymbol{\epsilon}_p\|/d]^2 + (v_t^{(I_1)})^2 \mathbb{E}_{\boldsymbol{E}^t, \boldsymbol{X}^0}[\|\sqrt{\bar{\alpha}_t}\boldsymbol{\mu}_u \\
&\quad - \sum_{i=1}^{P} \boldsymbol{x}_i^t \mathrm{softmax}_p(\frac{\boldsymbol{x}_i^{t\top} \boldsymbol{W}^{(I_1)} \boldsymbol{x}_p^t}{d})\|^2/d] + 2\mathbb{E}_{\boldsymbol{E}^t, \boldsymbol{X}^0}[\frac{\beta_1 v_t^{(I_1)2}\sqrt{\bar{\alpha}_t}}{d}] + 2\mathbb{E}_{\boldsymbol{E}^t, \boldsymbol{X}^0}[\frac{\beta_2(v_t^{(I_1)2}\sqrt{1 - \bar{\alpha}_t} - 1)}{d}] \\
&= ((v_t^{(I_1)})^2\rho^2\bar{\alpha}_t + (v_t^{(I_1)}\sqrt{1 - \bar{\alpha}_t} - 1)^2) + \mathbb{E}_{\boldsymbol{E}^t, \boldsymbol{X}^0}[(v_t^{(I_1)})^2\|\sqrt{\bar{\alpha}_t}\boldsymbol{\mu}_u - \sum_{i=1}^{P} \boldsymbol{x}_i^t \mathrm{softmax}_p(\frac{\boldsymbol{x}_i^{t\top} \boldsymbol{W}^{(I_1)} \boldsymbol{x}_p^t}{d})\|^2/d] \\
&\quad + 2\mathbb{E}_{\boldsymbol{E}^t, \boldsymbol{X}^0}[\frac{\beta_1 v_t^{(I_1)2}\rho\sqrt{\bar{\alpha}_t}}{d}] + 2\mathbb{E}_{\boldsymbol{E}^t, \boldsymbol{X}^0}[\frac{\beta_2 v_t^{(I_1)}(v_t^{(I_1)}\sqrt{1 - \bar{\alpha}_t} - 1)}{d}]
\end{aligned} \tag{147}
$$

Since that $\mathbb{E}_t[\bar{\alpha}_t^2] < \infty$, we have

$$\frac{\partial \mathbb{E}_{\boldsymbol{E}^t, \boldsymbol{X}^0}[\|v_t(\boldsymbol{x}_p^t - \sum_{i=1}^P \boldsymbol{x}_i^t \text{softmax}_p(\frac{\boldsymbol{x}_i^{t\top} \boldsymbol{W}^{(I_1)} \boldsymbol{x}_p^t}{d})) - \boldsymbol{\epsilon}_p\|^2/d]}{\partial v_t}$$

$$=2v_t\rho^2\bar{\alpha}_t + 2\sqrt{1-\bar{\alpha}_t}(v_t\sqrt{1-\bar{\alpha}_t} - 1) + \mathbb{E}_{\boldsymbol{E}^t,\boldsymbol{X}^0}[2v_t\|\sqrt{\bar{\alpha}_t}\boldsymbol{\mu}_u - \sum_{i=1}^P \boldsymbol{x}_i^t\text{softmax}_p(\frac{\boldsymbol{x}_i^{t\top}\boldsymbol{W}^{(I_1)}\boldsymbol{x}_p^t}{d})\|^2/d]$$

$$+ 4\mathbb{E}_{\boldsymbol{E}^t,\boldsymbol{X}^0}\Big[\frac{\beta_1 v_t \rho \sqrt{\bar{\alpha}_t}}{d}\Big] + 2\mathbb{E}_{\boldsymbol{E}^t,\boldsymbol{X}^0}\Big[\frac{2\beta_2\sqrt{1-\bar{\alpha}_t}v_t - \beta_2}{d}\Big] \tag{148}$$

$$=2(v_t \cdot \mathbb{E}_{\boldsymbol{E}^t,\boldsymbol{X}^0}[\bar{\alpha}_t\rho^2 + 1 - \bar{\alpha}_t + \|\sqrt{\bar{\alpha}_t}\boldsymbol{\mu}_u - \sum_{i=1}^P \boldsymbol{x}_i^t\text{softmax}_p(\frac{\boldsymbol{x}_i^{t\top}\boldsymbol{W}^{(I_1)}\boldsymbol{x}_p^t}{d})\|^2/d$$

$$+ \frac{2\beta_1\rho\sqrt{\bar{\alpha}_t}}{d} + \frac{2\beta_2\sqrt{1-\bar{\alpha}_t}}{d}] - (\sqrt{1-\bar{\alpha}_t} + \frac{\beta_2}{d})).$$

Denote

$$v_t^* = \frac{\sqrt{1-\bar{\alpha}_t} + \frac{\beta_2}{d}}{\bar{\alpha}_t\rho^2 + 1 - \bar{\alpha}_t + \mathbb{E}_{\boldsymbol{E}^t,\boldsymbol{X}^0}[\|\sqrt{\bar{\alpha}_t}\boldsymbol{\mu}_u - \sum_{i=1}^P \boldsymbol{x}_i^t\text{softmax}_p(\frac{\boldsymbol{x}_i^{t\top}\boldsymbol{W}^{(I_1)}\boldsymbol{x}_p^t}{d})\|^2/d] + 2\frac{\beta_1\rho\sqrt{\bar{\alpha}_t}+\beta_2\sqrt{1-\bar{\alpha}_t}}{d}}. \tag{149}$$

Define the error $e_t(s) = v_t^{(s)} - v_t^*$. Then, by gradient update,

$$e_t(s+1)$$

$$=v_t^{(s+1)} - v_t^*$$

$$=v_t^{(s)} - v_t^* - 2\eta \cdot (v_t^{(s)} \cdot \mathbb{E}_{\boldsymbol{E}^t,\boldsymbol{X}^0}[\bar{\alpha}_t\rho^2 + 1 - \bar{\alpha}_t + \|\sqrt{\bar{\alpha}_t}\boldsymbol{\mu}_u - \sum_{i=1}^P \boldsymbol{x}_i^t\text{softmax}_p(\frac{\boldsymbol{x}_i^{t\top}\boldsymbol{W}^{(I_1)}\boldsymbol{x}_p^t}{d})\|^2/d$$

$$+ \frac{2\beta_1\rho\sqrt{\bar{\alpha}_t}}{d} + \frac{2\beta_2\sqrt{1-\bar{\alpha}_t}}{d}] - (\sqrt{1-\bar{\alpha}_t} + \frac{\beta_2}{d}))$$

$$=(v_t^{(s)} - v_t^*) - 2\eta \cdot \mathbb{E}_{\boldsymbol{E}^t,\boldsymbol{X}^0}[\bar{\alpha}_t\rho^2 + 1 - \bar{\alpha}_t + \|\sqrt{\bar{\alpha}_t}\boldsymbol{\mu}_u - \sum_{i=1}^P \boldsymbol{x}_i^t\text{softmax}_p(\frac{\boldsymbol{x}_i^{t\top}\boldsymbol{W}^{(I_1)}\boldsymbol{x}_p^t}{d})\|^2/d \tag{150}$$

$$+ \frac{2\beta_1\rho\sqrt{\bar{\alpha}_t}}{d} + \frac{2\beta_2\sqrt{1-\bar{\alpha}_t}}{d}] \cdot (v_t^{(s)} - v_t^*)$$

$$=(1 - 2\eta \cdot \mathbb{E}_{\boldsymbol{E}^t,\boldsymbol{X}^0}[\bar{\alpha}_t\rho^2 + 1 - \bar{\alpha}_t + \|\sqrt{\bar{\alpha}_t}\boldsymbol{\mu}_u - \sum_{i=1}^P \boldsymbol{x}_i^t\text{softmax}_p(\frac{\boldsymbol{x}_i^{t\top}\boldsymbol{W}^{(I_1)}\boldsymbol{x}_p^t}{d})\|^2/d$$

$$+ \frac{2\beta_1\rho\sqrt{\bar{\alpha}_t}}{d} + \frac{2\beta_2\sqrt{1-\bar{\alpha}_t}}{d}]) \cdot e_t(s).$$

Hence, given

$$\eta \lesssim (\mathbb{E}_{\boldsymbol{E}^t,\boldsymbol{X}^0}[\bar{\alpha}_t\rho^2 + 1 - \bar{\alpha}_t + \|\sqrt{\bar{\alpha}_t}\boldsymbol{\mu}_u - \sum_{i=1}^P \boldsymbol{x}_i^t\text{softmax}_p(\frac{\boldsymbol{x}_i^{t\top}\boldsymbol{W}^{(I_1)}\boldsymbol{x}_p^t}{d})\|^2/d + \frac{2\beta_1\rho\sqrt{\bar{\alpha}_t}}{d}$$

$$+ \frac{2\beta_2\sqrt{1-\bar{\alpha}_t}}{d}])^{-1/2}, \tag{151}$$

i.e.,

$$\eta \lesssim \frac{1}{\max\{\rho, 1\} + \epsilon} \tag{152}$$

by (142), we can derive

$$|e_t(s)| \leq (1 - 2\eta \cdot \mathbb{E}_{\boldsymbol{E}^t,\boldsymbol{X}^0}[\bar{\alpha}_t\rho^2 + 1 - \bar{\alpha}_t + \|\sqrt{\bar{\alpha}_t}\boldsymbol{\mu}_u - \sum_{i=1}^P \boldsymbol{x}_i^t\text{softmax}_p(\frac{\boldsymbol{x}_i^{t\top}\boldsymbol{W}^{(I_1)}\boldsymbol{x}_p^t}{d})\|^2/d$$

$$+ \frac{2\beta_1\rho\sqrt{\bar{\alpha}_t}}{d} + \frac{2\beta_2\sqrt{1-\bar{\alpha}_t}}{d}])^s \cdot |e_t(I_1)|, \tag{153}$$

which means after

$$
\begin{aligned}
s \gtrsim I_2 :=& \log \frac{|e_t(I_1)|}{\epsilon} \Big/ \log(1 - 2\eta \cdot \mathbb{E}_{\boldsymbol{E}^t, \boldsymbol{X}^0}[\bar{\alpha}_t \rho^2 + 1 - \bar{\alpha}_t + \|\sqrt{\bar{\alpha}_t} \boldsymbol{\mu}_u - \sum_{i=1}^{P} \boldsymbol{x}_i^t \mathrm{softmax}_p(\frac{\boldsymbol{x}_i^{t\top} \boldsymbol{W}^{(I_1)} \boldsymbol{x}_p^t}{d})\|^2/d \\
& + \frac{2\beta_1 \rho \sqrt{\bar{\alpha}_t}}{d} + \frac{2\beta_2 \sqrt{1 - \bar{\alpha}_t}}{d}])^{-1} \\
=& \Theta(\log \frac{|e_t(I_1)|}{\epsilon})
\end{aligned}
\tag{154}
$$

iterations, we can achieve that $|e_t(s)| \leq \epsilon$ and $v_t^{(s)}$ converges to $v_t^*$. Given $v_t^{(0)} = \Theta(1)$, by (148), we have that for any $v_t \leq \Theta(1)$,

$$
\begin{aligned}
& \frac{\partial \mathbb{E}_{\boldsymbol{E}^t, \boldsymbol{X}^0}[\|v_t(\boldsymbol{x}_p^t - \sum_{i=1}^{P} \boldsymbol{x}_i^t \mathrm{softmax}_p(\frac{\boldsymbol{x}_i^{t\top} \boldsymbol{W}^{(I_1)} \boldsymbol{x}_p^t}{d})) - \boldsymbol{\epsilon}_p\|^2/d]}{\partial v_t} \\
& \leq \mathbb{E}_{\boldsymbol{E}^t, \boldsymbol{X}^0}[\bar{\alpha}_t \rho^2 + 1 - \bar{\alpha}_t + \bar{\alpha}_t + \frac{2\beta_1 \rho \sqrt{\bar{\alpha}_t}}{d} + \frac{2\beta_2 \sqrt{1 - \bar{\alpha}_t}}{d}] - (\sqrt{1 - \bar{\alpha}_t} + \frac{\beta_2}{d})] \\
& \lesssim \rho^2 + 1,
\end{aligned}
\tag{155}
$$

where the last step is by $\beta_1 \sim \mathcal{N}(0, \|\sqrt{\bar{\alpha}_t} \boldsymbol{\mu}_u - \sum_{i=1}^{P} \boldsymbol{x}_i^t \mathrm{softmax}_p(\frac{\boldsymbol{x}_i^{t\top} \boldsymbol{W}^{(I_1)} \boldsymbol{x}_p^t}{d})\|^2)$, $\beta_2 \sim \mathcal{N}(0, \|\sqrt{\bar{\alpha}_t} \boldsymbol{\mu}_u - \sum_{i=1}^{P} \boldsymbol{x}_i^t \mathrm{softmax}_p(\frac{\boldsymbol{x}_i^{t\top} \boldsymbol{W}^{(I_1)} \boldsymbol{x}_p^t}{d})\|^2)$, so that with a high probability,

$$
\beta_1, \beta_2 \lesssim \sqrt{d}.
\tag{156}
$$

Hence,

$$
\begin{aligned}
|v_t^{(I_1)}| =& \Big|v_t^{(0)} - \sum_{s=0}^{I_1-1} \frac{\partial \mathbb{E}_{\boldsymbol{E}^t, \boldsymbol{X}^0}[\|v_t^{(s)}(\boldsymbol{x}_p^t - \sum_{i=1}^{P} \boldsymbol{x}_i^t \mathrm{softmax}_p(\frac{\boldsymbol{x}_i^{t\top} \boldsymbol{W}^{(I_1)} \boldsymbol{x}_p^t}{d})) - \boldsymbol{\epsilon}_p\|^2/d]}{\partial v_t}\Big| \\
& \lesssim I_1(\rho^2 + 1),
\end{aligned}
\tag{157}
$$

and

$$
|e_t(I_1)| \lesssim I_1(\rho^2 + 1).
\tag{158}
$$

Then,

$$
I_2 = \Theta(\log \frac{I_1(\rho^2 + 1)}{\epsilon}).
\tag{159}
$$

$\square$

