# OpenReview forum: "Transformers Learn the Optimal DDPM Denoiser for Multi-Token GMMs"
_ICML.cc/2026/Conference — ICML 2026 regular_

### Official Review · Reviewer_Yw16 · 2026-03-02

**Soundness:** 3
**Presentation:** 3
**Significance:** 2
**Originality:** 2
**Overall Recommendation:** 3
**Confidence:** 3

**Summary:**

This paper analyzes the performance of DDPM-type diffusion models when Transformer-based neural networks are used as the denoiser. In particular, it considers the setting where the data distribution follows a multi-token Gaussian mixture model. The authors focus on a one-layer Transformer architecture and demonstrate that, through a so-called mean denoising mechanism, the model can approximate the oracle Minimum Mean Squared Error (MMSE) estimator, i.e., the estimator that assumes the latent variable Y_p is known, provided that training is sufficiently long.

A central idea in proving the above optimality result is that, for each input token the self-attention mechanism approximately outputs the mean of the tokens that share the same underlying pattern.

**Compliance With Llm Reviewing Policy:**

Affirmed.

**Final Justification:**

I remain somewhat skeptical about the extent to which this theoretical analysis translates into practical benefits for real-world transformer-based diffusion models.

**Key Questions For Authors:**

1, What types of real-world data distributions can be well approximated by the multi-token GMM assumption?

2, For the analysis in the paper to hold, how should M and P scale with respect to each other? In particular, can the stated optimality results still be achieved when M is very large?

3, Do the authors believe that the current proof can be extended to more general Transformer architectures with multiple layers?

4, Can the authors identify a real-world dataset whose data distribution can be well modeled by the multi-token GMM assumption, and for which the theoretical analysis in this paper is expected to hold?

**Limitations:**

1, The real-world data distribution may not be well modeled by the multi-token GMM

2, The analysis in the paper may be only valid when M and P scale in a certain way, which may not be the case for real-world data distribution.

3, The proof technique may be valid only for single-layer transformers

**Strengths And Weaknesses:**

**Strength:** This paper addresses a fundamentally important question in understanding DDPM-type diffusion models that use Transformer architectures. It provides a theoretical analysis of their performance and characterizes the conditions under which they achieve optimality. In particular, it tackles the critical problem of whether training enables the model to approximate the true score function.

**Weakness**:

1, The analysis is primarily based on a multi-token Gaussian mixture data distribution. However, it remains unclear to what extent real-world data distributions can be accurately modeled by such a multi-token GMM assumption.

2, It appears that for the “mean denoising mechanism” to be effective, the model must observe repeated instances of the same underlying patterns within a single data sample. This suggests that the analysis may only hold in regimes where the number of distinct patterns M is much smaller than the number of tokens P, so that each pattern is sufficiently represented within the sample.

3, It is unclear whether the proof technique developed for the one-layer Transformer can be generalized to multi-layer Transformer architectures that include MLP blocks and nonlinearities.

4, The experimental results are primarily conducted on synthetic or toy datasets. As a result, it is difficult to assess whether the multi-token GMM modeling assumptions and the corresponding analysis for this special case generalize to more realistic data distributions.

---

> ### Author Rebuttal · Authors · 2026-03-31
>
> We thank Reviewer Yw16 for the time and effort in the evaluation.
>
> First, we provide a **high-level response** to the concerns on data and model assumptions. This paper aims to establish a theoretical understanding and convergence analysis of DDPM training with Transformer-based models. Due to current limitations in theoretical tools, making appropriate assumptions on data distributions and models is necessary. We believe that theoretical insights become more predictive for real problems as assumptions are relaxed. In this regard, our assumptions are already more general and better capture key characteristics of real data compared to prior work (see A1). This is also reflected in other reviewers’ comments. For example, Reviewer Fudt indicates that our data distribution assumption "is much richer than distributions usually considered in the literature," and Reviewer KLEe mentions that it "strikes a good balance between tractability and realism." Extending the analysis to more general settings is an important direction for future work, and our results potentially provide a step toward this goal.
>
> Next, we address each concern in detail:
>
> **Q1 (Weakness 1, 4, Question 1, 4, and Limitation 1): What real-world data distributions can be well approximated by the MTGM assumption?**
>
> **A1**: Please refer to the discussion above regarding the role of assumptions in enabling tractable theoretical analysis. Next, we elaborate on why our data assumptions are better than those in prior work as follows. First, our MTGM assumption is designed as a tractable abstraction that captures key properties of real-world data, rather than to model it exactly. In particular, real data often exhibit: (i) recurring semantic or textural patterns (e.g., sky or grass in images), (ii) a decomposition into multiple tokens or patches for processing, and (iii) feature representations that cluster around a limited number of modes. Our MTGM model reflects these properties by allowing each data point to select patterns from a random subset of $K$ out of $M$ patterns, with multiple tokens corresponding to the same pattern. Second, the MTGM distribution is more general than assumptions used in prior work. Many recent theoretical studies (Min \& Vidal, 2025; Wang et al., 2024a) on classification or diffusion models assume Gaussian mixture models, which correspond to the special case $K=P=1$ in our Definition 1. In contrast, our setting with $K, P \geq 1$ captures a more general scenario where each data point contains multiple tokens and may involve multiple modes.
>
> **Q2 (Weakness 2 \& Question 2 \& Limitation 2): How does $M$ scale with $P$? The scaling may not be practical. Can the optimality results still be achieved when $M$ is very large?**
>
> **A2**: This is a great question. We want to kindly clarify that $P$ does not necessarily scale with $M$. By Theorem 1 (condition (iii), line 209 left), $P$ scales with a function of $K$, the number of distinct patterns per data point, rather than $M$, the total number of patterns in the distribution $\mathcal{D}(\tilde{\pi},K,\{\mu_i\}_{i=1}^M, \rho)$. Since $K$ can be much smaller than $M$, increasing $M$ does not necessarily increase the lower bound on $P$. However, for certain regimes, we agree that $P$ may scale with $M$. For example, when $K=M$ and $\tilde{\pi}$ is uniform with $\pi_u=1/M$, condition (iii) implies $P\geq \Omega(M(\rho^2+1)\epsilon^{-1}\log d)$, which increases with $M$. As discussed in A1, this reflects an abstraction of real data with many recurring patterns, where the number of tokens should scale accordingly.
>
>
>
> **Q3 (Weakness 3 \& Question 3 \& Limitation 3): Can the analysis be generalized to multi-layer Transformer architectures that include MLP blocks and nonlinearities?**
>
> **A3**: Thank you for the question. Extending the analysis to multi-layer Transformers with nonlinearities is highly nontrivial due to complex layer-wise interactions and increased non-convexity, requiring more advanced tools and possibly stronger assumptions. Notably, much of the recent theoretical work on Transformer optimization and learning dynamics [Li et al. (2023a; 2024a); Huang et al., 2024a; Shen et al., 2025] also focuses on one-layer architectures. Our work studies the one-layer setting as a first tractable framework, and to our knowledge, no prior work analyzes the convergence of DDPM training even in this case, highlighting the novelty of our results. Some findings are also empirically supported in multi-layer settings (Section 5.2). Lastly, as shown in Section 4.1, the optimal model corresponds to learning the oracle MMSE (Eqn. 13). Even with an additional MLP on the top of $v_t$, its optimal parameters will still correspond to those making the MLP to approximately act as a linear block. Therefore, we focus on analyzing the self-attention component while keeping this part fixed.

---

> > ### Author Rebuttal · Reviewer_Yw16 · 2026-04-01
> >
> > Thank you to the authors for the detailed responses to my questions. However, I remain concerned that the analysis is overly simplified and may not provide meaningful guidance for practical diffusion models, particularly those based on multi-layer transformers and real-world data distributions.

---

> > > ### Author Response · Authors · 2026-04-07
> > >
> > > We thank you for the follow-up question. We will incorporate the suggestions and our response in the revision. Your concerns regarding the assumptions and practical insights are very common and important, and we respond as follows.
> > >
> > > **First, our assumptions are not oversimplified. Instead, they are the mildest assumptions that still make the problem theoretically analyzable.** This misunderstanding may obscure our contributions and the difficulty of technical challenges as follows:
> > >
> > > 1. **No existing work explains why a nonlinear Transformer can be optimized via gradient-based methods in DDPM training to reach an optimal denoiser.** Existing works either study the optimal denoiser through loss landscape analysis [Wang et al., 2024a; Li et al., 2024e] or analyze training dynamics only under simplified, approximately linear models [Han et al., 2024; Wang \& Pehlevan, 2025; Bonnaire et al., 2025]. This is because both the convergence analysis of nonlinear Transformers and that of diffusion models are inherently difficult problems: the former is due to non-convex optimization in nonlinear models, and the latter is because analyzing denoising tasks differs from the commonly studied regression or classification tasks. **Our work even combines these two challenges, further increasing the difficulty.** Our contributions to mathematical and optimization theory are also recognized by Reviewer 24th.
> > >
> > > 2. **Analyzing multi-layer Transformers, or even multi-layer neural networks in general, is extremely challenging, while the one-layer case represents a stepping stone towards the multilayer case**. To the best of our knowledge, no existing works analyze the convergence of multi-layer nonlinear Transformers, due to the highly non-convex nature of the problem. Up till now, one-layer nonlinear Transformers are still among the most complicated models that can currently be analyzed theoretically in the family of Transformer models. In addition, we believe that our current analysis of a one-layer Transformer is useful for future extensions to multi-layer settings. A potential direction is to iteratively stack our analysis of how a one-layer model learns MTGM, so that we can characterize how multi-layer Transformers learn hierarchical MTGM data that is formulated based on patterns with multiple scales and levels. Therefore, our current analysis of a one-layer nonlinear Transformer is a crucial first step for this line of research.
> > >
> > > 3. **In the analysis of nonlinear Transformers, our data assumptions are already more mild and general than those in prior work, which is acknowledged by Reviewers Fudt and KLEe, and are therefore not overly simplified.** In fact, when $K=P=1$, our assumptions can be reduced to those used in existing studies. In addition, our MTGM assumption is a good model for images. In computer vision, image data can be formulated as a collection of various patterns, with each pattern corresponding to multiple patches. For example, an image of a bird may contain patterns such as the bird and the sky, where the sky consists of repeated blue pixel patches. Please see [**A1**](https://openreview.net/forum?id=6VyoaDj3zB&noteId=aYHXcKiej9) for details. **Therefore, we fundamentally disagree with the reviewer that MTGM is not representative of real data.**
> > >
> > > **Next, we would like to clarify that our theoretical conclusions provide explanations for empirical observations observed in [1,2,3,4].** Our Theorem 1 suggests that the convergence is faster when the time-averaged SNR is not too small. This is qualitatively consistent with recent empirical works [1,2] that promote time steps with non-trivial SNR through resampling or reweighting to accelerate diffusion training. Our Theorem 1 also predicts that less frequent patterns require more training iterations to learn. This is aligned with empirical evidence [3,4] from long-tailed diffusion modeling that tail classes are more challenging to learn and require imbalanced learning strategies.
> > >
> > > In summary, our work provides a necessary first step toward understanding the training and denoising mechanisms of Transformer-based diffusion models. The assumptions we adopt are not simplifications for convenience, but the minimal conditions under which this highly challenging problem can become analyzable. Our results also offer insights that align with recent empirical advances in diffusion training. We believe that our analysis provides meaningful guidance and establishes a foundation for future work on more complex and realistic settings. **We sincerely hope the reviewer can take the above into consideration and re-evaluate our contributions.**
> > >
> > > [1] Hang et al., ICCV 2023. Efficient Diffusion Training via Min-SNR Weighting Strategy.
> > >
> > > [2] Hang et al., ICCV 2025. Improved noise schedule for diffusion training.
> > >
> > > [3] Qin et al., CVPR 2023. Class-Balancing Diffusion Models.
> > >
> > > [4] Yan et al., 2024. Training class-imbalanced diffusion model via overlap optimization.

---

### Official Review · Reviewer_24th · 2026-03-02

**Soundness:** 3
**Presentation:** 3
**Significance:** 2
**Originality:** 3
**Overall Recommendation:** 5
**Confidence:** 2

**Summary:**

The paper does theoretical analysis on the following toy problem:

**Data distribution:** the data distribution studied is a Multi-Token Gaussian Mixture is generated at a high level by the following procedure:
1. Sample a latent bit string $Z$ with $M$ components uniformly over the set of binary vectors for which exactly $K$ bits are $1$
2. For each token, sample an index from a predefined base distribution, conditioned on corresponding bit string element taking a value of $1$, ie:
$$Y_p \sim \pi | Z_{Y_p} = 1$$
3. For each token, take the mean vector corresponding to its index $\mu_{Y_p}$ and sample it according to a gaussian with this mean:
$$x_p \sim \mathcal{N}(\mu_{Y_p}, \rho^2 I)$$

Crucially

**Transformer Model:** The paper considers a single-head attention layer for its transformer model, with decoupled learned scalars $v_1,v_2 \dots v_t$ for each noise timestep, ie:

$$ f(X, t) = v_t X (I - A),$$

Where $A = \texttt{softmax} ( \frac{X^\intercal W X}{d})$ is the attention matrix. Note that this formulation of the attention layer:
- does not use any value or output heads,
- considers the query and key heads as fused into the weight matrix $W = W_K^{\intercal} W_Q$, and implicitly assumes that this matrix has full rank, which is only the case if the query and key heads have a dimension greater than the embedding dimension $d$,
- uses $d$ as opposed to $\sqrt{d}$ in the denominator.

**Results:** A rough outline of the key results shown in the paper is that, for this specific formulation of a single-head attention layer, vanilla Stochastic Gradient Descent (SGD) will push the model towards assigning high attention scores to tokens with the same mean, such that after training and when passed a partially noised input $x$:

$$x_p - (X \texttt{softmax} ( X^\intercal W X/d)_p $$

$$\qquad \approx x_p - \frac{1}{\vert \{ q: Y_p = Y_q \}\vert}\sum_{q: Y_p = Y_q} x_q$$

And thus, using that $\frac{1}{\vert \{ q: Y_p = Y_q \}\vert}\sum_{q: Y_p = Y_q} x_q \approx \mu_{Y_p}$ the optimal prediction of the noise, ie $E \approx x_p -  \mu_{Y_p}$. The paper proves that under the assumptions of a sufficient number of sufficiently granular steps, SGD will converge to this solution with high probability, which is shown to be optimal. This result is validated on data synthetically generated to match the described distribution with the described attention layer.

**Compliance With Llm Reviewing Policy:**

Affirmed.

**Final Justification:**

I believe the authors addressed most of the weaknesses laid out. I am less confident in my assessment of the paper's significance, hinging on how well results on the MTGM distribution generalize the real world.

**Key Questions For Authors:**

- I believe there is a typo in equation (1). When it is written that $f(...)_p = v_t(X - X \texttt{softmax} ...)$, as my understanding $f(...)_p$ should be a vector, but the RHS would be a matrix. Is this supposed to be $f(...) = v_t(X - X \texttt{softmax} ...)$?
- Does the work generalize to the case of $W_Q$ and $W_K$ matrices being optimized separately? What about if $W$ is low-rank?
- Why use $d$ in the denominator of the softmax argument instead of $\sqrt{d}$ as is typical?
- Can the proposed results generalize to attention layers with output-value heads? How crucial is the negative sign preceding the softmax in the proof?
- It's not clear to me as a reader why the _minimal average pattern ratio_ is not simply $\min_{u} \mathbb{P} [Y_p = u]$. Why can't one do the derivation:

$$\mathbb{E} \left[ \sum_{p=1}^P  \mathbb{E} \mathbb{1} [Y_p = u]/P \right] =\sum_{p=1}^P  \mathbb{E} \mathbb{1} [Y_p = u] / P = \mathbb{E} \mathbb{1} [Y_p = u] = \mathbb{P} [Y_p = u].$$

If this derivation is valid, why not use it as a simpler definition?

**Limitations:**

In general, I find the discussion in Section 4.3 Proof Idea, Technical Novelty, and Limitations to be quite good. Here are some additional limitations that need further discussion:
- In the introduction it would be more accurate to describe the studied architecture as a single attention head, rather than a whole transformer.
- The authors should further clearly state that they are _not_ considering transformers with output-value heads and state why.
- If the work does not generalize trivially to low-rank $W$ or to the case of $W_Q$ and $W_K$ matrices being optimized separately, the authors should state this too.

**Strengths And Weaknesses:**

**Strengths**

Soundness - The proof is technically impressive and correct as far as I can tell.

Originality - The results are novel as far as I can tell.

Presentation - In general, I think the results are well presented, barring some clarifications discussed below.

**Weaknesses**

Significance - I'm not convinced that this general line of work, even after significant generalizations of the proposed results, could actually be used to improve the training of diffusion transformers in-the-wild. The regularity of the proposed data distribution is too high compared to real-world data, and is instrumental in the proof as . Nonetheless, I am inclined to accept interesting math and optimization theory for its own sake.

**Nitpicks**
- typo on line 283: by "Corollary 2 and Proposition 2, one obtain our main Theorem 1..." I believe this should read "obtains"
- potential typo in equation (1) mentioned in Key Questions For Authors

---

> ### Author Rebuttal · Authors · 2026-03-31
>
> We sincerely thank you for the careful and detailed evaluation of our paper. We especially appreciate the detailed questions on the technical aspect of the manuscript. We address the questions below one by one, and we will revise our paper accordingly to improve the clarity of our work.
>
> **Q1 (Weakness): The regularity of the proposed data distribution is too high compared to real data.**
>
> **A1**: Thank you for your question. Regarding why we adopt the MTGM data distribution, please refer to the high-level response and **[A1](https://openreview.net/forum?id=6VyoaDj3zB&noteId=aYHXcKiej9)** to Reviewer Yw16. We thank Reviewer 24th for recognizing our contributions to mathematical and optimization theory.
>
> \textbf{Q2 (Question 1): Is it supposed to be $f(\cdots)=v_t(X-X softmax\cdots)$?}
>
> A2: A sharp observation! Your understanding that $f(\cdots )_p$ should be a vector is correct. There is a typo in the second line of Eqn. 1, which should be corrected as $f(\Psi;X,t)_p=v_t(x_p-X\text{softmax}(\frac{X^\top W x_p}{d}))$. Here $X\in\mathbb{R}^{d\times P}$, $x_p\in\mathbb{R}^d$, $\text{softmax}(\frac{X^\top W x_p}{d})\in\mathbb{R}^P$. Hence, $f(\Psi;X,t)_p\in\mathbb{R}^d$ is a $d$-dimensional vector.
>
> **Q3 (Question 2 \& Limitation 2): Does the work generalize to when $W_Q$ and $W_K$ matrices are optimized separately? What if $W$ is low-rank?**
>
> **A3**: This is a great question. We will incorporate this discussion into our revision. Optimizing $W_K$ and $W_Q$ separately makes the loss landscape much more complicated. For example, it is not difficult to show that
> $W_K=W_Q=0$ is a stationary point that is not globally optimal. Therefore, under Gaussian initialization of $W_K$ and $W_Q$, characterizing the training dynamics of the Transformer requires additional analysis to escape saddle points or avoid spurious local minima. Existing works on Transformer training dynamics either, like ours, only consider training with $W=W_K^\top W_Q$, or adopt non-random initialization (Li et al., (2023a; 2024a)) for $W_K$ and $W_Q$ to obtain a favorable initial optimization direction. Using techniques in those works, we expect our results to extend to training with $W_K$ and $W_Q$, where $W=W_K^\top W_Q$ can be low-rank with a smaller embedding dimension, as long as the dimension is no smaller than $M$, since $W$ needs to learn all $M$ patterns.
>
> **Q4 (Question 3): Why use $d$ in the denominator of the softmax argument instead of $\sqrt{d}$?**
>
> **A4**: This is an interesting question. We would first like to clarify that this is a scaling choice tailored to our theoretical formulation, rather than a modification for practical applications. We choose the denominator to be $d$ to control $\frac{x_p^0{}^\top W x_j^0}{d}$ as $O(1)$ order with regard to $d$ for any $p,j\in[P]$ (shown in eq (18) in our Proposition 3) since the norm of each token is of order $\sqrt{d}$. If we choose the denominator to be $\sqrt{d}$, the gradient of $W$ with be scaled up by $\sqrt{d}$, we can still reach a similar convergence results for $W$ and $v_t$ if we scale down the step size $\eta$ by $\sqrt{d}$. However, note that since the step size of updating $v_t$ is also scaled down, it will take $O(\sqrt{d})$ times more iterations for $v_t$ to converge. Therefore, we choose $d$ as the denominator in our theoretical analysis.
>
> **Q5 (Question 4 \& Limitation 2): Can the results extend to attention layers with output-value heads, and how crucial is the negative sign?**
>
> **A5**: Thank you for the question. As shown in Section 4.1, the optimal model corresponds to learning the oracle MMSE in Eqn. 13. With an additional output-value layer, i.e., $f=v_t(X+W_VX\mathrm{softmax}[\cdots])$, the optimal $W_V$ remains close to $-I$. Therefore, we choose to simplify this part as a fixed $W_V=-I$ (which explains the negative sign) and focus on the training dynamics of self-attention. Extending the analysis to settings with output-value layers and additional nonlinearities is nontrivial due to more complex interactions and increased non-convexity, and would require more advanced tools. To our knowledge, no existing work theoretically studies the convergence of DDPM training even for one-layer Transformers, highlighting the novelty of our results. We leave these extensions to future work.
>
>
> **Q6 (Question 5): Why is the minimal average pattern ratio not $\min_u \Pr(Y_p=u)$.**
>
> **A6**: Thank you for the suggestion. The derivation is correct, and we can revise (7) as $\nu_u^{\tilde{\pi}}(K)=\min_u \Pr(Y_1=u)$, using $Y_1$ instead of $Y_p$ to avoid dependence on $p$. We will incorporate this in the revision.
>
> **Q7 (Limitation 1): The studied architecture is more accurately described as a single attention head**
>
> **A7**: Thank you for the suggestion. We refer to it as a Transformer since it includes a residual connection and a fixed output-value layer ($-I$). We will clarify this and emphasize the role of attention in the theoretical analysis and mechanism discussion.

---

> > ### Author Rebuttal · Reviewer_24th · 2026-04-02
> >
> > Thank you for your feedback. I believe you have addressed all most of my concerns and have updated my score accordingly.
> >
> > I acknowledge that the MTGM distribution is more general than prior work. Nonetheless, I think that it is still a key barrier to real-world applicability in AI: transformers are mostly applied in sequence learning problems with rich structure between tokens the sequence (eg. Induction Heads), the MTGM distribution only has a weak amount of structure shared between tokens.

---

> > > ### Author Response · Authors · 2026-04-07
> > >
> > > We are glad that your concerns have been fully addressed. We will incorporate the response in our revision. We greatly appreciate that you have increased your rating to 5. Your suggestions are very constructive, and we agree that Transformers are often used for sequence learning, and incorporating sequential structures among tokens into our theoretical framework is an important research direction. We also believe that extending our theory to various practical applications, including tasks related to sequence learning, is crucial. We leave this part as future work.

---

### Official Review · Reviewer_KLEe · 2026-03-07

**Soundness:** 3
**Presentation:** 3
**Significance:** 2
**Originality:** 3
**Overall Recommendation:** 4
**Confidence:** 3

**Summary:**

This paper studies the convergence of a single-layer, single-head transformer to a minimizer of the DDPM loss when trained via gradient descent on data from a toy distribution called a multi-token Gaussian mixture, which models data formed via compositions of parts. The main result shows that this simple transformer can achieve a risk arbitrarily close to the Bayes optimal value given sufficiently many gradient iterations provided the data satisfies a regularity condition on the number of tokens per sample. The authors also characterize the denoiser that is learned via this optimization procedure and study the properties of the learned attention mechanism. They complement their theoretical results with synthetic experiments and some cursory experiments on real-world models and data.

**Compliance With Llm Reviewing Policy:**

Affirmed.

**Final Justification:**

I maintain my positive assessment of this paper and continue to recommend acceptance.

**Key Questions For Authors:**

1. Can this theory make testable predictions about real-world diffusion models? Section 5.2 includes a small real-world experiment, but the  empirical findings seem to be only tangentially related to the main results in this paper.

2. How important is the orthogonality assumption in the MGTM patterns? This seems like a notable constraint on the model's generality.

3. Q2 in the introduction asks whether gradient descent converges to the optimal non-linear transformer under DDPM training. Can this theory make interesting and testable predictions about the denoiser's along the gradient descent path? I wonder if these results can be adapted to study generalization in diffusion models along the lines of Favero et al. (2025) and Bonnaire et al. (2025).

**Limitations:**

Yes

**Strengths And Weaknesses:**

This is an interesting contribution to the theory of diffusion models. I appreciate the use of the multi-token Gaussian mixture distribution as a toy model for data formed as a composition of multiple parts; this model seems to strike a good balance between tractability and realism. The results are plausible to the best of my knowledge, though I have not checked the proofs in the appendices in great detail. While this manuscript is quite dense, the writing quality is generally good, and the authors do a reasonable job of providing intuitive explanations for their main theoretical results. My main critique of this paper is that I have some concerns about whether this theory is able to make testable predictions about real-world diffusion models. While I understand the need for simplifying assumptions and toy models to prove rigorous results, I wonder to what extent this toy model is able to provide insights about real-world diffusion models and to what extent the techniques can be adapted to more general settings.

---

> ### Author Rebuttal · Authors · 2026-03-31
>
> We thank Reviewer KLEe for the time and effort in the evaluation. We address your concerns one by one as follows.
>
> **Q1 (Weakness \& Question 1): Can this theory make testable predictions about real-world diffusion models? The empirical findings seem to be only tangentially related to the main results in this paper.**
>
> **A1**: This is a great question. With regard to whether our formulation and assumptions are significant enough for real-world data and models, please refer to the high-level response and **[A1](https://openreview.net/forum?id=6VyoaDj3zB&noteId=aYHXcKiej9)** to Reviewer Yw16. In addition, we would like to clarify that although the MNIST dataset does not perfectly match the MTGM data we consider, the experimental results in Section 5.2 are still consistent with our theoretical insights from Theorem 1. Prior theoretical work (Han et al., 2025) on feature learning in DDPM also uses MNIST in their real-world data experiment. In the training data of our experimental setup, digit "2" has the smallest number of samples, and its pattern can be viewed as analogous to the pattern component with the smallest proportion in MTGM. Figure 3 shows that the training of digit "2" converges significantly more slowly than that of the other digits. As a result, whether the entire dataset is sufficiently learned depends on when the minority class of digit "2" is adequately learned. This is consistent with the insight from Equation (9) in Theorem 1, which states that the required number of training iterations depends on the minimal average pattern ratio. We believe that a promising future direction is to extend experiments to datasets where each data point contains richer patterns associated with particular attributes, such as CelebA, to further validate our theory.
>
> **Q2 (Question 2): How important is the orthogonality assumption in the MGTM patterns? This seems like a notable constraint on the model's generality.**
>
>
> **A2**: Orthogonality in our analysis is primarily used to formalize the separability between different patterns in a simple and tractable way. It can be viewed as a convenient version of the separability condition that enables us to characterize how the model distinguishes among different patterns. Similar assumptions are commonly adopted in prior theoretical works on feature learning (Li et al., (2024a; 2024b); Huang et al., 2023; Min \& Vidal, 2025), where some form of separation between patterns or features is required. Importantly, our analysis does not rely on exact orthogonality. The assumption can be relaxed to approximate orthogonality, or that different patterns are nearly orthogonal with high probability. Such conditions are well-motivated in high-dimensional settings, where randomly generated vectors tend to be close to orthogonal. Therefore, the orthogonality assumption mainly serves as a simplifying technical condition rather than a restrictive requirement.
>
> **Q3 (Question 3): Can this theory make predictions about the denoiser's along the gradient descent path? I wonder if these results can be adapted to study generalization along the lines of Favero et al. (2025) and Bonnaire et al. (2025).**
>
> **A3**: For your first question on whether our theory can predict the behavior of the denoiser along the gradient descent path, our analysis is indeed based on characterizing gradient updates at each step. In particular, Lemma 3 (Appendix B) shows that the gradient magnitude of $W$ has a positive lower bound along directions where the query–key pair shares the same pattern, and is small otherwise. This suggests that testable predictions along the gradient descent trajectory are possible, though a full characterization would require additional effort and is left for future work.
>
> For your second question, we would like to confirm whether Favero et al. (2025) refers to [1], which is not currently cited. If so, we believe there are conceptual connections. In our framework, generalization can be studied via denoising error under changes in the MTGM distribution. Corollary 4 already considers a shift of the fraction of patterns from $\tilde{\pi}$ during training to $\tilde{\pi}'$ during generation. Similarly, compositional generalization in [1] can be viewed as a change in the sampling distribution of $Z$ in Definition 1, corresponding to different pattern combinations. Extending our analysis to different pattern combinations during testing is an interesting direction for future work. We will cite these works and include the discussion in the revision.
>
> [1] Favero et al., ICML 2025. How Compositional Generalization and Creativity Improve as Diffusion Models are Trained.

---

> > ### Author Rebuttal · Reviewer_KLEe · 2026-04-03
> >
> > Thank you for this helpful rebuttal. I continue to have concerns about the extent to which this theory can make testable predictions about real-world diffusion models, so I maintain my weak accept rating. However, I agree with Reviewer 24th that the mathematical results are interesting in their own right, and I would be happy to see this work in this year's ICML program.

---

> > > ### Author Response · Authors · 2026-04-07
> > >
> > > Thank you for your response and for maintaining a positive score. We will incorporate the response in our revision. We appreciate that the reviewer recognizes the value of our theoretical contributions. The focus of this paper is to provide a theoretical analysis of the training dynamics and mechanisms of Transformer models in DDPM. Since prior work has not analyzed convergence under gradient-based optimization, even for simpler neural networks than Transformers, our work makes a unique contribution to the theoretical understanding of nonlinear models, including Transformers, in diffusion models. We agree with the reviewer that extending the analysis to more realistic and large-scale datasets and diffusion models, providing testable predictions aligned with the theory, and analyzing generalization, is an important future direction. Due to the substantial effort required for such extensions, we leave this as future work.

---

### Official Review · Reviewer_Fudt · 2026-03-12

**Soundness:** 4
**Presentation:** 3
**Significance:** 4
**Originality:** 4
**Overall Recommendation:** 6
**Confidence:** 4

**Summary:**

This paper focuses on a theoretical analysis of the properties of the Denoising Diffusion Probabilistic Model (DDPM) framework.
Specifically, it provides convergence analysis for training transformer-based diffusion models.
To that end, the paper assumes that the data distribution follows a multi-token Gaussian mixture.

The paper provides convergence guarantees supported by a simulation study and experiments on the simple MNIST database.

**Compliance With Llm Reviewing Policy:**

Affirmed.

**Final Justification:**

The paper presents a convergence analysis for transformer-based DDPM. Given the lack of such an analysis in the literature, the paper may serve as a stepping stone for future work. The authors have properly addressed all my concerns.
My recommendation is "strong accept" (6).

**Key Questions For Authors:**

1. How will your results change with a more complex transformer architecture?
Indeed, this is mentioned in the future work, and I do not expect a detailed analysis; rather, I expect more intuition.
2. How general is the Multi-Token Gaussian Mixture (MTGM) data distribution and why was it chosen?

**Limitations:**

Yes

**Strengths And Weaknesses:**

Strengths:
1. The transformer-based DDPM framework is a very popular generative model. It lacks a thorough convergence analysis, a gap that the paper addresses.
2. The data distribution is much richer than distributions usually considered in the literature.
3. Convergence analysis of the training stage.

Weaknesses:
1. The transformer architecture and the data distribution are still simplified. However, given the lack of such an analysis, it may serve as a stepping stone for future work.
2. Although consisting of real data, MNIST is a very simple database.

---

> ### Author Rebuttal · Authors · 2026-03-31
>
> We thank Reviewer KLEe for the time and effort in the evaluation. We address your concerns one by one as follows.
>
> **Q1 (Weakness 1 \& Questions 1 \& 2): The transformer architecture and the data distribution are still simplified. How will your results change with a more complex transformer architecture? How general is the Multi-Token Gaussian Mixture (MTGM) data distribution and why was it chosen?**
>
> **A1**: Thank you for the insightful question. We would like to first emphasize that this paper aims to provide a theoretical understanding and convergence analysis of DDPM training with Transformers, which requires appropriate assumptions to make the analysis tractable. Regarding the data, our assumptions are already more general and better capture key characteristics of real data compared to prior work (see our **[A1](https://openreview.net/forum?id=6VyoaDj3zB&noteId=aYHXcKiej9)** to Reviewer Yw16 for details). Regarding the architecture, we would like to elaborate more on how our results will change with a more complicated Transformer, which could potentially include the extension of adding MLP layers, training $W_Q$ and $W_K$ separately, and studing multi-layer model. For adding the MLP layer, we expect the optimal parameters still learn the oracle MMSE (Eqn. 13) by approximately acting as a linear block. Please see our **[A3](https://openreview.net/forum?id=6VyoaDj3zB&noteId=aYHXcKiej9)** to Reviewer Yw16 for details. For optimizing $W_Q$ and $W_K$ separately, we need more assumptions or advanced tools for the convergence analysis due to the more complicated optimization landscape. Please see our **[A3](https://openreview.net/forum?id=6VyoaDj3zB&noteId=exX8NgOkuJ)** to Reviewer 24th for details. For deeper Transformer architectures, we expect the main qualitative insights to hold, for example, the convergence is still faster under a more uniform distributions of patterns. However,  the analysis will become more complex due to layer-wise interactions and compositional representations. Please see our **[A3](https://openreview.net/forum?id=6VyoaDj3zB&noteId=aYHXcKiej9)** to Reviewer Yw16 for more details.
>
> **Q2 (Weakness 2): Although consisting of real data, MNIST is a very simple database.**
>
> **A2**: We thank the reviewer for this comment. We agree that MNIST is a relatively simple dataset. Our goal in including this experiment is to provide a controlled and interpretable setting that aligns with our theoretical assumptions, allowing us to validate the predicted behaviors of the model. Specifically, MNIST enables us to associate classes with patterns and introduce imbalance, which is important for testing the key mechanisms identified in our analysis. We believe this experiment demonstrates consistency between theory and practice. Similar simplified datasets are also used in previous theoretical works (Han et al., 2025). Extending the empirical results to more complex datasets is an interesting direction for future work.

---

> > ### Author Rebuttal · Reviewer_Fudt · 2026-03-31
> >
> > I am satisfied with the response.

---

> > > ### Author Response · Authors · 2026-04-07
> > >
> > > We are glad your concerns have been adequately addressed. We will incorporate the response in our revision. Thank you again for your suggestions!

---

### Decision · Program_Chairs · 2026-04-30

**Decision:**

Accept (regular)

**Comment:**

The paper studies how transformers behave when trained with a diffusion model objective. The analysis is done under a Gaussian mixture setting, where each data point is composed of multiple tokens drawn from different components. The main result is a convergence guarantee showing that a one-layer, single-head transformer trained with gradient descent can reach a near-optimal denoiser.

Reviewers find the analysis strong and not trivial. They also like that the paper uses a richer data model than earlier theory papers. The link between training and the final denoising behavior is clear, given that the the interpretation of attention as a form of mean denoising across tokens in which the self-attention mechanism approximately outputs the mean of the tokens that share the same underlying pattern. Some reviewers also say that the proof itself is technically strong and already a contribution on its own.

However, concerns are also raised. There are strong simplifying assumptions, such as using only one layer and one attention head, as well as orthogonality assumptions in the data. It is not clear how the results translate to practical models, or how they lead to useful insights for people building diffusion systems. After the rebuttal, the authors clarified why they made these assumptions and positioned their work as a first step.

In any case, several reviewers remain positive. They point out that these simplifications are common in theoretical deep learning, and see the contribution as meaningful for the theory community. Overall, even if the practical impact is still unclear, I lean toward acceptance given the theoretical contribution and its relevance for the community.